

# Wet deposition in the remote western and central Mediterranean as a source of trace metals to surface seawater

Karine Desboeufs[1], Franck Fu[1], Matthieu Bressac[2,3], Antonio Tovar-Sánchez[4], Sylvain Triquet[1],

Jean-François Doussin[5], Chiara Giorio[6,7], Patrick Chazette[8], Julie Disnaquet [9,10], Anaïs Feron[1],

Paola Formenti[1], Franck Maisonneuve[5], Araceli Rodríguez-Romero[4], Pascal Zapf[5], François

Dulac[8] and Cécile Guieu[3]

[1] Université de Paris and Univ Paris Est Creteil, CNRS, LISA, UMR 7583, F-75013 Paris, France

[2] Institute for Marine and Antarctic Studies, University of Tasmania, Hobart, Tasmania, Australia.

[3] Laboratoire d'Océanographie de Villefranche (LOV), CNRS-Sorbonne Université, INSU, Villefranche-sur-Mer, 06230, France.

[4] Department of Ecology and Coastal Management, Institute of Marine Sciences of Andalusia (CSIC), 11510 Puerto Real, Cádiz, Spain.

[5] Univ Paris Est Creteil and Université de Paris, CNRS, LISA, UMR 7583,  F-94010 Créteil, France

[6] Laboratoire de Chimie de l'Environnement (LCE), UMR 7376 CNRS, Aix-Marseille Université, Marseille, 13331, France.

[7] Yusuf Hamied Department of Chemistry, University of Cambridge, Lensfield Road, CB2 1EW, Cambridge, United Kingdom

[8] Laboratoire des Sciences du Climat et de l'Environnement (LSCE), UMR 8212 CEA-CNRS-UVSQ, Institut Pierre-Simon Laplace, Univ. Paris-Saclay, 91191 Gif-sur-Yvette, France.

[9] Marine Biology Research Division, Scripps Institution of Oceanography, UCSD, USA

[10] Sorbonne Université, CNRS, Laboratoire d'Océanographie Microbienne, LOMIC, France

*Correspondence to*: Karine Desboeufs (karine.desboeufs@lisa.ipsl.fr)





**Abstract.** This study reports the only recent characterisation of two contrasted wet deposition events
collected during the PEACETIME cruise in the Mediterranean open seawater, and their impact on
trace metals (TM$_S$) marine stocks. Rain samples were analysed for Al, 12 trace metals (TMs
hereafter, including Co, Cd, Cr, Cu, Fe, Mn, Mo, Ni, Pb, Ti, V and Zn) and nutrients (N, P, DOC)
concentrations. The first rain sample collected in the Ionian Sea (rain ION) was a wet typical
regional background deposition event whereas the second rain collected in the Algerian Basin (rain
FAST) was a Saharan dust wet deposition. The concentrations of TMs in the two rain samples were
significantly lower compared to concentrations in rains collected at coastal sites reported in the
literature, suggesting either less anthropogenic influence in the remote Mediterranean environment,
or decreased emissions during the last decades in the Mediterranean Sea. The TMs inventories in
the surface microlayer and mixed layer (0-20m) at ION and FAST stations before and after the
events, compared to atmospheric fluxes, showed that the atmospheric inputs were a significant
source of particulate TMs for both layers. At the scale of the western and central Mediterranean, the
atmospheric inputs were of the same order of magnitude as marine stocks within the ML for
dissolved Fe, Co and Zn, underlining the role of the atmosphere in their biogeochemical cycle in
the stratified Mediterranean Sea. In case of intense wet dust deposition event, the contribution of
atmospheric inputs could be critical for dissolved stocks of the majority of TMs.



## 1. Introduction

Atmospheric deposition of continental aerosol has long been recognized to influence trace element concentrations in remote oceanic surface waters (Buat-Ménard and Chesselet, 1979; Hardy, 1982; Buat-Ménard, 1983). In particular, the Mediterranean Sea (Med Sea) is an oligotrophic environment where marine biosphere growth is nutrient-limited during the long Mediterranean summer season characterized by a strong thermal stratification of surface waters (The Mermex Group, 2011). The Mediterranean atmosphere is characterized by the permanent presence of anthropogenic aerosols from industrial and domestic activities around the basin (e.g., Sciare et al., 2003; Kanakidou et al., 2011). In addition to this anthropogenic background, the Mediterranean basin is also subject to seasonal contributions of particles from biomass fires in summer (Guieu et al., 2005) and to intense sporadic Saharan dust inputs (e.g., Loÿe-Pilot and Martin, 1996; Vincent et al., 2016). Several studies emphasized that the atmospheric deposition of aerosols, notably through wet deposition, plays a significant role in marine cycles of both nutrient, such as nitrogen (N) and phosphorus (P) (e.g. Pulido-Villena et al., 2010; Richon et al., 2018 a and b; Violaki et al., 2018) and micronutrients, such as iron (Fe) (Bonnet and Guieu, 2006). Recently, atmospheric dust inputs were identified to have a fertilizing effect on the plankton stocks and fluxes, even in the presence of relatively high nutrients and Fe marine concentrations (Ridame et al., 2011; reviewed in Guieu and Ridame, 2021). Mackey et al. (2012) showed that TMs provided by dust deposition could explain this fertilizing effect. Indeed, some TMs, including Mn, Co, Ni (Mackey et al., 2012), Cu (Annett et al., 2008) and Zn (Morel et al., 1991), play physiological roles for phytoplanktonic organisms. These TMs are present in very low concentrations in oligotrophic systems such as the Med Sea, possibly limiting the phytoplankton growth (Pinedo-Gonzàles et al. 2015) and implying the role of dust deposition as source of TMs for planktonic communities. On the other hand, atmospheric deposition of European aerosol particles was identified to have a negative effect on chlorophyll concentrations (Gallisai et al. 2014), by providing trace metals, as Cu, at toxic levels (Jordi et al. 2012).

The atmospheric deposition of TMs in the Mediterranean is related to both dust and anthropogenic aerosols deposition (Desboeufs et al., 2018). The role of dust deposition as a source of TMs was observed from the correlation between the atmospheric deposition of mineral dust, and the enrichment of dissolved TMs (Cd, Co, Cu, Fe) in the Mediterranean Sea surface microlayer (Tovar-Sánchez et al., 2014). For the water column, the adding of dissolved Fe and Mn was emphasized in mesocosm experiments after dust addition mimicking intense wet dust deposition (Wuttig et al., 2013). Yet, the direct impact of wet deposition events on TMs concentrations in surface seawater has never been studied *in situ*, whether in the Mediterranean or in other oceanic regions. Moreover,





the two key criteria used to assess the potential impact of TMs and nutrients wet deposition are their
respective concentration (or flux) and solubility, i.e., the partitioning between dissolved and total
concentrations in rainwaters. Indeed, it is considered that the dissolved fraction of nutrients and
TMs can be directly assimilated by the phytoplankton. Few studies focussed on concentrations of
TMs in rainwater samples was collected around the Mediterranean basin: Al-Momani et al., (1998),
Kanellopoulou, (2001) and Özsoy and Örnektekin, (2009) in the eastern basin, and Losno (1989),
Guieu et al. (1997), Frau et al., (1996), Chester et al., (1997) and Guerzoni et al. (1999b) in the
western basin. These studies led to highly variable TMs concentration and solubility, illustrating a
more general large variability of TMs inputs during wet depositions in the Mediterranean Sea
(reviewed in Desboeufs, 2021). All these studies were performed at coastal sites. Offshore samples
of rainwater have rarely been studied in the literature so far. In the Mediterranean, to our knowledge,
trace element concentrations from only three rain samples collected at sea in April 1981 have been
reported in a PhD thesis (Dulac, 1986). However, due to the continental and local source of pollution
and the variety of anthropogenic aerosol sources (Amato et al., 2016), the TMs rain composition of
the coastal zone could not be representative of atmospheric deposition in the remote Mediterranean.
The PEACETIME cruise (ProcEss studies at the Air-sEa Interface after dust deposition in the
MEditerranean Sea) performed in spring 2017 aimed at studying the impacts of atmospheric
deposition, in particular Saharan dust events, on the physical, chemical and biological processes in
this marine oligotrophic environment (Guieu et al., 2020). We investigated here the concentration
and solubility of TMs and nutrients of two rainwaters sampled in the Mediterranean open sea during
the cruise. We compare our results on TMs concentrations and solubility in rainwater with previous
studies based on rainwater samples collected at coastal sites to investigate potential differences with
the open sea. Additionally, to assess the impact of wet deposition on the surface TMs
concentrations, surface seawater including the surface micro-layer was concurrently collected for
the first time with rain samples.
**2. Sampling and methods**
**2.1 Sampling and chemical analysis of rainwater**
The PEACETIME oceanographic campaign (https://doi.org/10.17600/17000300) took place in the
western and central Mediterranean Sea on-board the French research vessel (R/V) *Pourquoi Pas ?*
between 11 May and 10 June 2017, i.e. at the beginning of the Mediterranean stratification season
(Guieu et al., 2020).  The rain collector was installed on the upper deck (22 m above sea level)


where no on-board activities were taking place to avoid contamination. A rain collector was
equipped with an on-line filtration system to directly separate the dissolved and particulate fractions
(details of the filtration system are available in Heimburger et al., 2012) allowing for the calculation
of solubility of TMs in the rainwater at the time of collection. The filtration device was equipped
with a Nuclepore® polycarbonate membrane (PC) filter (porosity: 0.2 μm, diameter: 47 mm), and
the diameter of the funnel of the collector was 24 cm. The rain collectors were installed only when
rain was expected and kept closed within an acid-washed sealed plastic film until the rainfall began.
All the sampling materials were thoroughly washed in the laboratory prior to the cruise departure
(washing protocol described in Heimburger et al., 2012). No stabilizing setup was used to keep the
funnel level during the pitch and roll of the ship, preventing a precise assessment of the height of
rainfall from the collected water volume. During the rain sampling, the ship was always facing the
wind to avoid contamination by the smoke of the ship itself, as the chimney was situated on the
lower deck and behind the collector.
Immediately after sampling, the collector was disassembled under the laminar flow hood inside a
clean-room on-board laboratory. The dissolved fraction was separated in 4 aliquots dedicated to i)
dissolved organic carbon (DOC) by high-temperature catalytic oxidation (HTCO) on a Shimadzu
total organic carbon analyzer (as described in Van Wambeke et al 2021), ii) major ions by ion
chromatography (IC), iii) metals analysis by inductively coupled plasma methods (ICP), and iv) pH
measurement. For ICP measurements, the sample was acidified immediately to 1% by volume of
ultra-pure nitric acid: 67-69%, Ultrapur, Normatom®, VWR. For IC analysis, the sample was
immediately frozen. The filter (particulate fraction) was dried under the laminar flow hood, and
then put in a storage box and packed with a plastic bag to avoid contamination. After returning to
the laboratory, filters were acid digested by using the adapted protocol from Heimburger et al.
(2012) as follows: filters were placed in tightly capped Savillex™ PFA digestion vessels with 4 mL
of a mixture of $HNO_3$ (67-69%, Ultrapur, Normatom®, VWR), $H_2O$ and HF acids (40%, Ultrapur,
Normatom®, VWR) in a proportion (3: 1: 0.5), then heated in an oven at 130°C for 14 hours. After
cooling, acid was completely evaporated on a heater plate (ANALAB, 250, A4) at 140°C for about
2h, then 0.5 mL of $H_2O_2$ (30-32%, Romil-UpA™) and 1 mL of the acidified water (2% $HNO_3$) was
added to the vessels and heated during 30 min. to dissolve the dry residue in the bottom of the
vessels; finally, 12 mL of acidified water (1% $HNO_3$) was added to obtain 13.5 mL of solution in a
tube for ICP-MS analyses.
The dissolved fraction was analysed by IC (IC 850 Metrohm) for the inorganic and organic anions
($NO_2^-$, $NO_3^-$, $PO_4^{3-}$, $SO_4^{2-}$, $F^-$, $Cl^-$, $Br^-$, $HCOO^-$, $CH_3COO^-$, $C_2H_5COO^-$, MSA, $C_2O_4^{2-}$) and for the





cation $NH_4^+$ (Mallet et al., 2017). On the other hand, the dissolved fraction and solution from
digestion of particulate fractions were analysed by ICP-AES (Inductively Coupled Plasma Atomic
Emission Spectrometry, Spectro ARCOS Ametek®) for major elements (Al, Ca, K, Mg, Na, S)
(Desboeufs et al., 2014) and by HR-ICP-MS (High Resolution Inductively Coupled Plasma Mass
Spectrometry, Neptune Plus ™ at Thermo Scientific ™) for TMs: Cd, Co, Cr, Cu, Fe, Mn, Mo, Ni,
P, Pb, Ti, V and Zn. The speciation of dissolved P was estimated by determining the dissolved
inorganic phosphorus (DIP) from phosphate concentrations expressed as P and the dissolved
organic phosphorus (DOP) from the difference between total dissolved phosphorus (TDP), obtained
by ICP-MS, and DIP, obtained by IC.
In order to estimate the contamination of sampling and analytical protocols, 3 blanks of rain samples
(collected on-board during the cruise with the same protocol without rain events) were used and
processed. The procedural limit of detections (LoD) were defined as 3 x standard deviation of blank
samples both for dissolved and particulate fractions estimated after acid digestion. All samples
dissolved and particulate concentrations were higher than LoD, except for $NO_2^-$ in the two rain
samples. The blank concentrations represented 10.2% in average for TMs and were typically lower
than 20% of the sample concentrations, except for Cd (52%) and Mo (43%) in the dissolved fraction.
For the sample concentration computations, we subtracted these blanks values to elemental
concentrations obtained in rain samples.
**2.2 Atmospheric ancillary measurements**

The PEGASUS (PortablE Gas and Aerosol Sampling UnitS, www.pegasus.cnrs.fr) mobile platform
of LISA is a self-contained facility based on two standard 20-feet containers, adapted with air-
conditioning, rectified power, air intake and exhausts for sampling and online measurements of
atmospheric aerosols and gaseous compounds, and their analysis (Formenti et al., 2019). During the
PEACETIME cruise, only the sampling module of the facility was deployed on the starboard side
of deck 7 of the R/V. The PEGASUS instrumental payload of relevance to this paper included
measurements of the major gases such as NOx, $SO_2$, $O_3$ and CO by online analysers (Horiba APNA,
APSA, APOA and PICARRO respectively; 2-min resolution, detection limit 0.5 ppb and 1 ppb for
CO), that were used to estimate origins of collected air masses.
From the first of June (not operational before), additional measurements by an ALS450® Rayleigh-
Mie lidar (Leosphere ™; Royer et al., 2011) was used to monitor the vertical distribution of aerosols
over time and the associated integrated columns. The vertical lidar profiles were analysed to yield
the apparent backscatter coefficient (ABC) corrected from the molecular transmission, as well as



the volume depolarisation ratio (VDR). The inversion procedure (Chazette et al., 2016; 2019) to
retrieve the aerosol extinction coefficient (unit $km^{-1}$) uses a vertical-dependent lidar ratio that takes
into account two aerosol layers. The first layer corresponds to marine aerosols in the marine
boundary layer (MBL), the second to a desert aerosol layer that can extend between ~1 and 6 km
above mean sea level (amsl). In accordance with Chazette et al. (2016), for the same wavelength
and region, the lidar ratios were set to 25 and 55 sr, respectively. The vertical profile of the aerosol
extinction coefficient was retrieved from 0.2 km amsl on with a vertical resolution of 15 m. Based
on these profiles, the integrated column content of dust aerosols was estimated using a specific
extinction cross-section of 1.1 $m^2$ $g^{-1}$ as proposed by Raut and Chazette (2009).
In addition, detailed meteorological data such as air and sea temperature, atmospheric pressure,
relative humidity, atmospheric pressure, heat flux and wind speed and direction were provided on
a 30 seconds timestep basis by the ship's permanent instrumentation.

**2.3 Sampling and analysis of dissolved TMs in seawater**

Before and after each rain, seawater samples were collected in the surface microlayer (SML: <1
mm) and subsurface seawater (SSW: <1-m depth) (Tovar-Sánchez et al., 2020, this special issue).
SML and SSW samples were collected from a pneumatic boat 0.5-1 nautical mile away from the
R/V in order to avoid any potential contamination. SML samples were collected using a glass plate
sampler (Stortini et al., 2012; Tovar-Sánchez et al., 2019) which had previously been cleaned with
acid overnight and rinsed thoroughly with ultrapure water (MQ-water). The 39 x 25 cm silicate
glass plate had an effective sampling surface area of 1950 $cm^2$ considering both sides. In order to
check for procedural contamination, SML blanks were collected at some stations on board of the
pneumatic boat by rinsing the glass plate with ultra-pure water and collecting 0.5 L of ultra-pure
water using the glass plate system. The surface microlayer thickness was calculated following the
formula of Wurl (2009). SSW was collected by using an acid-washed Teflon tubing connected to a
peristaltic pump. The total fraction (i.e. T-SML) was directly collected from the glass plate system
without filtration in a 0.5 L acid cleaned LDPE bottles, while the dissolved fraction in the SML (i.e.
D-SML) and SSW (i.e. D-SSW) was filtered in situ through an acid-cleaned polypropylene cartridge
filter (0.22 μm; MSI, Calyx®).
TMs samples were also collected in the water column using the trace metals clean (TMC) rosette
before and after the rains (Bressac et al., 2021). Although rosette deployments were performed over
the whole water column, we focus here on the 0-20 m marine mixed-layer (ML). The water column
was sampled using the TMC titanium rosette mounted with 24 teflon-coated Go-Flo bottles.



Immediately after recovery, the Go-Flo bottles were transferred inside a class-100 clean laboratory
container. Seawater samples were directly filtered from the bottles through acid-cleaned 0.2-μm
capsule filters (Sartorius Sartobran-P-capsule 0.45/0.2-μm). All samples were acidified on board to
pH <2 with Ultrapure-grade HCl under a class-100 HEPA laminar flow hood. Metals (namely Cd,
Co, Cu, Ni, Mo, V, Zn and Pb) were pre-concentrated using an organic extraction method (Bruland
et al., 1979) and quantified by ICP-MS (Perkin Elmer ELAN DRC-e). In order to breakdown metal-
organic complexes and remove organic matter (Achterberg et al., 2001; Milne et al., 2010), total
fraction samples (i.e. T-SML) were digested prior to the pre-concentration using a UV system
consisting of one UV (80 W) mercury lamp that irradiated the samples (contained in quartz bottles)
during 30 min. The accuracy of the pre-concentration method and analysis for TMs was established
using Seawater Reference Material (CASS 6, NRC-CNRC) with recoveries ranging from 89% for
Mo to 108% for Pb. Due to the complexity of the analytical method, all the TMC samplings were
not analysed for these metals. Overall, 1 or 2 depths were obtained in the mixed layer (0-20 m).
Dissolved Fe and Al concentrations were also measured on board. Dissolved Fe concentrations were
measured using an automated Flow Injection Analysis (FIA) with online preconcentration and
chemiluminescence detection (Bonnet and Guieu, 2006), and dissolved Al concentrations using the
fluorometric method described by Hydes and Liss (1976). Sampling and analysis for dissolved Fe
and Al concentrations are fully described in Bressac et al. (2021), and covered at least 4 depths in
the 0-20 m mixed layer.
**2.4 Enrichment factor and solubility**
In order to better constrain the origin of TMs in the rain samples, their enrichment factor (EF; Rahn
1976) relative to the Earth's crust was calculated based on their total concentrations (dissolved +
particulate fractions) as:
$$EF = \frac{([X]/[Al])sample}{([X]/[Al])crust} \quad (1)$$

where $[X]/[Al]$ is the ratio between an element X and Al concentrations in rainwater samples (at
the numerator), and in the Earth's crust (denominator) from Rudnick and Gao (2003). Aluminium
is currently used as a reference element as it only has a crustal origin. For a given TM, EF >1
indicates an enrichment with respect to the average composition of the Earth's crust. To account for
the soil composition variability of mineral dust atmospheric sources, TMs with an EF value >10 are
considered derived from non-crustal sources (Rahn, 1976).
The relative solubility of TMs in the two rainwater events was calculated as:





$$S_X\% = \frac{[X]disssolved}{[X]total} \text{ x } 100 \qquad (2)$$

where $S_X\%$ is the relative solubility (in %) of an element X in the rainwater, $[X]_{dissolved}$ and $[X]_{total}$
are its soluble and total concentration, respectively.

**2.5 Atmospheric deposition fluxes**

Impacts on biogeochemical cycles and ecosystem functioning after a rain event occur on time scales
of a few days (2-3), and space scales of tens of km (about 20-50 km within the radius of the ship's
position). In the specific context of oceanographic cruising, the documentation of these impacts is
restricted to the vertical dimension at the prescribed temporal scale. In this vertical dimension, the
exchange of TMs across the ML was controlled both by atmospheric inputs over the R/V position
and by advection from surrounding water masses that may have been impacted by surrounding
rainfall. Therefore, we had to consider this process in our estimation of the atmospheric fluxes
contributions. For this purpose, the atmospheric fluxes have to be integrated to the extent of the rain
area that can impact the marine surface layers. We derived wet deposition fluxes by considering the
total precipitation accumulated during the duration of the rain over the area around the R/V location.
Thus, the wet deposition fluxes in our rain samples were calculated by multiplying the volume
weighted mean rainfall concentration by the total precipitation. The total precipitation of the rain
events was issued from the hourly total precipitation accumulated during the rain events over the
region from ERA5 ECMWF reanalysis (Herbasch et al., 2018) and from the rain rate composite
radar products from the European OPERA database (Saltikoff et al., 2019), when it is possible.
Although subject to uncertainties (Morin et al., 2003), a surface-based weather radar is probably the
best tool to estimate rainfall in the surroundings of the R/V. However, the OPERA database does
not include Italian radars, which anyway did not cover the central area of the Ionian Sea during the
cruise. ERA5 data are available on regular latitude-longitude grids at 0.25º x 0.25º resolution. The
accumulated precipitation was taken from the grid-points spanning the ship's location, more or less
0.25° around the central grid-point for integrating the regional variability. Surface rain rate radar
composite images were available every 15 minutes with a spatial resolution of 2 km x 2 km. The
accumulated precipitation was the sum of integral rain rates during the rain duration averaged over
the radar pixels spanning the ship location within a radius of about 25 km around the ship location.

**2.6 Stocks in the surface seawater**

For the surface microlayer (SML), stocks of TMs were estimated from the integration of TMs
concentration over the thickness of the layer. The thickness ranged from 32 to 43 μm and from 26
to 43 μm at ION and FAST, respectively (Tovar-Sanchez et al., 2020).





The trace metals stocks within the ML were calculated by trapezoidal integrations of marine
concentrations from SSW and TMC rosette samplings. The upper water column was stratified along
the cruise transect (Taillandier et al., 2020), with a  ML ranging from 7 to 21 m (11 to 21 m at ION
station and 11 to 19 m at FAST station (Van Wambeke et al., 2020). The ML depth (MLD)
fluctuations, for example due to wind peaks associated with rain events, could create rapidly
changing conditions of vertical advection from deeper waters. However, with no significant increase
in TMs concentrations being observed below the ML down to about 50 m (not shown), the
enrichment observed in the ML after rainfalls could not be attributed to any mixing with deeper
water due to high wind.  In consequence, stocks in the ML have been integrated over a constant
depth range of 0-20 m for comparison, as Bressac et al. (2021).  For Cu, Fe, Ni, Zn, stocks were
estimated both for the dissolved and particulate fractions in the SML and ML, for Co, Cd, Mo, Pb
and V for the dissolved fraction only in the ML and for both fractions in the SML and for Mn and
Ti only for the particulate fraction in the ML.
The partitioning coefficient between the particulate and dissolved phases (Kd
=[particulate]/[dissolved]) was used to investigate exchanges between dissolved and particulate
pools of TMs.
**3. Results**
**3.1 General conditions**
The general meteorological conditions during the cruise indicated that the ION and FAST stations
were highly affected by cloudy weather conditions. During these periods, 2 significant rains
occurred on the R/V position and have been collected: The first rain (hereafter Rain ION) was
collected during the 4-day ION station in the Ionian Sea in the early morning of 29/05 at 03:08
UTC, and the second rain (hereafter Rain FAST) occurred during the 5-day "Fast action" station,
(hereafter 'FAST') in the Algerian Basin during the night of 05/06 at 00:36 UTC (Table 1). The two
rain samples coincide with peaks in relative humidity and wind speed, and minima in air
temperature (not shown).
**Table 1: Information regarding the two rains collected during the PEACETIME cruise.**

| Sample | Sampling time | Station name (dates) and rain location | Estimated total precipitation |
|---|---|---|---|





| Rain ION | 29 May 2017, 03:08-04:00 (UTC) 05:08–06:00 (local time) | ION (25–29 May) 35.36°N, 19.92°E | 3.5 ±1.2 mm |
|---|---|---|---|
| Rain FAST | 05 June 2017, 00:36-01:04 (UTC) 02:36–03:04 (local time) | FAST (2–7 June) 37.94°N, 2.91°E | 6.0 ±1.5 mm |


### 3.1.1. Rain ION

The ERA 5 data reanalysis shows 2 periods of precipitation in the surrounding of the ship's position,
i.e. in the morning and evening of June 26 (not shown) and in the night between June 28 and 29, in
agreement with on-board visual observations. The rain event collected at ION was the product of a
large cloud system, covering an area of about 90 000 $km^2$ around the R/V position, spreading over
the Ionian and Aegean Seas (Fig. 1). No radar measurements being available for this area, the
accumulated rate was estimated from ERA 5 data reanalysis on the grid-point corresponding to the
ION station and was 3.5 ±1.2 mm around the R/V position (±0.25°). The wash-out of the
atmospheric particles was revealed by the decrease in aerosol number concentrations monitored on-
board from about 1900 to 300 part.$cm^{-3}$ (supplementary material Fig. S1). Air mass back-trajectories
showed that the scavenged air masses came from Greece both in the marine boundary layer and in
the free troposphere (Fig. S1). The satellite observations showed low aerosol optical thickness
during this period (not shown), meaning low amounts of aerosols in the atmospheric column. No
significant European pollution influence was monitored by on-board measurements during this
event, with major gas mixing ratio and aerosol concentrations in the average values of the cruise
(Fig. S1) and typical of clean atmospheric concentrations, i.e. under detection limit for NOx, 1.2
ppb for $SO_2$, 51 ppb for $O_3$, 80 ppb for CO and 3000 part.$cm^{-3}$. On this basis, this wet event was
representative of a Mediterranean background marine rain event.



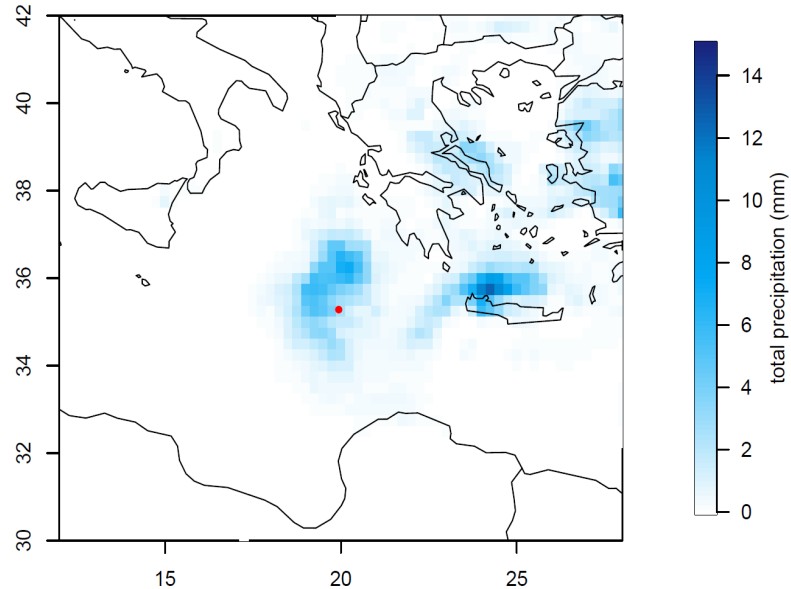


**Figure 1: Total precipitation (mm) between May 28 at 20:00 UTC and 29 at 10:00 UTC from ERA5**

**ECMWF reanalysis. The red circle indicates the R/V position.**

*3.1.2. Rain FAST*

As detailed in Guieu et al. (2020), the FAST station position was decided on the basis of regional
model forecast runs and satellite observations, in the purpose to catch a wet dust deposition event.
Significant dust emissions were observed from NASCube (http://nascube.univ-lille1.fr/, Gonzales
and Briottet, 2017) over North Africa from the night of 30-31 May, then new dust emissions in the
night from 3 to 4 June in Algeria and southern Morocco associated with a northward atmospheric
flux. On 30 May, the SEVIRI AOD satellite product (https://www.icare.univ-lille.fr/data-
access/browse-images/geostationary-satellites/, Thieuleux et al., 2005) confirmed the presence of
atmospheric dust in a cloudy air mass over the western part of the Mediterranean, and from 2 June
the export of a dust plume from North Africa south of the Balearic Islands with high AOD (>0.8)
on the Alboran Sea was observed (Fig. S2). The dust plume was transported to the NE up to Sardinia
on June 4, with AOD<0.5 in all the area and clear sky with low AOD was left west of 4°E on June
5.

On-board lidar measurements (Fig. 2 a,b,c) showed that the aerosol plume was present over the ship
position from 2 June at 21:00 UTC until the rain event, and corresponded to a dust aerosol layer
well highlighted by the high depolarization. The dust plume was concentrated between 3 and 4 km





at the beginning of the station, then expanded to the end of the day on June 3 down to the marine
boundary layer (about 500 m amsl). The mass integrated contents of dust aerosols derived from the
profiles of aerosol extinction ranged from a minimum of 0.18 ±0.005 g m$^{-2}$ just before the rain to a
maximum of 0.24 ±0.009 g m$^{-2}$, where standard deviations indicate the temporal variability (1
sigma).

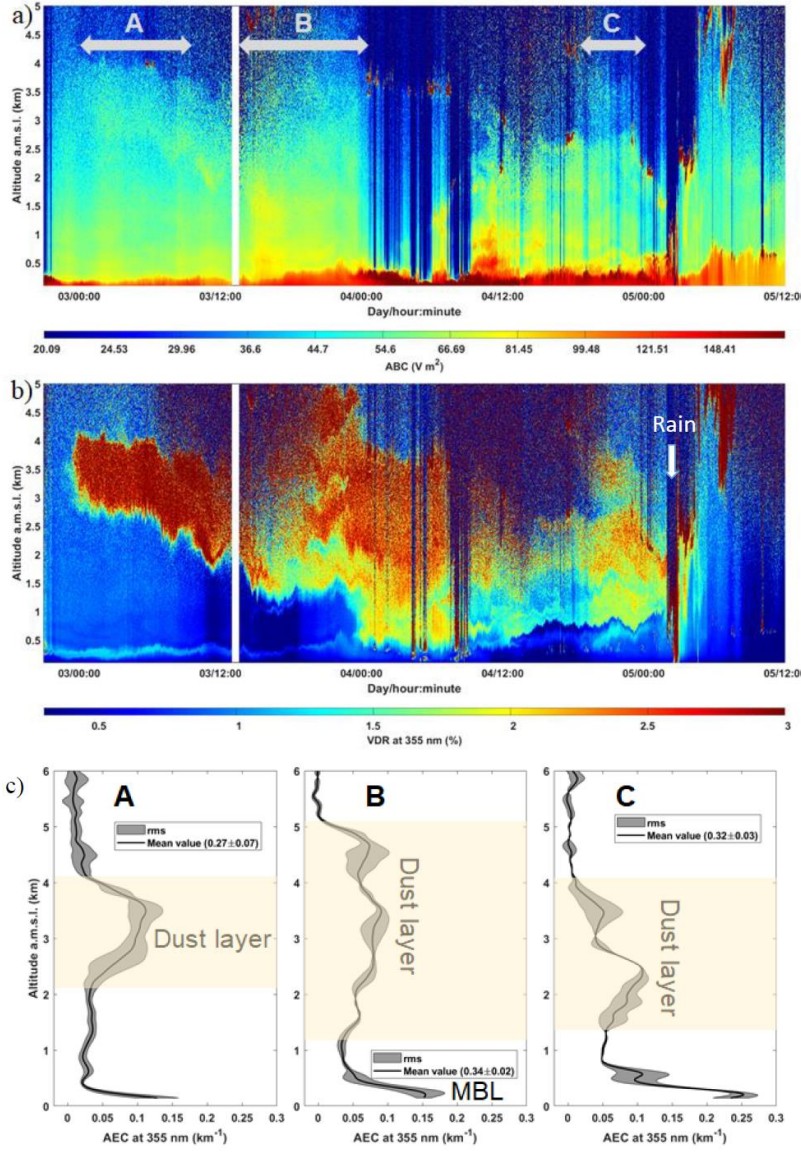






**Figure 2: On-board lidar-derived a) Apparent backscatter coefficient (ABC), (b) Temporal evolution (in Local Time) of the lidar-derived volume depolarization ratio (VDR) where the dust plume is highlighted for values higher than ~1.7 (yellow to red colors) and the rain by values higher than 3 (indicated by the arrow), and c) vertical profiles of the aerosol extinction coefficient (AEC) in cloud free condition, integrated over 3 periods along the dust plume event, noted A, B and C in figure a. The grey shade represents the root mean square (rms) variability along the time of the measurement. The dust layer is highlighted on the profiles. The mean aerosol optical thickness is given in the boxed legend with its temporal variability (1 sigma). The location of the marine boundary layer (MBL) is also pointed.**

Rainfalls were observed by weather radar images in the neighbouring area of the R/V from 3 June at 7:00 UTC. The rainfalls recorded around FAST station were associated with 2 periods of rain: the 03/06 from 7:00 to 14:00 UTC, and from the 04/06 at 16:00 UTC to 05/06 at 06:00 UTC. For this latter, a rain front (100 000 km$^2$), moving eastward from Spain and North Africa regions, reached the FAST station the night between the 4 and 5 June (Fig. 3). Wet deposition between the 4 and early 5 June in the FAST station area were confirmed by radar imagery, showing several other rain spots around the R/V position before and after the rain sampling (Fig. 3). Continuous on-board lidar measurements confirmed the below-cloud deposition during the rain event of early 5 June (Fig. 2b). Rain FAST was a wet deposition event occurring at the end of an episode of transport of Saharan dust, whereas precipitation of the 3 June occurred during the maximum of the dust plume (Fig. 2b and S2). The surface concentrations of gas and particles, measured on-board, suggest no clear dust or anthropogenic influence in the atmospheric boundary layer during the period of wet deposition, in agreement with back trajectories of low altitude air masses (Fig S2.), presuming no local mixing between dust and anthropogenic particles into rain samples. The total precipitation estimated from radar rainfall estimates yield an accumulated precipitation of 6.0 ±1.5 mm (±25 km around), in agreement with ECMWF reanalysis ERA5 (Fig. S2) for the wet deposition on the night of 4-5 June (5.7 ±1.4 mm in the grid-point spanning the R/V position, i.e. ±0.25° around).



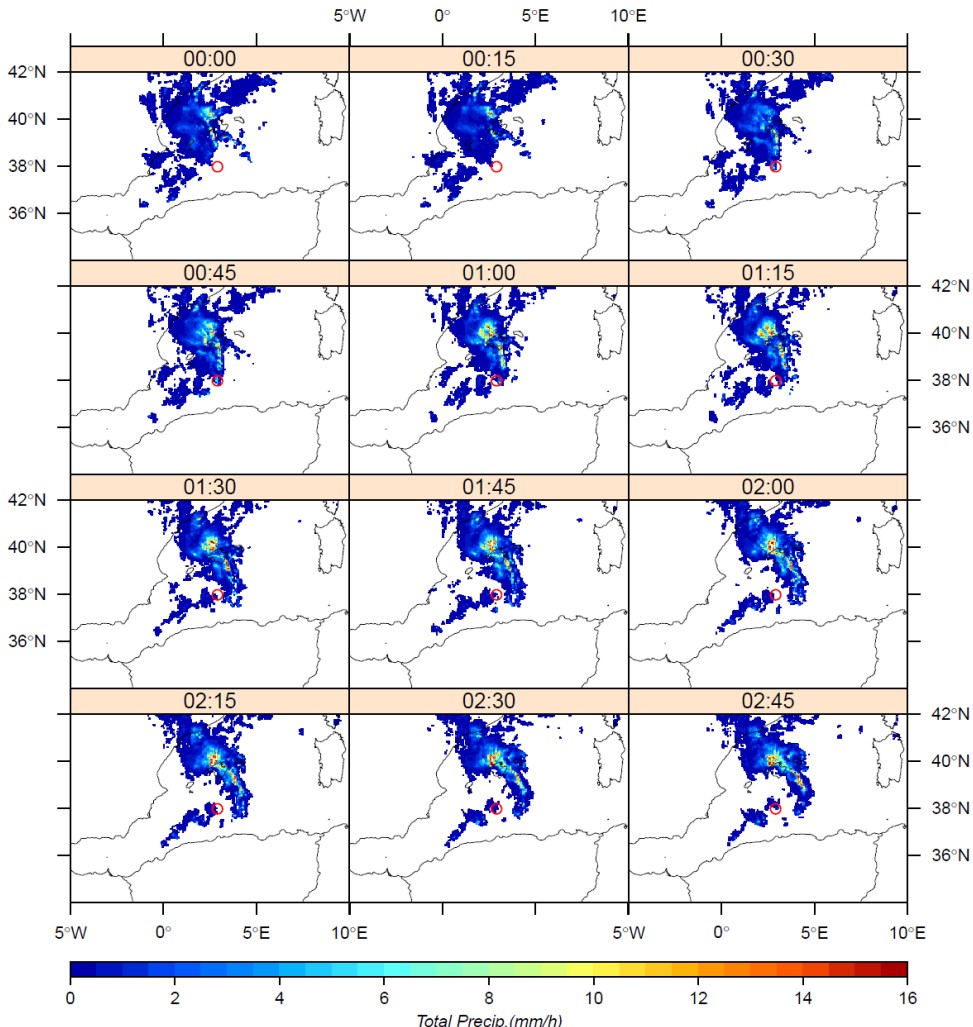

370

**Figure 3: Rain rates (mm/h) during the night between the 4 and 5 June, when Rain Fast has been**
**collected on board, issued from European rain radar composites (OPERA programme) of June 5**
**between 00:00 and 02:45 UTC.**

**3.2. Chemical composition of rains**

Dissolved and total concentrations of nutrients and TMs in the rain samples are presented in Table
2. Among all measured dissolved concentrations, $NO_3^-$ was the most abundant nutrient, followed
by ammonium (Table 2). The nitrite concentration was under the limit of detection for the two rain
samples. Regarding TMs in rain, Fe and Zn presented the highest concentrations in rain samples
with the same order of magnitude (10 to 25 µg $L^{-1}$). Co, Cd and Mo had the lowest concentrations





(<0.1 µg L⁻¹ in both events), whereas the other TMs concentrations ranged between 0.1 and 10 µg
L⁻¹ in both rains (Table 2). Concentrations of nutrients and the majority of TMs were higher in the
dust-rich rain, except Pb (similar concentrations in both rains) and Cr (3 times higher concentration
in Rain ION relative to Rain FAST).
**Table 2: Dissolved and total concentrations of nutrients and TMs in the two rains collected during the**
**PEACETIME cruise in µg.L⁻¹ or ng.L⁻¹ and µmol.L⁻¹ or nmol.L⁻¹ in the parentheses (sd = standard deviation**
**from three replicates).**

| | | | Rain ION | | | | Rain FAST | | | |
| | | | Dissolved | | Total | | Dissolved | | Total | |
| | | | concentrations | ±sd | concentrations | ±sd | concentrations | ±sd | concentrations | ±sd |
|---|---|---|---|---|---|---|---|---|---|---|
| Nutrients | $NO_3^-$ | µg L⁻¹(µmol L⁻¹) | 1185 (19.1) | 71 (1.1) | | | 3694 (59.6) | 222 (3.6 | | |
| | $NH_4^+$ | µg L⁻¹(µmol L⁻¹) | 366 (20.3) | 11 (0.6) | | | 654 (36.3) | 19 (1.1 | | |
| | DIN | µg L⁻¹(µmol L⁻¹) | 552 (39.4) | 82 (1.7) | | | 1343 (96) | 241 (17 | | |
| | $PO_4^{3-}$ | µg L⁻¹(nmol L⁻¹) | 18.1 (189) | 0.5 (6) | | | 19.0 (200) | 0.6 (6) | | |
| | DIP | µg L⁻¹(nmol L⁻¹) | 5.87 (189) | 0.18 (6) | | | 6.20 (200) | 0.19 (6) | | |
| | DOP | µg L⁻¹(nmol L⁻¹) | 8.63 (278) | 1.94 (75) | | | 4.91 (158) | 1.56 (57) | | |
| | TP | µg L⁻¹(nmol L⁻¹) | 14.51 (468) | 2.52 (81) | 16.6 (536) | 1.0 (33) | 11.11 (358) | 1.95 (63) | 58.7 (1894) | 3.5 |
| | DIN/DIP | | (208) | | | | (480) | | | |
| | DOC | (µmol L⁻¹) | (105.7) | (2.2) | | | (95.5) | (1.2) | | |
| Metals | Al | µg L⁻¹(nmol L⁻¹) | 13.0 (480) | 0.8 (30) | 14.6 (540) | 0.9 (32) | 23.4 (867) | 0.7 (24) | 440 (16308) | 7 |
| | Cu | µg L⁻¹(nmol L⁻¹) | 0.71 (11.1) | 0.02 (0.3) | 0.73 (11.5) | 0.02 (0.3) | 1.15 (18.0) | 0.04 (0.6) | 1.63 (25.7) | 0.06 |
| | Fe | µg L⁻¹(nmol L⁻¹) | 15.1 (270) | 0.4 (6) | 17.9 (321) | 0.6 (11) | 19.2 (344) | 0.1 (2) | 231 (4140) | 7 |
| | Mn | µg L⁻¹(nmol L⁻¹) | 0.55 (10.0) | 0.02 (0.3) | 0.60 (10.9) | 0.02 (0.4) | 3.17 (57.8) | 0.07 (1.2) | 5.26 (95.7) | 0.12 |
| | Ni | µg L⁻¹(nmol L⁻¹) | 0.52 (8.8) | 0.02 (0.3) | 0.67 (11.4) | 0.02 (0.4) | 0.59 (10.1) | 0.02 (0.4) | 0.84 (14.3) | 0.03 |
| | Ti | µg L⁻¹(nmol L⁻¹) | 0.48 (10.0) | 0.04 (0.8) | 0.65 (13.6) | 0.48 (3.2) | 0.22 (4.7) | 0.01 (0.1) | 33.36 (697) | 0.51 |
| | V | µg L⁻¹(nmol L⁻¹) | 0.37 (7.4) | 0.01 (0.2) | 0.38 (7.42) | 0.01 (0.25) | 1.37 (26.9) | 0.03 (0.5) | 2.02 (39.7) | 0.04 |
| | Zn | µg L⁻¹(nmol L⁻¹) | 24.8 (379) | 0.8 (12) | 25.3 (387) | 0.8 (12) | 22.7 (347) | 0.6 (8) | 26.3 (402) | 0.7 |
| | Cd | ng L⁻¹(pmol L⁻¹) | 12.9 (115) | 6.4 (57) | 13.1 (117) | 6.3 (56) | 20.2 (180) | 10.3 (92) | 23.7 (210) | 6.8 |
| | Co | ng L⁻¹(pmol L⁻¹) | 44 (749) | 13 (229) | 47.4 (804) | 14.5 (246) | 82 (1386) | 20 (347) | 157 (2661) | 28 |
| | Cr | ng L⁻¹(pmol L⁻¹) | 241 (4636) | 16 (300) | 628 (12079) | 5 (95) | 79 (1522) | 14 (260) | 443 (8514) | 43 |
| | Mo | ng L⁻¹(pmol L⁻¹) | 28 (288) | 10 (106) | 4.1 (43) | 1.4 (14) | 82 (855) | 11 (113) | 92.1 (960) | 16.2 |
| | Pb | ng L⁻¹(pmol L⁻¹) | 170 (822) | 11 (54) | 175.1 (845) | 1.4 (7) | 166 (801) | 9 (41) | 604 (2917) | 19 |


### 3.3. Marine concentrations and stocks

All the TMs had significantly higher concentrations in the ML compared to deep water masses, in
agreement with a stratified profile associated with atmospheric input. The particulate and dissolved
trace metal concentrations within the ML (0-20 m) and the SML are displayed in Fig. 4.
Concentrations were of the same order of magnitude in the two studied stations, except for the
particulate phase in the SML where the concentrations of Cu and Co were significantly lower at
ION station. The TMs were mainly in dissolved forms in the ML, except for Fe, whose dissolved
and particulate concentrations were in the same order of magnitude. On the contrary, the particulate
phase contribution dominated for TMs in the SML, in particular at the ION station. At both stations,
the highest TMs concentrations in the surface seawater were found for Mo in the dissolved fraction,



in agreement with the abundance of dissolved Mo in seawater (Smedley and Kinniburgh, 2017),
and Fe in the particulate fraction. All the particulate and dissolved TMs concentrations measured
during the cruise were representative for the Mediterranean Sea (Sherrell and Boyle, 1988; Saager
et al., 1993; Morley et al., 1997; Yoon et al., 1999; Wuttig et al., 2013; Baconnais et al., 2019;
Migon et al., 2020). Zn presented the largest range of concentrations within the ML both in the
particulate and dissolved phases, due to some high concentrations. However, the concentrations
stayed in the typical range of values found in the Mediterranean Sea (Bethoux et al., 1990, Yoon et
al., 1999). In the SML, the concentrations were lower than in the ML and Pb dominated both in
dissolved and particulate phases. Tovar-Sanchez et al. (2020) showed that the TMs concentrations
in the SML during the PEACETIME campaign were generally lower than those previously
measured in the Mediterranean Sea, except in the particulate phase during the FAST station after
dust deposition.

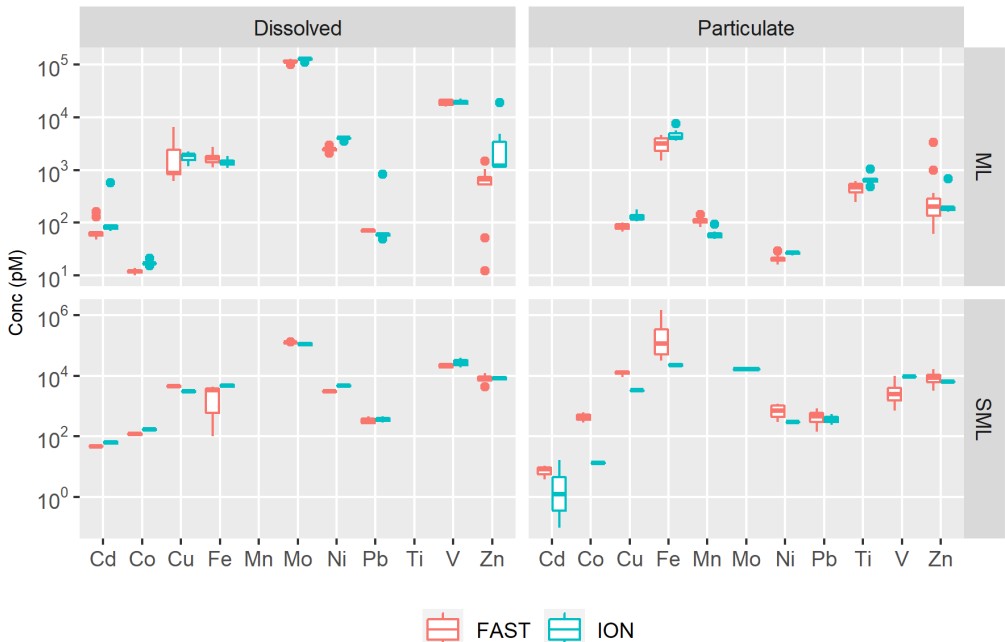


**411   Figure 4: Boxplots of dissolved (left panels) and particulate (right panels) marine concentrations (pM)**

**412   for the different TMs within the ML (upper panels) and the SML (lower panels) at ION and FAST**

**413   stations. In the box plots, the box indicates the interquartile range, i.e. the 25th and the 75 th percentile,**

**414   and the line within the box marks the median. The whiskers indicate the quartiles ±1.5 times the**

**415   interquartile range. Points above and below the whiskers indicate outliers outside the 10 th and 90th**

**416   percentile.**





## 4. Discussion

### 4.1. Composition of rain collected over the remote Med Sea

#### 4.1.1. Concentrations

Regarding nutrients, nitrogen species concentrations in rain samples were in good agreement with those reported in Mediterranean rain samples, ranging from 1130 to 5100 µg L$^{-1}$ for NO$_3^-$ and between 207 and 1200 µg L$^{-1}$ for NH$_4^+$ (Loye-Pilot et al., 1990; Avila et al., 1997; Al Momani et al., 1998; Herut et al., 1999; Violaki et al., 2010; Izquieta-Rojano et al., 2016; Nehir and Koçak, 2018). The FAST rain concentrations were in the average values, whereas the ION rain was in the low range, confirming a background signature. The rainwater samples presented a large dominance of N in comparison to P, as observed from the N/P ratio derived from DIN/DIP (Table 2) ranging from 208 at ION to 480 at FAST. Previous observations showed a predominance of N relative to P in the atmospheric bulk deposition over the Mediterranean coast, with ratio higher than Redfield ratio (Markaki et al., 2010, Desboeufs et al., 2018). The highest ratio observed reached 1200 (in case of DIN/TDP), but were on average around 100 in bulk deposition. The highest ratio could be linked to a washout effect of the gaseous N species (as NOx and NH$_3$) by rain (Ochoa-Hueso et al., 2011). At the two stations, no high NOx concentrations were observed in the boundary layer before wet deposition. The presence of nitrate and ammonium in the background aerosols has been emphasized during recent campaigns in the remote Mediterranean atmosphere (e.g. Mallet et al., 2019). To our knowledge, no data are available on both P and N concentrations in Mediterranean aerosols. The lowest concentrations of P relative to N in aerosol particles in Mediterranean have been observed during the cruise (Fu et al., in prep.). The TDP concentrations were consistent with the average value of 8.4 µg L$^{-1}$ measured in African dust rain samples collected in Spain over the 1996-2008 period (Izquierdo et al., 2012). Inorganic phosphorus predominated in the dust-rich rain, whereas organic P was dominant in the background rain as the contribution of DOP to the TDP was 60% and 44% in Rain ION and Rain FAST, respectively. The DOP/TDP ratio presents a very large range in Mediterranean rains, spanning from 6% in Spanish dusty rain samples (Izquierdo et al., 2012) to 75-92% in rains from Crete Islands (Violaki et al., 2018). A reason for this wide range could be that Mediterranean European aerosols, as opposed to Saharan dust particles, are dominated by organic phosphorus compounds associated with bacteria (Longo et al., 2014).





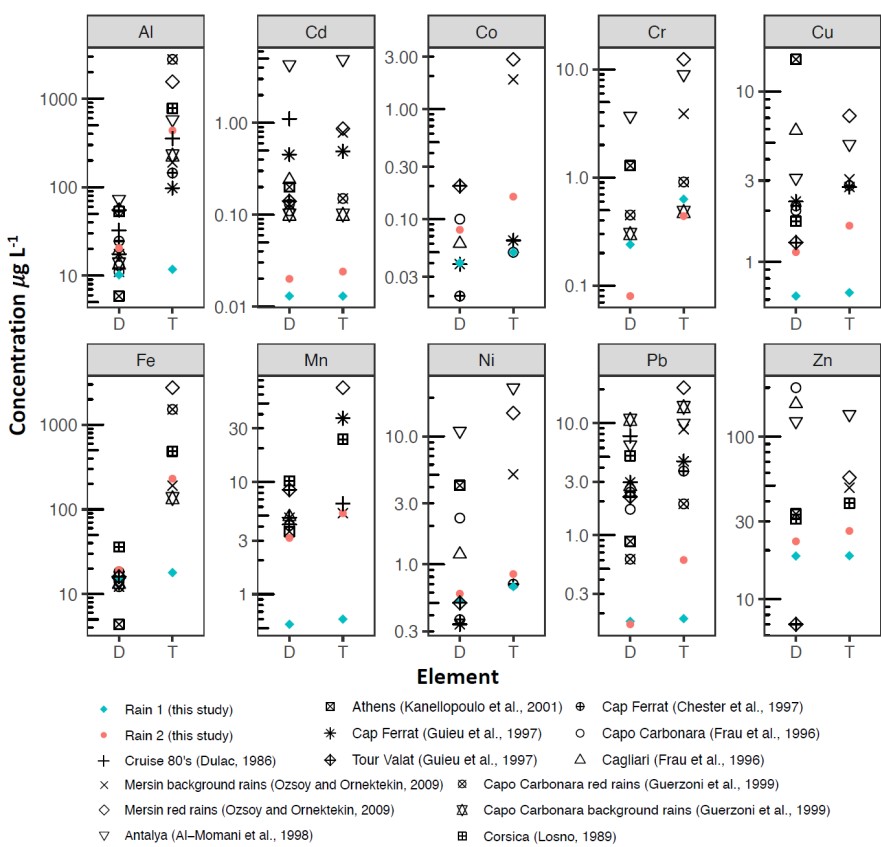


**Figure 5: Comparison of dissolved (D) and total (T) TMs concentrations along with data from 14 former studies**
**carried out in the eastern and western Mediterranean Sea.**
The dissolved and total TMs concentrations in the PEACETIME rains were lower than those
reported in coastal areas (eastern Basin: Özsoy and Örnektekin, 2009; Al-Momani et al., 1998;
Kanellopoulou et al, 2001 and western Basin: Guieu et al., 1997; Guerzoni et al., 1999b; Chester et
al., 1997; Losno, 1989; Frau et al., 1996) (Fig. 5), notably for the background Rain ION. This
suggests the probable effect of both local anthropogenic influence at coastal sites due to higher
aerosol concentrations in comparison to remote Mediterranean (Fu et al., in prep.) and due to the
reduction of anthropogenic emission for some elements since most of the referenced works on
coastal rainwaters date from the late 1990s. This is particularly true for Cd and Pb whose emissions
have strongly decreased over the last decades, notably due to removal of lead in gasoline and
reduction of coal combustion (Pacyna et al., 2007). This has resulted in a clear decrease in the
particulate concentrations of these metals in the Mediterranean atmosphere (Migon et al., 2008),



consistent with the fact that concentrations in the PEACETIME rains are one to two orders of
magnitude lower than reported in the literature. For these metals, the discrepancy is also observed
between the concentrations in our open-sea rain samples and the concentrations measured in three
rains collected at sea in April 1981 (Dulac, 1986), confirming the large decrease of concentrations
could be related with the decrease of emission. Thus, our results show the former literature cannot
be used as current reference about coastal rain composition due to recent environmental mitigation
on metals emissions.
*4.1.2. Enrichment factor*
EF and solubility values of TMs and P observed during the two rain events were very contrasted
(Fig. 6). In Rain ION, almost all elements were significantly enriched relative to earth crust (EF
>10, and up to ~$10^3$ for Cd and Zn), whereas in Rain FAST, only Zn (73), Cd (48) and Mo (15)
were slightly enriched. Only Ti, Fe, and Mn did not present a significant enrichment (EF < 10) in
Rain ION, in agreement with previous studies in the Mediterranean environment showing that these
metals are mainly associated with mineral dust in  atmospheric deposition (e.g., Guieu et al., 2010;
Desboeufs et al., 2018). Both rains, EF of Zn was in average five times higher than EF found in the
rains previously studied in the Mediterranean region (Özsoy and Örnektekin, 2009; Al-Momani et
al., 1998; Losno, 1989). However, extremely high enrichments of Zn in rainwater have also been
reported by Frau et al. (1996) with a geometric mean EF of about 6500 in both crust-rich and crust-
poor rains from two sites in southern Sardinia. Fu et al. (2017) also reported EF higher than 1000
for Zn in atmospheric bulk deposition in Lampedusa Island. The anthropogenic origin of particulate
TMs and P have been reported by several studies on atmospheric deposition in the western
Mediterranean (e.g., Guieu et al., 2010; Sandroni and Migon, 2002; Desboeufs et al., 2018). For
example, Desboeufs et al. (2018) showed that there is a large contribution of anthropogenic
combustion sources in P, Cr, V and Zn background deposition fluxes. Aerosol composition
monitoring over Mediterranean coastal area showed the role of land-based sources and ship traffic
sources on TMs contents (Bove et al., 2016; Becagli et al., 2017). However, all these sampling sites
were located in coastal areas, where it was difficult to discriminate the potential local influences.
Here, even if the on-board atmospheric gas and particle measurements did not show a specific
anthropogenic influence during the period of Rain ION, the particles scavenged by this rain
presented a clear anthropogenic signature for all TMs except Ti, Fe and Mn; however for Fe and
Mn, an influence of non-crustal sources in Rain ION is visible through a clear increase in the EF
values compared to FAST (Fig. 6). This means that even over remote Med Sea, the chemical





composition of background aerosol particles is likely continuously impacted by anthropogenic
sources.
Moreover, the EF values of TMs for Rain FAST were significantly lower than for Rain ION (Fig.
6) but similar to Saharan rains (Guerzoni et al., 1999b; Özsoy and Örnektekin, 2009). The
comparison between dust-rich and background rains generally reveals a net difference of
concentrations (at least higher by a factor 3 in dust-rich), notably for Al, Fe, Mn and Cr (Guerzoni
et al., 1999b; Özsoy and Örnektekin, 2009). Such contrast was indeed observed for Al, Fe and Mn
in the PEACETIME rains (Fig. 5), but also for Cu and Pb. The combination of higher concentrations
and EF values <10 found in Rain FAST confirm that the dust contribution was important on
deposition fluxes of many TMs.
*4.1.3. Solubility*
The solubility values were also larger in Rain ION than in the dusty Rain FAST, except for Mo for
which the difference between both rain samples is not significant (Fig. 6). For Rain ION, TMs and
P presented solubility higher than 78%, except for Cr (38%). In Rain FAST, solubility values <10%
were observed for Al and Fe, more than 10 times lower than in Rain ION. For the other TMs, the
highest difference in solubility was observed for Pb whose solubility decreased from 97% in Rain
ION to 27% in Rain FAST. In a review on TMs solubility in Mediterranean rainwaters collected in
coastal areas, Desboeufs (2021) emphasize the large range of solubility for all the TMs: Fe (0.8-
41%), Cr (6-80%), Pb (5-90%), Ni (22-93%), Mn (16-95%), Cu (22-96%), Zn (14-99%), V (35-
99%) and Cd (72-99%). The solubility ranges found in this study were generally consistent with
those reviewed by Desboeufs (2021). In particular, the Mn solubility values in FAST (60%) and
ION (92%) rains are close to those reported by Dulac (1986) from a dust-rich (57%) and an
anthropogenic (83%) rain collected at sea in the Ligurian Sea and west of Sardinia in April 1981,
respectively. Only Fe solubility (84%) found in Rain ION was higher than the average values
previously reported. In the Rain FAST, Fe solubility was 8%, this is 10 times lower than average
Fe solubility in 10 dust-rich rains collected on the southeastern coast of Sardinia by Guerzoni et al.
(1999b), but consistent with Saharan dust wet deposition collected in the Atlantic Ocean (Chance
et al. 2015; Powell et al., 2015; Baker et al., 2017).

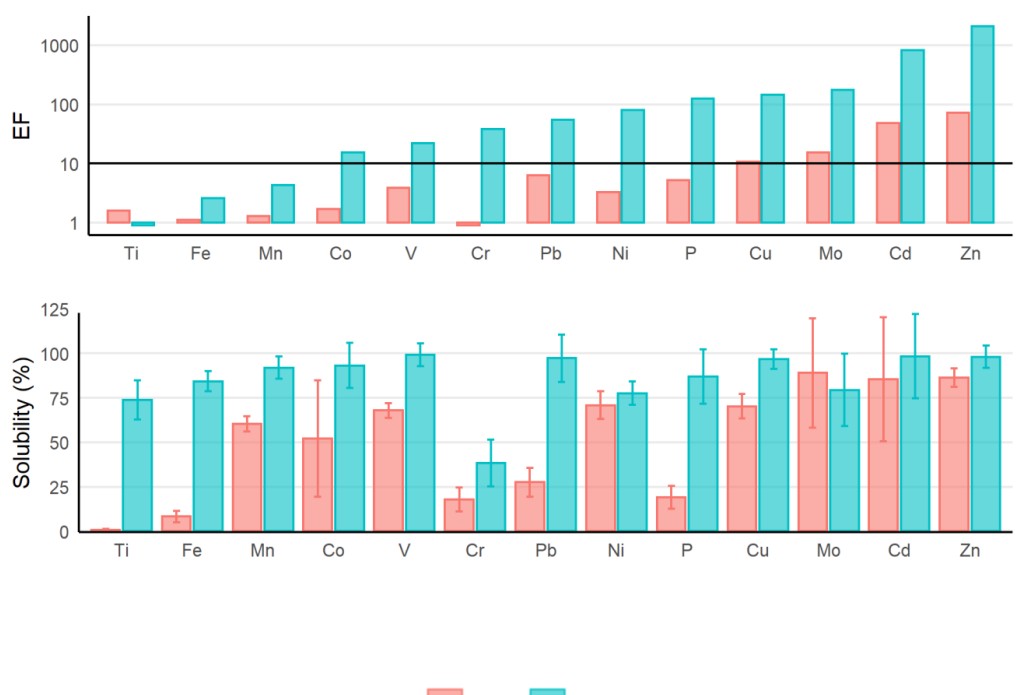

**Figure 6: Enrichment Factors (EF, upper panel) and solubility (%, bottom panel) of phosphorus (P) and TMs ordered by increasing EF in the two rainwater samples.**

Few studies compared TMs solubility between dust-rich and background rains in the Mediterranean. In Sardinia, Guerzoni et al. (1999b) observed an increase in solubility values from dust-rich to background rains for Al, Cr, Fe and Pb (and hardly for Cd), and reported an inverse relationship between the particle concentration and solubility of Al, Fe, Pb and Cd. Similarly, Theodosi et al. (2010) showed a decrease in TMs solubility with the increase in dust load in rains collected on Crete Island. In those two studies, this decrease was dependent on the considered metal, with Pb presenting the highest decrease in solubility. The decrease in solubility from background to dust-rich rains was also observed for P in Spain by Izquierdo et al. (2012), with values of solubility decreasing from 25% to 7%, and for Mn in offshore rains as mentioned above (Dulac, 1986). Metal partitioning in rainwater can be influenced by a number of parameters, such as pH, presence of dissolved organic complexing ligands, in-cloud processes, particle origin and load (Desboeufs et al., 1999; Bonnet and Guieu, 2004; Paris and Desboeufs, 2013; Heimburger et al., 2013). However, the particulate desert dust load, reflecting dust vs anthropogenic signature, is the main control of TMs solubility in the Mediterranean rainwater (Özsoy and Örnektekin, 2009; Theodosi et al., 2010). A much lower solubility of TMs in Rain FAST than in Rain ION (except for Mo) is consistent with





the EFs in Rain FAST (Fig. 6), although no correlation between solubility and EF values could be
observed. The case of Mo is unique, since its solubility was comparable in Rains ION and FAST
despite a >20 times higher EF in Rain ION. As Mo solubility is seldom studied in the literature, we
could not conclude on the reason for this particular outcome. It is interesting to note that despite the
desert signature, the majority of metals have solubility greater than 50% in rain FAST.
**4.2. Atmospheric wet deposition as a source of TMs to the surface seawater**
*4.2.1. Atmospheric fluxes*
As mentioned before, the two collected rains were part of large rain systems, associated with patchy
rainfalls that lasted several hours or days (section 3.1). This spatio-temporal variability led to
heterogeneity in both rainwater concentrations and accumulated precipitation across the studied
region. Such spatial variability has been observed by Chance et al. (2015) in the Atlantic Ocean.
Moreover, even weak lateral advection can transfer surface water impacted by intense precipitation
in the vicinity of the vessel. On this basis, the spatial extrapolation of wet deposition fluxes seems
subject to a large uncertainty when the rain samples are not collected across the rain area (Chance
et al., 2015). To best counteract this effect, spatial variability was taken into account to quantify  the
total precipitation i.e. 3.5 ±1.2 mm for rain ION and  6.0 ±1.5 mm for rain FAST (see 3.1) in order
to quantify the wet deposition fluxes.
From the total (dissolved + particulate) Al concentration measured in the rain FAST sample, we
estimated the wet mineral dust deposition flux at 65 ±18 mg m$^{-2}$, assuming 7.1% Al in dust (Guieu
et al., 2002). The vertical distribution of dust particles (Fig. 2b) and the absence of high Al
concentrations close to the sea surface (Fu et al., in prep.) indicate that dust dry deposition can be
neglected. Based on the increase in total Al in the upper 20 m of the column water following the
deposition events, Bressac et al. (2021) derived an average dust deposition flux of ~55 mg m$^{-2}$ at
FAST station, comparable to our estimate. Although low compared to deposition fluxes reported in
the western Mediterranean (Bergametti et al., 1989; Loÿe-Pilot and Martin, 1996; Ternon et al.,
2010), such flux values are among the most intense weekly dust deposition fluxes recorded more
recently in Corsica between 2011 and 2013 and is equivalent to the mean weekly flux observed in
Majorca Island during the same period (Vincent et al., 2016). The aerosol columnar during the dust
event being estimated between 0.18 and 0.24 g m$^{-2}$ (see 3.1.), the expected maximum values of
atmospheric dust flux could be in this range. The comparison with the estimated flux indicates that
the atmospheric column was probably not totally washed-out by the short rain event. Indeed, Fig.
2b shows that a significant depolarization was observed immediately after the rain ended on the





ship, before atmospheric advection could have brought dusty air possibly not affected by rain.
Satellite products (Fig. S2) confirm that on 5 June, the dusty air mass was transported farther to the
north-east from the station where it was replaced by clear air.

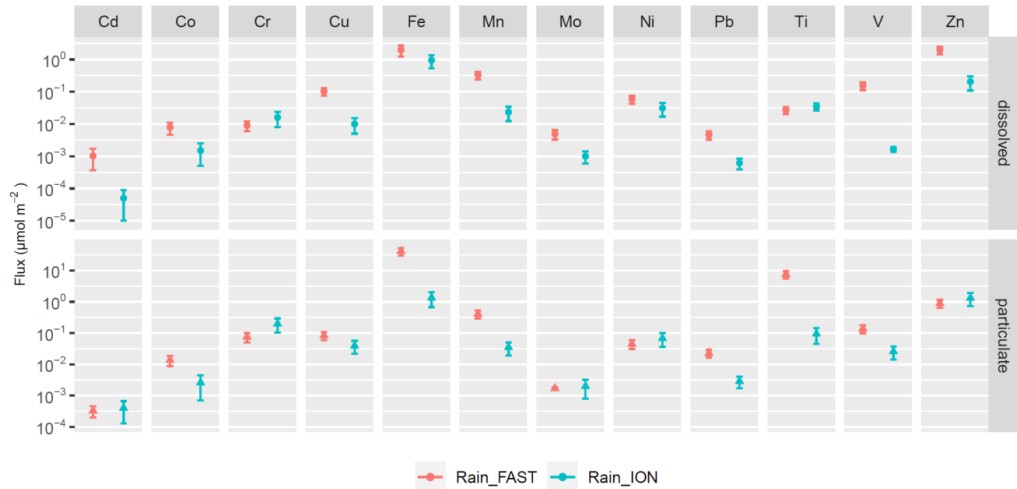


**Figure 7: Dissolved (upper panels) and particulate (lower panels) wet deposition fluxes (μmol m⁻²) for**
**the different TMs estimated from the two rains sampled on board, considering the standard deviation**
**on the TMs concentrations and the spatial variability of total precipitation over the area of sampling**
**(Rain ION in blue and Rain FAST in red).**
The atmospheric dissolved and particulate wet deposition fluxes of TMs, derived from the chemical
composition of rain samples and total precipitation, are presented in Fig. 7. Co, Mo and Cd
presented the lowest fluxes in the two rainfalls. Zn and Fe fluxes were in the same order of
magnitude and were the highest dissolved fluxes compared to the other TMs in the two rains. The
comparison showed that almost all the dissolved TMs fluxes were higher in the dusty rain, except
Cr and Ti. For the particulate phase, the fluxes were mainly increased by dust deposition for Co,
Fe, Mn, Pb, Ti and V. Our results emphasized that dust deposition, even here in case of a moderate
deposition flux, enabled higher atmospheric inputs of TMs than a low perturbed-anthropogenic
background rain. This is notably the case for dissolved Cd, Cu, Mn, V and Zn and for particulate
Fe and Ti with more than one order of magnitude fluxes difference between the two rains. The
orders of magnitude found in this study could be used as "typical atmospheric fluxes" to estimate
atmospheric inputs of TMs by a rain event to the western Med Sea. We must keep in mind, however,
that annual and long-term deposition fluxes of dust-related elements (e.g. Ternon et al., 2010), but





also nitrogen (e.g., Richon et al., 2018b), in the Med Sea, are dominated by a few atypical intense
deposition events when they occurred.
*4.2.2. Comparison between TMs wet deposition inputs and marine stocks at ION and FAST stations*
Marine sampling sequences carried out before and after rains were, to our knowledge, the first direct
observations to trace the fate of atmospheric metals and nutrients in the water column after wet
deposition event. The time chart of the sampling of rain and column water (surface microlayer,
subsurface seawater and mixed layer) is presented in Fig. 8. The impact of the two wet depositions
on nutrients stocks in the Mediterranean surface waters is discussed in van Wanbeke et al. (2020)
and Pulido-Villena et al. (2021).  Both nitrate and DIP increased in the ML following the rains.
Although the closure of the N and P budgets had to take into account post-deposition processes as
new nutrient transfer through the microbial food web (uptake, remineralisation, and
adsorption/desorption processes on sinking particles), it was shown that wet deposition was a
significant source of nutrients for ML during the cruise. We focus here on the role of TMs deposition
as a source of metals to the column water. To do so, we estimated the potential enrichment of SML
and ML from the rain by calculating the difference (delta) in TMs stocks before and after rains.

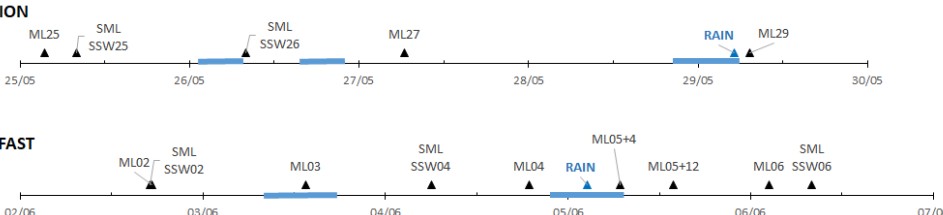


**Figure 8: Sampling chronology during the ION and FAST stations for SML, SSW and ML. The blue**
**periods correspond to rainfall in the station area (after ERA 5 reanalysis and radar imagery, see**
**section 3.1). Samplings were performed 4 days and 2 days before and 2 h after Rain ION, and at a**
**higher frequency at the FAST station: 57, 37 and 7.5 hours before and 4.5, 12, 24 hours after Rain**
**FAST. SML and SSW samples could not be collected immediately before and after the rains because**
**of bad weather conditions, and were collected 3 and 4 days before Rain ION, and 57 and 20 h before**
**and 30 h after Rain FAST.**
At ION, no SML sampling was done after rain, preventing the study of the rain effect. For ML, the
large variability in total and dissolved stocks between the two casts ML25 and ML27 before the
rain makes the establishment of a background concentration levels before rain difficult. ML27 was
used as initial conditions since it is the closest sampling from post-rain measurements (ML29). As





mentioned previously, dust rain deposition over the FAST station area started on 3 June. Bressac et
al. (2021) showed that the dust signature, traced by changes in Al and Fe stocks in the ML, was
already visible from the ML03 sampling. We defined the enrichment of seawater layers as the
difference between the maximum stocks after rains (from SML 04 or 06 and ML05+4) and the
initial seawater stocks (SML02 and ML02).
At ION, only particulate Cu (+27%) and Zn (+44%) stocks increased in the ML after the rain. Even
if the dissolved forms of Cu and Zn predominated in the ML, this increase was accompanied by
increasing Kd values, i.e. in the particulate/dissolved partitioning (0.07 vs 0.12 for Cu and 0.14 vs
0.2 for Zn). This was also the case for Fe (Kd increased from 2.6 to 4.3), although no significant
difference (<5%) was observed on the particulate Fe stock. The Kd values in the ION rain sample
being higher than in the marine stock before the rain, that suggests the wet deposition at ION is
mainly an additional source of particulate TMs.
At FAST, the ML stocks increased in the particulate phase for Fe (+61%), Mn (+15%) and Zn
(+9%) and in the dissolved phase for Cu (+9%), Fe (+46%), Pb (+8%) and Zn (+15%) (Fig. 9). In
addition of marine inventories, the particulate TMs input by rain was also observed on Kd values
and total X/Al in the ML. For example, Kd(Fe) increased from 0.14 to 0.17 in ML and its Kd was
0.25 in the rain. Even for Ni for which no change in stock could be evidenced, Kd(Ni) decreased
from 0.1 to 0.07 and its Kd in the rain was 0.006. For Mn/Al which fell from 0.27 before the rain to
0.008 after the rain (ML05+4), in accordance with the rain ratio (0.004). In the SML, the dissolved
and particulate stocks increased following rains for all TMs, from a factor 1.5 (Mo) to 10 (Fe) for
the dissolved phase and from a factor 1.6 (Ni) to 67 (Fe) for the particulate phase (Fig. 9).

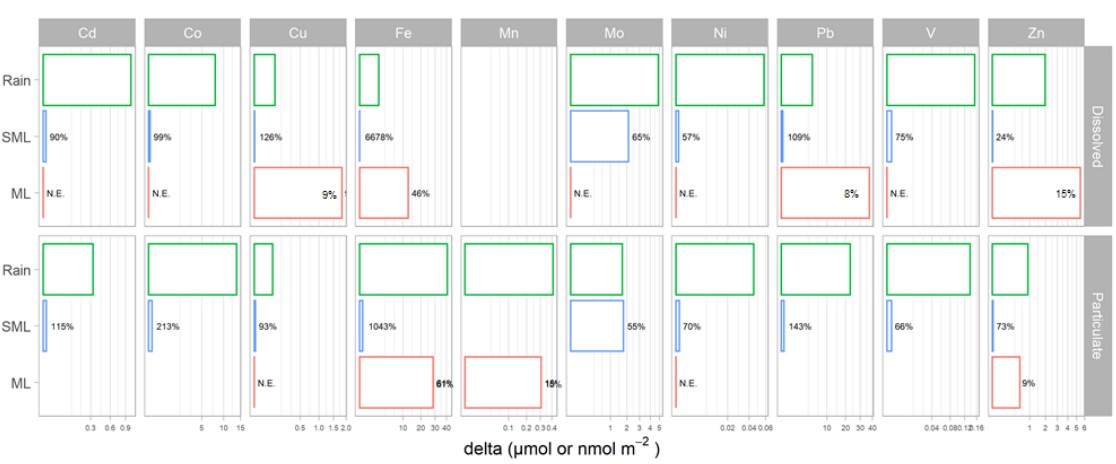



**Figure 9: Comparison between TMs wet deposition fluxes (in green) and TMs marine stock delta (before and after the rain) in the SML (in blue) and in the ML (in red) at FAST. Dissolved = upper panels and particulate = lower panels. Marine stocks increase are expressed in absolute values (Cd, Co and Pb stocks in nmol m$^{-2}$, and the other TMs in µmol m$^{-2}$) and in relative values (%). N.E.: not enhanced (increase <5%).**

The comparison between the observed enhancements in the SML stocks and the rain inputs at FAST (Fig. 9) indicates that the atmospheric fluxes support all observed deltas. Indeed, the atmospheric particulate and dissolved fluxes of TMs were 2 to 4 orders of magnitude higher than the mean stocks present in the SML, except Mo which was in the same order of magnitude. In the ML, the magnitude of atmospheric particulate inputs was higher or similar to the particulate marine delta of Fe, Mn and Zn. For Cu, Fe, Pb and Zn, the increase in dissolved stocks within the ML was 2 to 10 times higher than what could provide the atmospheric inputs. As described in Guieu et al. (2020), marine dynamical conditions at FAST were favorable to observe any change in the water masses strictly attributed to external inputs coming from the atmosphere on a short time scale. However, a cumulative effect of previous and surrounding wet deposition events could explain this difference between atmospheric inputs from rain FAST and increase of marine dissolved stocks. We cannot exclude the possibility of lateral transport of metals from surrounding waters being enriched by the rain events of June 3 for example, as revealed by the increase in the 0-20 m Fe and Al inventories (Bressac et al., 2021). Another hypothesis to explain that higher stock increases of metals in the ML than the one derived from atmospheric deposition is related to post-deposition processes. Indeed, once deposited, the atmospheric particulate fraction could still be partly solubilized in seawater, as the solubilisation of TMs (e.g. Fe) could occur over several hours or days (Wagener et al., 2008; Wuttig et al., 2013; Desboeufs et al., 2014). This could lead to an underestimate of dissolved TMs atmospheric inputs. Moreover, the time lag between the rain and the first SML sampling (1 day) does not allow us to conclude on the role played by SML as a "trap" of the added dissolved metals by rain. However, results showed clearly the increase of dissolved TMs in the SML even 24 h after the rain (Fig. 9). Even if this increase could be due to dissolution processes (Tovar-Sanchez et al., 2020), we cannot exclude that the residence time of dissolved atmospheric TMs in the SML was sufficient to mask the atmospheric inputs in the ML05+4 sample. It is also known that dissolved concentrations in the ML are subject to various biological processes such as phytoplankton uptake (Morel et al., 2003). The comparison between ML05+4 and ML05+12 samples performed after the rain shows that the dissolved and particulate stocks decreased quickly for all the TMs (not shown), in agreement with the predominance of removal processes (sedimentation, biological transfer, adsorption) on these stocks. However, the rate of decrease depended on the TMs, showing that some





removal processes predominate over others depending on the metal. For example, the dissolved
metals decrease could correspond to scavenging onto particles, which is a common physical-
chemical process occurring in the ocean for Fe (Wagener et al., 2010; Bressac and Guieu., 2013) or
Co (Migon et al., 2020). Finally, our results show that dust wet deposition was a net source of all
the studied trace metals for SML both in the dissolved and particulate fraction. For ML, atmospheric
dust inputs were also a net source of particulate Fe, Mn, and Zn, and dissolved Cu, Fe, Pb and Zn.
Due to various marine post-deposition processes, it was more complicated to observe the effect of
wet deposition on dissolved stocks, explaining why the SML and ML particulate stocks were more
impacted by rain than the dissolved stocks. On a timescale of hours, Fe inventory was the most
impacted by the dusty rain input, both in dissolved and particulate phase, confirming that the dust-
rich rains are a net source of Fe to the surface Mediterranean Sea (Bonnet and Guieu, 2006, Bressac
et al., 2021).
*4.2.3. Comparison between TMs wet atmospheric inputs and marine stocks at the scale Western*
*and Central Mediterranean Sea*
As observed from dissolved TMs stocks measured before and after the rains, a large part of
uncertainties in the data analysis results from various removal processes of TMs after wet
deposition, which could have time resolution shorter than the sampling step. In order to limit the
effect on these potential processes in data analysis, here we further study the role of wet deposition
by comparing atmospheric dissolved fluxes to marine dissolved stocks by using TMs profiles in the
ML at all 13 marine stations, i.e. 22 ML samplings, throughout the whole cruise (Fig. 10). Indeed,
considering that the collected rains were originating from large rain systems covering more than
50 000 km$^2$ around the sampling zone and were typical of Mediterranean wet deposition, we
hypothesized that they could have occurred in any of the explored areas during the cruise.
Exceptional intense dust deposition events have been recorded in the Mediterranean, reaching 20 g
m$^{-2}$ (Bonnet et al., 2006). Sporadic and intense wet dust deposition higher than 1 g m$^{-2}$ are regularly
observed in the spring in the western Mediterranean basin (e.g., Vincent et al., 2016). At the
beginning of the cruise, an intense wet dust deposition event (not collected) occurred over the South
of Sardinia and over the Tyrrhenian Sea with fluxes reaching about 9 g m$^{-2}$ (Bressac et al., 2021).
In order to take into account the effect of such an event, we also estimated the atmospheric fluxes
of dissolved metals based on a 9 g m$^{-2}$ wet dust deposition event considering solubility values found
in the rain FAST (Fig. 10). The metal solubility decreasing with increasing dust load (Theodosi et
al., 2010), this estimation constitutes probably a maximum value of the dissolved inputs of trace
metals by such a dust deposition event. In addition to removal processes, the impact of rain inputs



on TMs marine stocks is also controlled by MLD fluctuations that we ignored in the work described
above by using a fixed depth for FAST an ION stations. However, the variability of this MLD (7-
21 m during the cruise, typical of Med thermal stratified period) could change the marine budgets
by a factor of 3. So we considered the measured MLD (Van Wambeke et al., 2020) for calculating
marine budgets of TMs at each station.

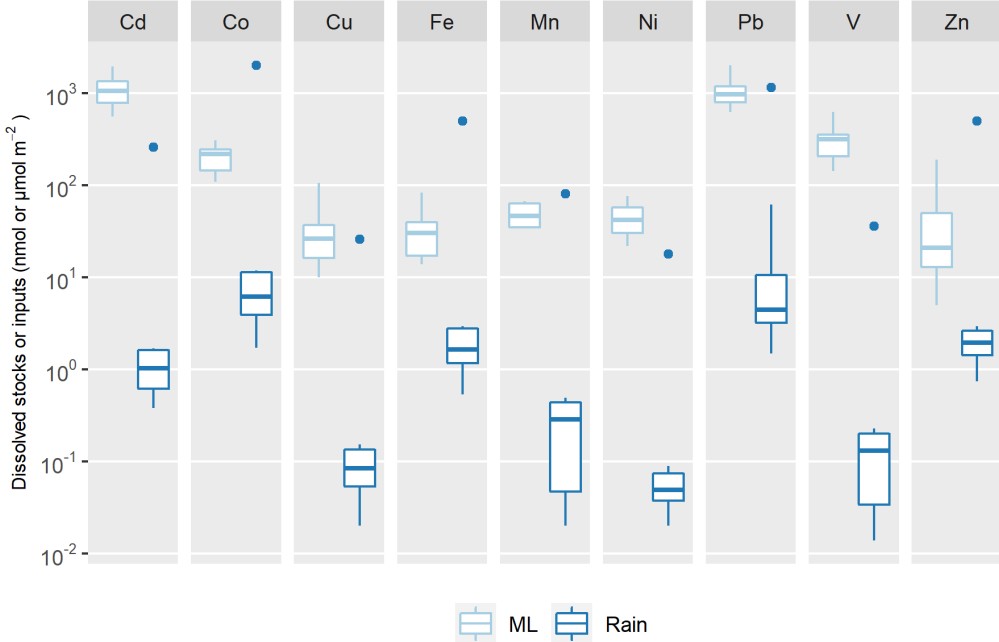


**Figure 10: Comparison of marine stocks in the ML at all the stations occupied during the PEACETIME cruise with atmospheric inputs estimated (1) from ION and FAST rains (Boxes) and (2) from an intense wet dust deposition event of 9 g m$^{-2}$ (blue dots). Cd, Co and Pb stocks are in nmol m$^{-2}$, and the other TMs in µmol m$^{-2}$. For Mn, marine stocks are derived from surface concentrations close to Corsica coasts (Wuttig et al., 2013: samples OUT at 0, 5 and 10 m) and in the Ionian Sea (Saager et al., 1993: Bannock basin at 0, 10, and 15 m), as no measurement is available from the PEACETIME cruise. Boxes and whiskers as in Fig. 4.**

Applying to the whole transect, the atmospheric inputs, obtained from our rain composition, were
at least 100-fold smaller than the dissolved stocks in the mixed layer, except for Co, Fe and Zn. The
atmospheric inputs represented more than 30% of the dissolved Zn stocks and 10 to 18% for Fe.
This significant input of dissolved Fe is in agreement with our field observations in the ML. For
Co, the maximum atmospheric fluxes estimated during the cruise represented >10% of stocks. Here





the comparison is based only on dissolved TMs in rain water, yet as discussed previously, the
solubilisation post-deposition of atmospheric particles in the water column could further enrich the
marine dissolved stocks. In the case of the intense dust deposition event, the dissolved inputs are of
the same order of magnitude as marine stocks for Co, Fe, Mn, Pb and Zn. The enrichment in
dissolved Fe and Mn was previously observed by Wuttig et al. (2013) after artificial dust seeding
in large mesocosms (simulating a wet deposition event of 10 g m$^{-2}$). The surface seawater could be
significantly affected by the deposition of these dissolved elements in the case of wet dust
deposition. The marine TMs concentrations measured during the cruise being typical of
Mediterranean surface seawater concentrations, we conclude from these comparisons that wet
deposition events, notably wet dust deposition events, prove to be an external source of dissolved
TMs for the Mediterranean Sea during the period of thermal stratification.
## 5.     Conclusions
This study provides both the dynamical properties and chemical characterization of two rainwaters
collected in the open Mediterranean Sea, concurrently with TMs marine stocks in surface seawater.
Our results are the only recent report of concentrations, EF and solubility values for TMs in rain
samples collected in remote Med Sea. By highlighting the discrepancy between TMs concentrations
with the previous and coastal rain studies, this work demonstrates the need to provide a new and
recent database on metal composition in Mediterranean rains in order to estimate the role of
atmospheric TMs deposition. We showed the representativeness of rain FAST as typical Saharan
dust wet deposition as well in its chemical composition as in its magnitude and extent, whereas Rain
ION is a typical low perturbed anthropogenic background rain of remote Med Sea. On this basis,
we suggest to use the chemical composition of PEACETIME rains as a new reference for the studies
of TMs on wet deposition in Med Sea.
Since atmospheric TMs have been identified as critical oligo-nutrients for marine biosphere, our
study is the first *in situ* evidence that atmospheric wet deposition constitutes a significant external
source for some of these elements to surface stratified Mediterranean seawater. Our results show
that the original approach developed here is very relevant in this purpose and could be used in other
part of the world where atmospheric deposition is suspected to impact marine biosphere, as HNLC
areas.






**Data availability.** Guieu et al., Biogeochemical dataset collected during the PEACETIME cruise. SEANOE. https://doi.org/10.17882/75747 (2020). Atmospheric Data are accessible on http://www.obs-vlfr.fr/proof/php/PEACETIME/peacetime.php.

**Author contributions.** KD and FF designed the study and wrote the manuscript; FF, ST, JFD, Ch.G made the on-board atmospheric measurements and sampling during the cruise; FF, ST and JD analysed the rain samples; MB, ATS, and ARR made the marine TMs sampling and analyses; PF was the reference scientist of PEGASUS, AF and FM managed all the technical preparation of atmospheric samplings, PC analysed the lidar data; KD, FD and Ce.G designed the cruise strategy; KD and Ce.G coordinated the PEACETIME project, FD coordinated the ChArMEx funding request, and near-real time and forecast survey of atmospheric conditions during the cruise; all the authors commented on the manuscript and contributed to its improvement.

**Competing interests.** The authors declare that they have no conflict of interest.

**Special issue statement.** This article is part of the special issue "Atmospheric deposition in the low-nutrient–low-chlorophyll (LNLC) ocean: effects on marine life today and in the future (ACP/BG inter- journal SI)". It is not associated with a conference.

**Acknowledgements.** The authors wish to thank Thierry Alix the captain of the R/V *Pourquoi Pas ?* as well as the whole crew and technical staff for their involvement in the scientific operation. We gratefully thank Thibaut Wagener for his involvement in the trace-metals clean marine sampling and Mickaël Tharaud for the HR-ICP-MS analysis. We thank the Leosphere Technical support team and especially Alexandre Menard for their remote assistance with LIDAR repair under difficult off-shore conditions. Hélène Ferré and the AERIS/SEDOO service are acknowledged for real-time collection during the cruise of maps from operational satellites and forecast models used in this study, with appreciated contributions of EUMETSAT and AERIS/ICARE for MSG/SEVIRI products. EUMETNET is acknowledged for providing the pan-European weather radar composite images through its OPERA programme. We acknowledge the US National Oceanic and Atmospheric Administration (NOAA) Air Resources Laboratory (ARL) for the provision of the HYSPLIT (HYbrid Single-Particle Lagrangian Integrated Trajectory) model via NOAA ARL READY website ( http://ready.arl.noaa.gov) used in this publication. This study is a contribution to the PEACETIME project (http://peacetime-project.org; last accessed 05/04/2021), a joint initiative of the MERMEX and ChArMEx programmes supported by CNRS-INSU, IFREMER, CEA and Météo-France as part of the decadal meta-programme MISTRALS coordinated by CNRS-INSU. PEACETIME was endorsed as a process study by GEOTRACES and is also a contribution to IMBER and SOLAS international programs.

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



**Figure Captions:**

Figure 1: Total precipitation (mm) between May 28 at 20:00 UTC and 29 at 10:00 UTC from ERA5 ECMWF reanalysis. The red circle indicates the R/V position.

Figure 2: On-board lidar-derived a) Apparent backscatter coefficient (ABC), (b) Temporal evolution (in Local Time) of the lidar-derived volume depolarization ratio (VDR) where the dust plume is highlighted for values higher than ~1.7 (yellow to red colors) and the rain by values higher than 3 (indicated by the arrow), and c) vertical profiles of the aerosol extinction coefficient (AEC) in cloud free condition, integrated over 3 periods along the dust plume event, noted A, B and C in figure a. The grey shade represents the root mean square (rms) variability along the time of the measurement. The dust layer is highlighted on the profiles. The mean aerosol optical thickness is given in the boxed legend with its temporal variability (1 sigma). The location of the marine boundary layer (MBL) is also pointed.

Figure 3: Rain rates (mm/h) during the night between the 4 and 5 June, when Rain Fast was collected on-board, issued from European rain radar composites (OPERA programme) of June 5 between 00:00 and 02:45 UTC.

Figure 4: Boxplots of dissolved (left panels) and particulate (right panels) marine concentrations (pM) for the different TMs within the ML (upper panels) and the SML (lower panels) at ION and FAST stations. In the box plots, the box indicates the interquartile range, i.e. the 25th and the 75th percentile, and the line within the box marks the median. The whiskers indicate the quartiles ±1.5 times the interquartile range. Points above and below the whiskers indicate outliers outside the 10th and 90th percentile.

Figure 5: Comparison of dissolved (D) and total (T) TMs concentrations along with data from 14 former studies carried out in the eastern and western Mediterranean Sea.

Figure 6: Enrichment Factors (EF, upper panel) and solubility (%, bottom panel) of phosphorus (P) and TMs ordered by increasing EF in the two rainwater samples.

Figure 7: Dissolved (upper panels) and particulate (lower panels) wet deposition fluxes ($\mu$mol m$^{-2}$) for the different TMs estimated from the two rains sampled on-board, considering the standard deviation on the TMs concentrations and the spatial variability of total precipitation over the area of sampling (Rain ION in blue and Rain FAST in red).

Figure 8: Sampling chronology during the ION and FAST stations for SML, SSW and ML. The blue periods correspond to rainfall in the station area (after ERA 5 reanalysis and radar imagery, see section 3.1). Samplings were performed 4 days and 2 days before and 2 h after Rain ION, and at a higher frequency at the FAST station: 57, 37 and 7.5 hours before and 4.5, 12, 24 hours after Rain FAST. SML and SSW samples could not be collected immediately before and after the rains because of bad weather conditions, and were collected 3 and 4 days before Rain ION, and 57 and 20 h before and 30 h after Rain FAST.



Figure 9: Comparison between TMs wet deposition fluxes (in green) and TMs marine stock delta (before
and after the rain) in the SML (in blue) and in the ML (in red) at FAST. Dissolved = upper panels and
particulate = lower panels. Marine stocks increase are expressed in absolute values (Cd, Co and Pb stocks in
nmol m$^{-2}$, and the other TMs in µmol m$^{-2}$) and in relative values (%). N.E.: not enhanced (increase <5%).
Figure 10: Comparison of marine stocks in the ML at all the stations occupied during the PEACETIME
cruise with atmospheric inputs estimated (1) from ION and FAST rains (Boxes) and (2) from an intense wet
dust deposition event of 9 g m$^{-2}$ (blue dots). Cd, Co and Pb stocks are in nmol m$^{-2}$, and the other TMs in
µmol m$^{-2}$. For Mn, marine stocks are derived from surface concentrations close to Corsica coasts (Wuttig et
al., 2013: samples OUT at 0, 5 and 10 m) and in the Ionian Sea (Saager et al., 1993: Bannock basin at 0, 10,
and 15 m), as no measurement is available from the PEACETIME cruise. Boxes and whiskers as in Fig. 4.