# Peer review of "Wet deposition in the remote western and central Mediterranean"

_Atmospheric Chemistry and Physics, 2021_

## Referee Comment (RC2)

c Author(s) 2021. CC BY 4.0 License.

[revised manuscript text omitted]

Sea, possibly limiting (or co-limiting)  phytoplankton growth (Pinedo-Gonzàles et al. 2015), pointing to the importante of dust deposition as source of TMs for planktonic communities. On the other hand, atmospheric deposition of European aerosol  was identified to have a negative effect on chlorophyll concentrations (Gallisai et al. 2014) by providing

Cu at toxic levels (Jordi et al. 2012).

The atmospheric deposition of TMs in the Mediterranean is related to both dust and anthropogenic aerosol deposition (Desboeufs et al., 2018). The role of dust deposition as a source of TMs was observed from the correlation between the atmospheric deposition of mineral dust, and the enrichment of dissolved TMs (Cd, Co, Cu, Fe) in the Mediterranean Sea surface microlayer (Tovar Sánchez et al.,

2014). For the water column, the adding of dissolved Fe and Mn was emphasized in mesocosm experiments after dust addition mimicking intense wet dust deposition (Wuttig et al., 2013). Yet, the direct impact of wet deposition  on TM concentrations in surface seawater has  not been studied and reported *in situ* where TM concentrations were determined in both rainwater and seawater samples collected from the same location before the present study, whether in the Mediterranean Sea

**Commented [RS(-S5]:** Define/give examples

**Commented [RS(-S6]:** Replace with microbial. Biosphere is too general

**Commented [RS(-S7]:** Which nutrients?

[Figure]

or in other oceanic regions. Moreover, the two key criteria used to assess the potential impact of TMs and nutrients wet deposition are and their respective concentrations (or fluxes) and solubility, i.e., the partitioning between dissolved and total concentrations in rainwaters. Indeed, it is considered that the dissolved fraction of nutrients and TMs can be directly assimilated by the phytoplankton (ref). A fFew studies have focussed onreported concentrations of

TMs in rainwater samples was collected around the Mediterranean basin: Al-Momani et al., (1998),

Kanellopoulou, (2001) and Özsoy and Örnektekin, (2009) in the eastern basin, and Losno (1989),

Guieu et al. (1997), Frau et al., (1996), Chester et al., (1997) and Guerzoni et al. (1999b) in the western basin. These studies led to highly variable TMs concentrations and solubility, illustrating a more generalthe large variability of TMs inputs during wet deposition eventss in the Mediterranean Sea (reviewed in Desboeufs, 2021). All these studies were performed at coastal sites. Offshore samples of rainwater have rarely been studied reported in the literature so far. In the Mediterranean, to our knowledge, trace element concentrations from only three rain samples collected at sea in April 1981

have been reported in a PhD thesis (Dulac, 1986). However, due to the continental and local sources of pollution and the variety of anthropogenic aerosol sources (Amato et al., 2016), the TMs rain composition of the coastal zone could may not be representative of atmospheric deposition in to the remote Mediterranean.

The PEACETIME cruise (ProcEss studies at the Air-sEa Interface after dust deposition in the

MEditerranean Sea) performed in spring 2017 aimed at to studying the impacts of atmospheric deposition, in particular Saharan dust events, on the physical, chemical and biological processes in this marine oligotrophic environment (Guieu et al., 2020). We investigated here the concentration and solubility of TMs and nutrients of from two rain eventswaters sampled in the open Mediterranean open Ssea during the cruise. We compare our results on of TMs concentrations and solubility in rainwater with previous studies based on rainwater samples collected at coastal sites to investigate potential differences with the open sea. Additionally, to assess the impact of wet deposition on the surface TMs concentrations, surface seawater, including the surface micro-layer, and rain water was concurrently collected. To the best of our knowledge, this is the first time TM data for these marine compartments have all been presented at the same time. for the first time with rain samples.

**2. Sampling and methods**

**2.1 Sampling and chemical analysis of rainwater**

The PEACETIME oceanographic campaign (https://doi.org/10.17600/17000300) took place in the western and central Mediterranean Sea on-board the French research vessel (R/V) *Pourquoi Pas ?*

between 11 May and 10 June 2017, i.e. at the beginning of the Mediterranean stratification season (Guieu et al., 2020). The rain collector was installed on the upper deck (22 m above sea level) where

**Commented [RS(-S8]:** 'To the best of our knowledge….

**Commented [RS(-S9]:** Coastal atmospheric TM concentrations are not usually representative of open ocean concentrations due to deposition of material en route

**Commented [RS(-S10]:** It's definitely not the first time this has been done but might be the first time the data has been reported in the same place c Author(s) 2021. CC BY 4.0 License.

[Figure]

no on-board activities were taking place to avoid contamination. A The rain collector was equipped with an on-line filtration system to directly separate the dissolved and particulate fractions at the time of collection (details of the filtration system are available in Heimburger et al., 2012) allowing for the calculation of solubility of TMs in the rainwater at the time of collection. The filtration device was equipped with a Nuclepore® polycarbonate membrane (PC) filter (porosity: 0.2 μm, diameter: 47

mm). T, and the diameter of the funnel of the collector was 24 cm. The rain collectors were was installed opened only when rain was expected and kept closed withincovered by an acid-washed, sealed plastic film until the rainfall beganwhen not in use. All the sampling materials were thoroughly acid-washed in the laboratory prior to the cruise departure (washing protocol described in Heimburger et al., 2012). No stabilizing setup device was used to keep the funnel level during the pitch and roll of the ship, preventing a precise assessment of the height of rainfall from the collected water volume.

During the rain sampling, the ship was always facing the wind to avoid contamination by the smoke ship's exhaust of the ship itself, as the chimney was situated on the lower deck and behind the collector.

Immediately after sampling, the collector was disassembled under the a laminar flow hood inside an on-board clean-room on board laboratory. The dissolved fraction was separated into 4 four aliquots dedicated to i) dissolved organic carbon (DOC) determimation by high-temperature catalytic oxidation (HTCO) on a Shimadzu total organic carbon analyzer (as described in Van Wambeke et al

2021), ii) major ions by ion chromatography (IC), iii) metals analysisTM determination by inductively coupled plasma coupled methods (ICP), and iv) pH measurement. For ICP measurements, the sample was acidified immediately to 1% by volume of ultra-pure nitric acid: 67-69%, Ultrapur, Normatom®,

VWR. For DOC analysis… For IC analysis, the filtrate sample (dissolved fraction) was immediately frozen. The filter (particulate fraction) was dried under the laminar flow hood, and then put in a storage box and packed with a plastic bag to avoid contamination. After returning to the laboratory, filters were acid digested by using the adapted protocol adapted from Heimburger et al.

(2012) as follows: filters were placed in tightly capped Savillex™ PFA digestion vessels with 4 mL

of a mixture of $HNO_3$ (67-69%, Ultrapur, Normatom®, VWR), $H_2O$ and HF acids (40%, Ultrapur,

Normatom®, VWR) in a proportion of (3: 1: 0.5), then heated in an oven at 130°C for 14 hours. After cooling, the acid solution was completely evaporated on a heater plate (ANALAB, 250, A4) at 140°C

for about

2h, then 0.5 mL of $H_2O_2$ (30-32%, Romil-UpA™) and 1 mL of the acidified water (2% $HNO_3$) was added to the vessels and heated during for 30 min. to dissolve the dry residue in the bottom of the vessels; finally, 12 mL of acidified water (1% $HNO_3$) was added to obtain 13.5 mL of solution in a tube for ICP-MS analyses.

The dissolved fraction was analysed by IC (IC 850 Metrohm) for the inorganic and organic anions

Commented [RS(-S11]: At the point when it would have been deposited to the surface ocean.

Commented [RS(-S12]: No need to define as you don't use this abbreviation elsewhere

Commented [RS(-S13]: The

Commented [RS(-S14]: This should be number 1 as it is the focus of the paper. List = i) TMs, ii) DOC, iii) major seawater and atmospheric ions, iv) pH. Then a description of the methods used for each, in the order they are listed

($NO_2^-$, $NO_3^-$, $PO_4^{3-}$, $SO_4^{2-}$, $F^-$, $Cl^-$, $Br^-$, $HCOO^-$, $CH_3COO^-$, $C_2H_5COO^-$, MSA, $C_2O_4^{2-}$) and for the cation

$NH_4^+$ (Mallet et al., 2017).  the dissolved fraction and solution from the digestion of the particulate fraction of the rain samples  were analysed by ICP-AES (Inductively Coupled

Plasma Atomic Emission Spectrometry, Spectro ARCOS Ametek®) for major elements (Al, Ca, K,

Mg, Na, S) (Desboeufs et al., 2014) and by HR-ICP-MS (High Resolution Inductively Coupled Plasma

Mass

Spectrometry, Neptune Plus ™ at Thermo Scientific ™) for TMs: Cd, Co, Cr, Cu, Fe, Mn, Mo, Ni, P,

Pb, Ti, V and Zn. The speciation of dissolved P was estimated by determining  dissolved inorganic phosphorus (DIP) from phosphate concentrations expressed as P and the dissolved organic phosphorus (DOP) from the difference between total dissolved phosphorus (TDP), obtained by ICP-MS, and DIP, obtained by IC.

In order to estimate the contamination of sampling and analytical protocols,  three blanks of rain samples (collected on-board during the cruise with the same protocol without rain events) were used and processed. The procedural limit of detections (LoD) were defined as 3 x standard deviation of blank samples both for dissolved and particulate fractions estimated after acid digestion. All dissolved and particulate sample concentrations were higher than LoD, except for $NO_2^-$ in the two rain samples. The blank concentrations represented 10.2% in average for TMs and were typically lower than 20% of the sample concentrations, except for Cd (52%) and Mo (43%) in the dissolved fraction.

Blank concentrations were subtracted from all sample concentrations.

**2.2 Atmospheric ancillary measurements**

The PEGASUS (PortablE Gas and Aerosol Sampling UnitS, www.pegasus.cnrs.fr) mobile platform of LISA is a self-contained facility based on two standard 20-feet containers, adapted with air conditioning, rectified power, air intake and exhausts for sampling and online measurements of atmospheric aerosols and gaseous compounds, and their analysis (Formenti et al., 2019). During the

PEACETIME cruise, only the sampling module of the facility was deployed on the starboard side of deck 7 of the R/V. The PEGASUS instrumental payload of relevance to this paper included measurements of the major gases such as NOx, $SO_2$, $O_3$ and CO by online analysers (Horiba APNA,

APSA, APOA and PICARRO respectively; 2-min resolution, detection limit for all analytes was 0.5

ppb and 1 ppb for CO). These gases,  were used to estimate the origins of the sampled air.

From 1st June 2017 (?) (not operational before), additional measurements by an ALS450®

Rayleigh-

**Commented [RS(-S15]:** You don't present this data here – note this

**Commented [RS(-S16]:** Move up to where the rest of the ICP stuff is

**Commented [RS(-S17]:** More detail needed. Not clear what the rain blanks were. What were the blanks for the dissolved fraction? Empty bottles swished out with MQ or acidified MQ and a filter? Blanks and LoDs can be found where?

[revised manuscript text omitted]

The relative solubility of TMs in the two rainwater events was calculated as:

> **Commented [RS(-S19]:** Which metals? Mo and Pb or not all samples were preconcentrated for all metals listed?

> **Commented [RS(-S20]:** Not exactly but the vast majority of Al is crustal. Therefore, it is currently accepted as one of the best proxy elements. It is predominantly crustal c Author(s) 2021. CC BY 4.0 License.

[Figure]

$$S_X\% = \frac{[X]disssolved}{[X]total} \text{ x } 100 \qquad (2)$$

where $S_X\%$ is the relative solubility (in %) of an element X in the rainwater, $[X]_{dissolved}$ and $[X]_{total}$ are its soluble and total concentration, respectively.

**2.5 Atmospheric deposition fluxes**

Impacts on biogeochemical cycles and ecosystem functioning after a rain event occur on time scales of a few days (2-3), and space scales of tens of km (about 20-50 km within the radius of the ship's position). In the specific context of oceanographic cruising, the documentation of these impacts is restricted to the vertical dimension at the prescribed temporal scale. In this vertical dimension, the exchange of TMs  into the ML was controlled both by atmospheric inputs over the R/V position and by advection from surrounding water masses that may have been impacted by surrounding rainfall. Therefore, we had to consider this process in our estimation of the atmospheric flux contributions. For this purpose, the atmospheric fluxes have to be integrated to the extent of the rain area that can impact the marine surface layer. We derived wet deposition fluxes by considering the total precipitation accumulated during the duration of the rain over the area around the R/V location.

Thus, the wet deposition fluxes in our rain samples were calculated by multiplying the volume weighted mean rainfall concentration by the total precipitation (equation and explanation for using

VWM rainfall here). The total precipitation of the rain events was issued from the hourly total precipitation accumulated during the rain events over the region from ERA5 ECMWF reanalysis (Herbasch et al., 2018) and from the rain rate composite radar products from the European OPERA

database (Saltikoff et al., 2019), when it was possible. Although subject to uncertainties (Morin et al.,

2003), a surface-based weather radar is probably the best tool to estimate rainfall in the surroundings of the R/V because…. However, the OPERA database does not include Italian radars, which did not cover the central area of the Ionian Sea during the cruise anyway. ERA5 data are available on regular latitude-longitude grids at 0.25º x 0.25º resolution. The accumulated precipitation was taken from the grid-points spanning the ship's location, approximately 0.25° around the central grid-point for integrating the regional variability. Surface rain rate radar composite images were available every 15 minutes with a spatial resolution of 2 km x 2 km. The accumulated precipitation was the sum of integrarated rain rates during the rain duration averaged over the radar pixels spanning the ship's location within a radius of about 25 km around the ship location.

**2.6 Stocks in the surface seawater**

For the surface microlayer (SML), stocks of TMs were estimated from the integration of TM

concentration over the thickness of the SML. The thickness ranged from 32 to 43 µm and from

26 to 43 µm at ION and FAST, respectively (Tovar-Sanchez et al., 2020).

**Commented [RS(-S21]:** Sometimes ship, sometimes R/V, perhaps best to stick with one c Author(s) 2021. CC BY 4.0 License.

[revised manuscript text omitted]

**Commented [RS(-S26]:** Rate requires a time dimension

**Commented [RS(-S27]:** Onboard is one word – several cases need correcting c Author(s) 2021. CC BY 4.0 License.

[Figure]

[Figure]

[Figure]

Commented [RS(-S28]: It would be nice to have S3 in this figure for comparison instead of in the SI

**Figure 1: Total precipitation (mm) between 28 May 201728 at 20:00 UTC and 29 May 2017 at 10:00 UTC from ERA5 ECMWF reanalysis. The red circle indicates the R/V position.**

*3.1.2. Rain FAST*

As detailed in Guieu et al. (2020), the FAST station position was decided on the basis of regional model forecast runs and satellite observations, in for the purpose to of catching a wet dust deposition event. Significant dust emissions were observed from NASCube (http://nascube.univ-lille1.fr/,

Gonzales and Briottet, 2017) over North Africa from the night of 30-31 May, then new dust emissions in the night from 3 to 4 June in Algeria and southern Morocco associated with a northward atmospheric flux.   On   30   May,   the   SEVIRI   AOD   satellite   product   (https://www.icare.univ- lille.fr/dataaccess/browse-images/geostationary-satellites/, Thieuleux et al., 2005) confirmed the presence of atmospheric dust in a cloudy air mass over the western part of the Mediterranean, and from 2 June  the export of a dust plume from North Africa south of the Balearic Islands with high

AOD (>0.8) on the Alboran Sea was observed (Fig. S2). The dust plume was transported to the NE

up to Sardinia on June 4, with AOD <0.5 in all the area and c. Clear sky with low AOD was left observed west of 4°E on June

5.

On-board lidar measurements (Fig. 2 a,b,c) showed that the aerosol plume was present over the ship's position from 2 June at 21:00 UTC until the rain event, and corresponded to a dust aerosol layer well highlighted by the high depolarization. The dust plume was concentrated between 3 and 4 km at the beginning of the station occupation, then expanded down to the marine boundary layer (about 500 m c Author(s) 2021. CC BY 4.0 License.

amsl) by the end of the day on 3 June 201.

The mass integrated  concentration of dust aerosols derived from the profiles of aerosol extinction ranged from a minimum of 0.18 ±0.005 g m$^{-2}$ just before the rain to a maximum of 0.24

±0.009 g m$^{-2}$, where standard deviations indicate the temporal variability (1 sigma).

[Figure]

**Figure 2: On-board lidar-derived a) Apparent backscatter coefficient (ABC), (b) Temporal evolution**

**(in  local time) of the lidar-derived volume depolarization ratio (VDR) where the dust**

**plume is highlighted for values higher than ~1.7 (yellow to red colours) and the rain by values higher**

**than 3 (indicated by the arrow), and c) vertical profiles of the aerosol extinction coefficient (AEC) in**

**cloud free condition, integrated over 3 periods along the dust plume event, noted A, B and C in**

**Commented [RS(-S29]:** Can these panels be relabelled? You have panels a-c and labelling within panel a of arrows a-c, and in panel c of box a-c. All these a-cs get a bit confusing

[Figure]

**panelfigure a (top). The grey shade represents the root mean square (rms) variability along the time**

**of the measurement. The dust layer is highlighted on the profiles. The mean aerosol optical thickness**

**is given in the boxed legend with its temporal variability (1 sigma). The location of the marine**

**boundary layer (MBL) is also pointed.**

Rainfalls were was observed by weather radar images in the neighbouring area of the R/V

neighbouring from 3 June at 7:00 UTC. The rainfalls recorded around the FAST station were was associated with 2 two periods of rain: the 03/06 from 07:00 to 14:00 UTC on 03/06, and from the

04/06 at 16:00 UTC on 04/06 to 05/06 at 06:00 UTC on 05/06. For this latter case, a rain front (100

[revised manuscript text omitted]

> **Commented [RS(-S32]:** Are you really talking about deep water masses or just the water below the ML?

c Author(s) 2021. CC BY 4.0 License.

[Figure]

[Figure]

in agreement with the abundance of dissolved Mo in seawater (~ 107 nM, Smedley and Kinniburgh,

2017), and Fe in the particulate fraction (value). All the particulate and dissolved TMs concentrations measured during the cruise were  within the range previously published for the

Mediterranean Sea (Sherrell and Boyle, 1988; Saager et al., 1993; Morley et al., 1997; Yoon et al.,

1999; Wuttig et al., 2013; Baconnais et al., 2019; Migon et al., 2020). Zn presented the largest range of concentrations within the ML both in the particulate and dissolved phases (values), due to some high concentrations. However, the concentrations stayed in the typical range of values  reported for the Mediterranean Sea (Bethoux et al., 1990, Yoon et al., 1999). The concentrations  in the

SML were lower than in the ML and Pb dominated both in dissolved and particulate phases. Tovar-Sanchez et al. (2020) showed that the TMs concentrations in the SML during the PEACETIME campaign were generally lower than those previously measured in the

Mediterranean Sea, except  for the particulate phase  at the FAST station  following dust deposition.

> **Commented [RS(-S33]:** Did any of these studies determine pMo and/or dMo? If not, say that yours are the first

> **Commented [RS(-S34]:** Was contamination suspected?

> **Commented [RS(-S35]:** I don't understand this statement

[Figure]

**Figure 4:** **Box_plots of dissolved (left panels) and particulate (right panels) marine concentrations**
**(pM) for the different TMs within the ML (upper panels) and the SML (lower panels) at ION (green)**
**and FAST (red) stations. In the box plots, the box indicates the interquartile range, i.e. the 25th and**
**the 75 th percentile, and the line within the box marks the median. The whiskers indicate the quartiles**

> **Commented [RS(-S36]:** Could these be plotted on the same scale to make it easier to compare the SML and ML concs?

[Figure]

**±1.5 times the interquartile range. Points above and below the whiskers indicate outliers outside the**

**10-th and 90th percentile.**

**4. Discussion**

**4.1. Composition of rain collected over the remote Mediterranean Sea**

*4.1.1. Concentrations*

Regarding nutrients, nitrogen species concentrations in rain samples were in good agreement with those reported in Mediterranean rain samples, ranging from 1130 to 5100 µg L$^{-1}$ for NO$_3^-$ and between

207 and 1200 µg L$^{-1}$ for NH$_4^+$ (Loye-Pilot et al., 1990; Avila et al., 1997; Al Momani et al., 1998;

Herut et al., 1999; Violaki et al., 2010; Izquieta-Rojano et al., 2016; Nehir and Koçak, 2018). The

FAST rain concentrations were in the published range, whereas the ION rain was in the low range, confirming a background signature at this station. The rainwater samples presented a large dominance of N in comparison to P, as observed from the N/P ratio derived from DIN/DIP

(Table 2) ranging from 208 at ION to 480 at FAST. Previous observations showed a predominance of

N relative to P in  atmospheric bulk deposition over the Mediterranean coast, with ratios higher than the Redfield ratio (Markaki et al., 2010, Desboeufs et al., 2018). The highest ratio observed reached 1200 (in the case of DIN/TDP), but  averaged  about 100 in bulk deposition (unfiltered rain). The highest ratio could be linked to a washout effect of  gaseous N species (

NOx and NH$_3$) by rain (Ochoa-Hueso et al., 2011). At the two stations, no high NOx concentrations were observed in the boundary layer before wet deposition (range). The presence of nitrate and ammonium in the background aerosols has been  observed during recent campaigns in the remote Mediterranean atmosphere (e.g. Mallet et al., 2019). To our knowledge, no data are available on both P and N concentrations in Mediterranean aerosols. The lowest concentrations of P relative to

N in aerosol particles in the Mediterranean were observed during the cruise (value, Fu et al., in prep.). The TDP concentrations were consistent with the average value of 8.4 µg L$^{-1}$ measured in African dust rain samples collected in Spain over the 1996-2008 period (Izquierdo et al., 2012).

Inorganic phosphorus predominated in the dust-rich rain, whereas organic P was dominant in the background rain as the contribution of DOP to the TDP was 60% and 44% in Rain ION and Rain

FAST, respectively. The DOP/TDP ratio presents a very large range in Mediterranean rains, spanning from 6% in Spanish dusty rain samples (Izquierdo et al., 2012) to 75-92% in rains from Crete (Violaki et al., 2018). A reason for this wide range could be that Mediterranean European aerosols, as opposed to Saharan dust particles, are dominated by organic phosphorus compounds associated with bacteria (Longo et al., 2014).

**Commented [RS(-S37]:** Wet +dry?

c Author(s) 2021. CC BY 4.0 License.

[Figure]

**Figure 5: Comparison of dissolved (D) and total (T) TMs concentrations**  to previous studies in **the eastern and western Mediterranean Sea.**

The dissolved and total TM concentrations in the PEACETIME rains were lower than those reported in coastal areas (eastern Basin: Özsoy and Örnektekin, 2009; Al-Momani et al., 1998; Kanellopoulou et al, 2001 and western Basin: Guieu et al., 1997; Guerzoni et al., 1999b; Chester et al., 1997; Losno, 1989; Frau et al., 1996) (Fig. 5),  especially for the background Rain ION. This suggests the probable effect of both local anthropogenic influence at coastal sites due to higher aerosol concentrations in comparison to the remote Mediterranean (Fu et al., in prep.) and due to the reduction of anthropogenic emission for some elements since most of the referenced works on coastal rainwaters date from the late 1990s. This is particularly true for  Pb and Cd whose emissions have strongly decreased over the last decades, notably due to removal of lead in gasoline and reduction of coal combustion (Pacyna et al., 2007). This has resulted in a clear decrease in the particulate concentrations of these metals in the Mediterranean atmosphere (Migon et al., 2008), consistent with the fact that concentrations in the PEACETIME rains are one to two orders of magnitude lower than reported in

**Commented [RS(-S38]:** Are you comparing like with like here? You state that concentrations likely decrease offshore and, although emissions of TMs may have decreased, if the literature data is from coastal sites you don't have direct evidence for this decline over the open ocean. However, you could infer it from reductions in leaded fuel and coal combustion. Is the decreasing atmospheric input evident in the full depth water column samples? Perhaps this is what the Fu et al data shows but the reader can't check this as the reference is 'in prep'.
Later you mention the Dulac thesis as evidence. This paragraph needs rewriting and making more concise in order to strengthen your final, important statement.
As it stands, this section is too speculative.

c Author(s) 2021. CC BY 4.0 License.

[Figure]

the literature. For these metals, the discrepancy is also observed between the concentrations in our open-sea rain samples and the concentrations (state which elements) measured in three rains collected at sea in April 1981 (Dulac, 1986), confirming that the large decrease of concentrations could be related with to the decrease of in anthropogenic emissions. Thus, our results show the former literaturedata from before [year] cannot should not be used as a current reference about for coastal rain composition due to recent environmental mitigation on of metals TM emissions.

**Commented [RS(-S39)]:** Your rain was open ocean not coastal so doesn't really show that historical coastal TM concentrations are higher. The comparison is with the thesis. You could say that coastal values are not representative of open ocean values generally

*4.1.2. Enrichment factors*

**Commented [RS(-S40)]:** This section jumps around a bit, Rain ION then Rain FAST, the Rain ION again, then both, etc. It would be easier to read if one station was discussed and the other one contrasted to it

Enrichment Factors (EF) and solubility values of TMs and P observed during the two rain events were very contrasted (Fig. 6). In Rain ION, almost all elements were significantly enriched relative to the earthupper continental crust (EF >10, and up to ~$10^3$ for Cd and Zn), whereas in Rain FAST, only Zn (73), Cd (48) and Mo (15) were slightly enriched. Only Ti, Fe, and Mn did not present a significant enrichment (EF < 10) in Rain ION, in agreement with previous studies in the Mediterranean environment showing that these metals are mainly associated with mineral dust in atmospheric deposition (e.g., Guieu et al., 2010; Desboeufs et al., 2018). For bBoth rainsrain samples, the EF of Zn was oin average five times higher than the EF found in the rains previously studied inrain samples from coastal sites (?) in the Mediterranean region (Özsoy and Örnektekin, 2009; Al-Momani et al., 1998; Losno, 1989). However, extremely high enrichments of Zn in rainwater have also been reported from island sites in the Mediterranean Sea, for example, by Frau et al. (1996) reported with a geometric mean EFs of about~ 6500 in both crust-rich and crust-poor rains from two sites in southern Sardinia, and. Fu et al. (2017) also reported EF higher than> 1000 for Zn in atmospheric bulk deposition oin Lampedusa Island. The Zn EF at station ION is the same order of magnitude as at these island sites which suggests – something about the anthro background signal being high in the open Med.

**Commented [RS(-S41)]:** Change to dust-rich and dust-poor

**Commented [RS(-S42)]:** Do you mean atmos bulk wet depo here? If so, for clarity best to just say rain. If bulk dry you need to state that too

//The anthropogenic origin of particulate TMs and P concentrations in seawater have been reported by several studies on atmospheric deposition in the western Mediterranean (e.g., Guieu et al., 2010; Sandroni and Migon, 2002; Desboeufs et al., 2018). For example, Desboeufs et al. (2018) showed that there is a large contribution of anthropogenic combustion sources to the in P, Cr, V and Zn background deposition fluxes. Aerosol composition monitoring over the Mediterranean coastal area showed the role of land-based sources and ship traffic sources on TMs contents (Bove et al., 2016; Becagli et al., 2017). However, all these sampling sites were located in coastal areas, where it was difficult to discriminate the potential local influences. Here, even if the on-board atmospheric gas and particle measurements did not show a specific anthropogenic influence during the period of Rain ION, the particles scavenged by this rain presented a clearsuggest an anthropogenic signature for all TMs

**Commented [RS(-S43)]:** What do you mean here? To resolve the contribution of various local industries and shipping or local inputs from long range transport c Author(s) 2021. CC BY 4.0 License.

[Figure]

[Figure]

except Ti, Fe and Mn.; Hhowever for Fe and Mn, an the influence of non-crustal sources in Rain ION
is visible through a clear increase in the EF values compared to FAST (Fig. 6). This means that even
over remote the Med Sea, the chemical composition of background aerosol particles is likely
continuously impacted by anthropogenic sources.

> **Commented [RS(-S44]:** These two sentences disagree with each other. Need to be reworded.

Moreover, the EF values of TMs for Rain FAST were significantly lower than for Rain ION (Fig. 6)
but similar to Saharan rains (Guerzoni et al., 1999b; Özsoy and Örnektekin, 2009). The comparison
between dust-rich and background rains generally reveals a net difference of concentrations (at least
higher by a factor 3 in dust-rich), notably for Al, Fe, Mn and Cr (Guerzoni et al., 1999b; Özsoy and
Örnektekin, 2009). Such contrast was indeed observed for Al, Fe and Mn in the PEACETIME rains
(Fig. 5), but also for Cu and Pb. The combination of higher concentrations and EF values <10 found
in Rain FAST confirm that the dust contribution was important on deposition fluxes of many TMs.

> **Commented [RS(-S45]:** How does this study compare to the factor of three increase in concs between background and dust influenced rains – use your values – and state how this would impact EFs – more Al drives down EFs, Al predominantly from dust, etc.

> **Commented [RS(-S46]:** Vague, just say that concs were higher and EFs suggest the reason for this was the impact of the dust plume. The EFs don't really tell us this. This section needs tightening up.

*4.1.3. Solubility*

The solubility values were also larger higher in Rain ION than in the dusty Rain FAST, except for Mo
for which the difference between both rain samples wais not significant (Fig. 6). For Rain ION, TMs
and P presented solubility higher than 78%, except for Cr (38%). In Rain FAST, solubility values
<10% were observed for Al and Fe, more than 10 times lower than in Rain ION. For the other TMs,
the highest difference in solubility was observed for Pb whose solubility decreased from 97% in Rain
ION to 27% in Rain FAST. In a review on TMs solubility in Mediterranean rainwaters collected in
coastal areas, Desboeufs (2021) emphasize the large range of solubility for all the TMs: Fe (0.8-41%),
Cr (6-80%), Pb (5-90%), Ni (22-93%), Mn (16-95%), Cu (22-96%), Zn (14-99%), V (35-99%) and
Cd (72-99%). The solubility ranges found in this study were generally consistent with those reviewed
by Desboeufs (2021). In particular, the Mn solubility values in FAST (60%) and ION (92%) rains are
close to those reported by Dulac (1986) from a dust-rich (57%) and an anthropogenic (83%) rain
collected at sea in the Ligurian Sea and west of Sardinia in April 1981, respectively. Only Fe solubility
(84%) found in Rain ION was higher than the average values previously reported. In the Rain FAST,
Fe solubility was 8%, this is 10 times lower than the average Fe solubility in 10 dust-rich rains
collected on the southeastern coast of Sardinia by Guerzoni et al. (1999b), but consistent with Saharan
dust wet deposition collected in the Atlantic Ocean (Chance et al. 2015; Powell et al., 2015; Baker et
al., 2017).

> **Commented [RS(-S47]:** Big target to hit!

> **Commented [RS(-S48]:** Can you suggest a reason for the difference?

c Author(s) 2021. CC BY 4.0 License.

[Figure]

[Figure]

**Figure 6: Enrichment Factors (EF, upper panel) and solubility (%, bottom panel) of phosphorus (P) and TMs ordered by increasing EF in the two rainwater samples.**

> **Commented [RS(-S49]:** Could you make the two panels the same size and the bars the same width?

Few studies have compared TMs solubility between dust-rich and background rains in the Mediterranean. In Sardinia, Guerzoni et al. (1999b) observed an increase in solubility values from dust-rich to background rains for Al, Cr, Fe and Pb (and hardly but only a slight increase for Cd), and reported an inverse relationship between the particle concentration and solubility of Al, Fe, Pb and Cd. Similarly, Theodosi et al. (2010) showed a decrease in TMs solubility with the increase in dust load in rains collected on Crete Island. In those two studies, this the magnitude of the decrease was dependent on the considered metal specific to the TM, with Pb presenting the highest decrease in solubility. The decrease in solubility from background to dust-rich rains was also observed for P in Spain (where?) by Izquierdo et al. (2012), with values of solubility decreasing from 25% to 7%, and for Mn in offshore rains as mentioned above (Dulac, 1986).

> **Commented [RS(-S50]:** Could draw parallels with aero samples here, e.g. Jickells et al. (2016)

//Metal partitioning in rainwater can be influenced by a number of parameters, such as pH, presence of dissolved organic complexing ligands, in-cloud processinges, particle origin and load (Desboeufs et al., 1999; Bonnet and Guieu, 2004; Paris and Desboeufs, 2013; Heimburger et al., 2013). However, the particulate desert mineral dust load, reflecting the dust vs versus anthropogenic signature, is the main control of TMs solubility in the Mediterranean rainwater (Özsoy and Örnektekin, 2009;

> **Commented [RS(-S51]:** Dust loading is only one proposed control and may not be a direct control – provide explanation e.g. perhaps the CaCO3 concentration is or the reduced impact of acidic gases, low RH, etc is – ref Baker et al., 2021

c Author(s) 2021. CC BY 4.0 License.

[Figure]

[Figure]

Theodosi et al., 2010). A much lower solubility of TMs in Rain FAST than in Rain ION (except for

Mo) is consistent with the EFs in Rain FAST (Fig. 6), although no correlation between solubility and

EF values could be observed. The case of Mo is unique, since its solubility was comparable in Rains

ION and FAST despite a >20 times higher EF in Rain ION. As Mo solubility is seldom studied in the literature, we could not conclude a reason for this particular outcome. It is interesting to note that despite the desert signature, the majority of metals have solubility greater than 50% in rain FAST.

**4.2. Atmospheric wet deposition as a source of TMs to the surface seawater**

*4.2.1. Atmospheric fluxes*

As mentioned before, the two collected rains were part of large rain systems, associated with patchy rainfalls that lasted several hours or days (section 3.1). This spatio-temporal variability led to heterogeneity in both rainwater concentrations and accumulated precipitation across the studied region. Such spatial variability has been observed by Chance et al. (2015) in the Atlantic Ocean.

Moreover, even weak lateral advection can transfer surface water impacted by intense precipitation in the vicinity of the vessel. On this basis, the spatial extrapolation of wet deposition fluxes seems subject to  large uncertainty when the rain samples are not collected across the rain area (Chance et al.,

2015). To best counteract this effect, spatial variability was taken into account to quantify the total precipitation i.e. 3.5 ±1.2 mm for rain ION and  6.0 ±1.5 mm for rain FAST (see 3.1) in order to quantify the wet deposition fluxes.

From the total (dissolved + particulate) Al concentration measured in the Rrain FAST sample, we estimated the wet mineral dust deposition flux at 65 ±18 mg m$^{-2}$, assuming 7.1% Al in dust (Guieu et al., 2002). The vertical distribution of dust particles (Fig. 2b) and the absence of high Al concentrations close to the sea surface (Fu et al., in prep.) indicate that dust dry deposition can be neglected. Based on the increase in total Al in the upper 20 m of the column water following the deposition events, Bressac et al. (2021) derived an average dust deposition flux of ~55 mg m$^{-2}$ at

FAST station which is, comparable to our estimate. Although low compared to deposition fluxes reported in the western Mediterranean (Bergametti et al., 1989; Loÿe-Pilot and Martin, 1996; Ternon et al., 2010),  our flux  estimates are  similar to the most intense weekly dust deposition fluxes recorded from Corsica between 2011 and 2013 (range)

equivalent to the mean weekly flux (value) reported for Majorca during the same period (Vincent et al., 2016). The  columnar aerosol concentration during the dust event at the

FAST station was estimated to be between 0.18 and 0.24 g m$^{-2}$ (see Section 3.1.), the expected maximum values of atmospheric dust flux could be in this range. The comparison with the estimated

**Commented [RS(-S52]:** Therefore, your data does not support this argument. Disagreement within statement. The EFs close to crustal TMs in RainFAST combined with the higher concentrations point to the role of mineral dust in reducing the fractional solubility of TMs in rainwater – or the presence of mineral dust overwhelms the background signal resulting in a net decrease in fractional solubility – although the net effect is the same, I favour the latter explanation

**Commented [RS(-S53]:** Mo has a high conc in SW. If it has a predominantly marine source this could explain your uniform solubility. It would not explain your increase in EF necessarily but if there was an increase in seasalt aerosols it could do. You have the data to test this hypothesis

**Commented [RS(-S54]:** Marine source? How reliable were the solubility data? Close to LoD and blanks?

**Commented [RS(-S55]:** How large?

**Commented [RS(-S56]:** This is different than the Al proportion used in your EF calculations (Rudnick and Gao, 2003). Your EFs would be lower if you used concentrations based on 7.1% Al (Guieu et al., 2002) or your fluxes would be higher if you used the Rudnick and Gao(~ 8 %). It would be worth noting this variability. There is certainly an argument for using Saharan elemental ratios in your EF calcs (as discussed in Shelley et al. (2105)

**Commented [RS(-S57]:** Fluxes aren't measured directly so reported, calculated or estimated are better words

**Commented [RS(-S58]:** This is x3-4 higher than your flux estimates. A sentence to clarify why you support this view.

[Figure]

flux indicates that the atmospheric column was probably not totally washed-out by the short rain event.

Indeed, Fig. 2b shows that a significant depolarization was observed immediately after the rain ended on the ship, before atmospheric advection could have brought dusty air possibly not affected by rain.

Satellite products (Fig. S2) confirm that on 5 June, the dusty air mass was transported farther to the north-east from the station where it was replaced by clear air.

[Figure]

**Commented [RS(-S59]:** It would be useful to see the Al wet deposition flux as this is what you are using as the basis to estimate the bulk wet depo flux. It would also be nice to have a third panel showing the ratio or the Kd of dissolved to particulate TMs

**Figure 7: Dissolved (upper panels) and particulate (lower panels) wet deposition fluxes (µmol m$^{-2}$) for**

**the different TMs estimated from the two rains sampled on-board, considering the standard deviation**

**on the TMs concentrations and the spatial variability of total  precipitation over the area of sampling**

**(Rain ION in blue and Rain FAST in red). Note different scales on the y axes.**

The atmospheric dissolved and particulate wet deposition fluxes of TMs, derived from the chemical composition of rain samples and total precipitation, are presented in Fig. 7. Co, Mo and Cd presented the lowest fluxes in the two rainfalls. Zn and Fe fluxes were oin the same order of magnitude and were the highest dissolved fluxes compared to the other TMs in the two rains. The comparison showed that almost all the dissolved TMs fluxes were higher in the dusty rain, except Cr and Ti. For the particulate phase, the fluxes were mainly increased by the presence of mineral dust deposition for Co, Fe, Mn,

Pb, Ti and V. Our results emphasized support previous studies that report that the presence of mineral dust deposition, even here in the case of a the moderate deposition input flux reported here, enabled resulted in higher atmospheric inputs of TMs than from a low perturbed, anthropogenic background rain. We found this to mostThis is notably be the case for dissolved Cd, Cu, Mn, V and Zn and for particulate Fe and Ti with more than one an order of magnitude fluxes difference in input fluxes between theestimated from the two rain eventss. The orders of magnitude found in this study could be used as "typical a benchmark atmospheric fluxes" to estimate atmospheric inputs of TMs by a rain

**Commented [RS(-S60]:** Explain how these two parameters differ. Are they not the same thing here?

**Commented [RS(-S61]:** Explain why. They also have the greatest uncertainties – low concentrations and high blank contributions and >100% CRM recovery for Mo ?

**Commented [RS(-S62]:** Because these elements were primarily found in the particulate phase?

**Commented [RS(-S63]:** Emphasised is stretching it for data from just two samples

**Commented [RS(-S64]:** With the exception of Mn, these are predominantly considered pollution derived elements. It would be worth mentioning this.

**Commented [RS(-S65]:** Poorly soluble, lithogenic elements. Al should also be reported here. I'm assuming it showed the same behaviour as Fe and Ti?

**Commented [RS(-S66]:** Do you mean the range?

c Author(s) 2021. CC BY 4.0 License.

[Figure]

event from wet deposition to the western Med Sea. However, wWe must keep in mind, however, that annual and long-term deposition fluxes of dust-related elements (e.g. Ternon et al., 2010), but also nitrogen species (e.g., Richon et al., 2018b), in the Med Sea, are dominated by a few atypical, intense deposition events in the Med Seawhen they occurred, as is the case in many other oceanic regions (Duce et al., 1991).

*4.2.2. Comparison between TM: wet deposition inputs and marine stocks at ION and FAST stations*

Our sampling strategy of Marine sampling sequences carried outcollecting seawater before and after the rains were, to the best of our knowledge, the first direct observations intended to trace the fate of atmospheric metals TMs and nutrients in the water column after wet deposition events. The time chart of the sampling of rain and column water (surface microlayer, subsurface seawater and mixed layer)

is presented in Fig. 8. The impact of the two wet deposition events on nutrients stocks in the

Mediterranean surface waters is discussed in detail in van Wanbeke et al. (2020) and Pulido-Villena et al. (2021). To briefly summarise, bBoth nitrate and DIP increased in the ML following the rains.

Although the closure of the N and P budgets had to necessarily take into account post-deposition processes such as new nutrient transfer through the microbial food web (uptake, remineralisation, and adsorption/desorption processes on sinking particles), it was shown that wet deposition was a significant source of nutrients for ML during the cruise (roughly what %?). We focus here on the role of TMswet deposition as a source of metals TMs to the column water. To do so, we estimated the potential enrichment of the SML and ML from the rain by calculating the difference (delta) in TMs stocks before and after rains.

[Figure]

**Figure 8: Sampling chronology during the ION and FAST stations for SML, SSW and ML. The blue**

**periods correspond to rainfall in the station area (after ERA 5 reanalysis and radar imagery, see**

**section 3.1). Samplings were was performed 4 days and 2 days before and 2 h after Rain ION, and at**

**a higher frequency at the FAST station: 57, 37 and 7.5 hours before and 4.5, 12, 24 hours after Rain**

**FAST. SML and SSW samples could not be collected immediately before and after the rains because**

**of bad weather conditions, and were collected 3 and 4 days before Rain ION, and 57 and 20 h before**

**and 30 h after Rain FAST.**

**Commented [RS(-S67]:** What does this mean? All elements discussed or only lithogenic elements?

**Commented [RS(-S68]:** I'm wondering if concentrations or partitioning are a better choice of word than stocks (this applies throughout the paper)

**Commented [RS(-S69]:** … in an open ocean setting

**Commented [RS(-S70]:** Move to Methods

**Commented [RS(-S71]:** Move to methods

**Commented [RS(-S72]:** Given how quickly the SML TM concentrations respond to atmospheric inputs and their relatively short residence times in the SML (hours – Ebling and Landing, 2015), how can you be sure that your SML sampling resolution was capturing the impact of the wet deposition events of the SML, especially given that wind speed (and sea state) is a critical factor in determining the integrity of the SML?

c Author(s) 2021. CC BY 4.0 License.

[Figure]

[Figure]

At ION, no SML sampling was done after rain, preventing the study of the rain effect. For the ML,
the large variability in total and dissolved stocks between the two casts ML25 and ML27 before the
rain makes the establishment of a background concentration levels before rain difficult. ML27 was
used for theas initial conditions since as it wais the closest sampling from point to the post-rain
measurements sample collection (ML29). As mentioned previously, dusty rain deposition over the
FAST station area started on 3 June. Bressac et al. (2021) showed that the dust signature, traced by
changes in Al and Fe stocks in the ML, was already visible from the ML03 sampling. We defined the
enrichment of seawater layers as the difference between the maximum stocks after rains (from SML
04 or 06 and ML05+4) and the initial seawater stocks (SML02 and ML02).

At ION, only particulate Cu (+27%) and Zn (+44%) stocks increased in the ML after the rain. Even if
the dissolved forms of Cu and Zn predominated in the ML, this increase was accompanied by
increasing Kd values, i.e. in the particulate/dissolved partitioning (0.07 vs 0.12 for Cu and 0.14 vs 0.2
for Zn). This was also the case for Fe (Kd increased from 2.6 to 4.3), although no significant difference
(<5%) was observed on the particulate Fe stock. The Kd values in the ION rain sample being higher
than in the marine stock before the rain, that suggests the wet deposition at ION is mainly an additional
source of particulate TMs.

At FAST, the ML stocks increased in the particulate phase for Fe (+61%), Mn (+15%) and Zn (+9%)
and in the dissolved phase for Cu (+9%), Fe (+46%), Pb (+8%) and Zn (+15%) (Fig. 9). In addition
of to the marine inventories, the particulate TMs inputs by rain was also observed on Kd values and
total X/Al in the ML. For example, Kd(Fe) increased from 0.14 to 0.17 in ML and its Kd was 0.25 in
the rain. Even for Ni, for which no change in stock could be evidenced, Kd(Ni) decreased from 0.1 to
0.07 and its Kd in the rain was 0.006. For Mn/Al, the Kd which fell from 0.27 before the rain to 0.008
after the rain (ML05+4), in accordance with the rain ratio (0.004). In the SML, the dissolved and
particulate stocks increased following rains for all TMs, from a factor 1.5 (Mo) to 10 (Fe) for the
dissolved phase and from a factor 1.6 (Ni) to 67 (Fe) for the particulate phase (Fig. 9).

**Commented [RS(-S73]:** As it reforms so quickly and rain disrupts it, how can you be sure that the SML sampled after the rain at FAST was in contact with the atmosphere and accumulating RW TMs?

**Commented [RS(-S74]:** Suggesting that there was a surface advective current?

**Commented [RS(-S75]:** Dissolved or particulate or both?

**Commented [RS(-S76]:** I wonder if this is because of the mismatch between SML sampling and rosette casts.

**Commented [RS(-S77]:** Additional to what?

**Commented [RS(-S78]:** Particles scavenging dissolved TMs?

**Commented [RS(-S79]:** Again, the SML samples may not have been representative of the wet depo inputs. In contrast, the ML likely was

[Figure]

[Figure]

delta (µmol or nmol m$^{-2}$)

**Figure 9: Comparison between TMs wet deposition fluxes (in green) and TMs marine stock deltas (before and after the rain) in the SML (in blue) and in the ML (in red) at FAST. Dissolved = upper panels and particulate = lower panels. Marine stocks increases are expressed in absolute values (Cd, Co and Pb stocks in nmol m$^{-2}$, and the other TMs in µmol m$^{-2}$) and in relative values (%). N.E. = not enhanced (increase <5%).**

The comparison between the observed enhancements in the SML stocks and the rain inputs at FAST (Fig. 9) indicates that the atmospheric  inputs can explain  all observed deltas. Indeed, the atmospheric particulate and dissolved fluxes of TMs were 2 to 4 orders of magnitude higher than the mean stocks present in the SML, except Mo which was in the same order of magnitude. In the ML, the magnitude of atmospheric particulate inputs was higher or similar to the particulate marine delta of Fe, Mn and Zn. For Cu, Fe, Pb and Zn, the increase in dissolved stocks within the ML was 2 to 10 times higher than what could be provided from the atmospheric inputs. As described in Guieu et al. (2020), marine dynamic conditions at FAST were favourable to observe any change in the water masses strictly attributed to external inputs coming from the atmosphere on a short time scale. However, a cumulative effect of previous and surrounding wet deposition events could explain this difference between atmospheric inputs from rain FAST and increase of marine dissolved stocks. We cannot exclude the possibility of lateral transport of  TM from surrounding waters being enriched by the rain events of 3 June  for example, as revealed by the increase in the 0-20 m Fe and Al inventories (Bressac et al., 2021). Another hypothesis to explain that higher stock increases of metals in the ML than the one derived from atmospheric deposition is related to post-deposition processes. Indeed, once deposited, the atmospheric particulate fraction could still be partly solubilized in seawater, as the solubilisation of TMs (e.g. Fe) could occur over several hours or days (Wagener et al., 2008; Wuttig et al., 2013; Desboeufs et al., 2014). This could lead to an underestimate of the

**Commented [RS(-S80]:** If we assume that the SML samples were representative of this layer before and after rain (and even if we don't), the similarity in Mo concentrations suggests to me that this element is not primarily delivered by atmospheric inputs

**Commented [RS(-S81]:** From rain or total atmos depo?

**Commented [RS(-S82]:** Days?

**Commented [RS(-S83]:** Some ML residence time calculations could be of use here

**Commented [RS(-S84]:** Many people have shown a delayed response of the dissolved pool to atmospheric inputs c Author(s) 2021. CC BY 4.0 License.

[Figure]

[Figure]

dissolved TMs from atmospheric inputs. Moreover, the time lag between the rain and the first SML

sampling (1 day) does not allow us to conclude on the role played by SML as a "trap" of the added dissolved metals by rain. However, results showed clearly the increase of dissolved TMs in the SML

even 24 h after the rain (Fig. 9). Even if this increase could be due to dissolution processes (Tovar-

Sanchez et al., 2020), we cannot exclude that the residence time of dissolved atmospheric TMs in the

SML was sufficient to mask the atmospheric inputs in the ML05+4 sample. It is also known that dissolved concentrations in the ML are subject to various biological processes such as phytoplankton uptake (Morel et al., 2003). The comparison between ML05+4 and ML05+12 samples performed after the rain shows that the dissolved and particulate stocks decreased quickly for all the TMs (not shown), in agreement with the predominance of removal processes (sedimentation, biological transfer, adsorption) on these stocks. However, the rate of decrease depended on the TMs, showing that some removal processes predominate over others depending on the metal. For example, the dissolved

TM decreases could correspond to scavenging onto particles, which is a common physicchemical process occurring in the ocean for Fe (Wagener et al., 2010; Bressac and Guieu., 2013)  Co (Migon et al., 2020) and many other TMs.

//Finally, our results show that dusty wet deposition was a net source of all the studied trace metals for SML both in the dissolved and particulate fraction. For ML, atmospheric dust inputs were also a net source of particulate Fe, Mn, and Zn, and dissolved Cu, Fe, Pb and Zn. Due to various marine post-deposition processes, it  is more complicated to observe the effect of wet deposition on dissolved stocks, explaining why the SML and ML particulate stocks were more impacted by rain than the dissolved stocks. On a timescale of hours, the Fe inventory was the most impacted by the dusty rain input, both in dissolved and particulate phases, confirming that the dust-rich rains are a net source of Fe to the surface Mediterranean Sea (Bonnet and Guieu, 2006, Bressac et al., 2021).

*4.2.3. Comparison between TMs wet atmospheric inputs and marine stocks in the *

*western and  central Mediterranean Sea*

As observed from dissolved TMs stocks measured before and after the rains, a large part of the uncertainties in the data analysis results from various removal processes of TMs after wet deposition, which could have time resolution shorter than the sampling step. In order to limit the effect  of these potential processes  on data analysis, here we further study the role of wet deposition by comparing atmospheric dissolved fluxes to marine dissolved stocks by using TMs profiles in the ML at all 13

marine stations, i.e. 22 ML samplings, throughout the whole cruise (Fig. 10). Indeed, considering that the collected rains were originating from large rain systems covering more than 50 000 km$^2$ around the sampling zone and were typical of Mediterranean wet deposition, we hypothesized that they could

> **Commented [RS(-S85]:** It would be useful to see this – put in Supplementary Materials

> **Commented [RS(-S86]:** I'm not completely convinced it does because of the resolution problem. It certainly suggests it might given Tovar-Sanchez's findings of increases 24 h after rain but the signal is likely to be (significantly) diminished after this time.

c Author(s) 2021. CC BY 4.0 License.

[revised manuscript text omitted]

---

## Author Comment (AC1)

Refere 1: Rachel Shelley

*We thank Rachel Shelley very much for their valuable review of our manuscript. Below are our detailed responses to her questions and comments. The reviewer's comments are in plain text, the author's answers in italic, quotes from the manuscript are in quotation marks. All English mistakes have been corrected in the revised manuscript from manuscript with tracks changes provided by the reviewer (these corrections are not detailed in this reply). We provide with this reply a revised version of the manuscript and a track version allowing to visualizing changes from the submitted manuscript.*

General comments

The manuscript, 'Wet deposition in the remote western and central Mediterranean as a source of trace metals to surface seawater', by Desboeufs et al. presents data on trace metals (TM) in rain and surface water samples. The current title is generic and could be more descriptive such as 'A comparison of two contrasting wet deposition events in the remote western and central Mediterranean and their impact of trace metal behaviour in surface waters.'

The authors present rainwater TM (12 elements), Al, P and N species concentrations and EFs, and discuss these parameters in terms of their fluxes and impacts on concentrations of dissolved and particulate fractions in the upper 20 m of the water column, including the SML. The relatively large number of parameters under discussion results in rather a long paper. Nevertheless, this dataset is the first of its kind for the remote Mediterranean Sea and highlights differences between TM concentrations in rainwater from studies (mainly) in the 1990s conducted at coastal locations. Therefore, this study provides valuable new data from a relatively remote region that is influenced by anthropogenic emissions from Europe, episodically interspersed by pulses of Saharan dust, both of which are delivered to the surface ocean via atmospheric deposition (wet and dry). Wet deposition is thought to have a disproportionately high influence on TMs in surface waters as they are thought to be delivered already in a soluble (and potentially bioaccessable) form. This study illustrates that only some of the TMs studied were delivered predominantly in soluble form; an important point that the authors could emphasise more. Furthermore, rain samples are notoriously challenging to collect at sea. Having paired water column and rain samples is a valuable addition to the scientific literature. I recommend that this manuscript is accepted pending major revisions, given that a restructuring is suggested.

**Specific comments**

There is no mention of dry deposition – it is important to note that dry deposition is also an important source of TMs.

*R: We are agreed with the reviewer. In general dry deposition could be a source at least equal and even higher to wet deposition. However, the Mediterranean climate with short periods of intense rains favours the role of wet deposition. It is the reason why the paper is here focused on wet deposition. This information was not mentioned, so we have added a sentence in this purpose in the introduction, p2 L86: "Due to Mediterranean sporadic and intense storms, the rain events by scavenging loaded air masses with anthropogenic aerosols or Saharan dust could lead to higher deposition TM fluxes than dry deposition (Desboeufs et al., 2021)."*
*Moreover, we have added a specific comment in the conclusion: "As the wet deposition fluxes decreased since the 90's due to mitigation, it is highly probable that the dry deposition fluxes were also*

*changed. Further measurements on dry deposition in open Med Sea are needed in order to estimate its contribution in TM atmospheric inputs."*

Specific values from this study are needed in the Results section text. I have made suggestions in the text as to where these would be of value.

*R: We have added values where requested, e.g. section 3.3., p16, L422: " The TMs were mainly in dissolved forms in the ML (Kd from 0.006 to 0.5), except for Fe, whose dissolved and particulate concentrations were in the same order of magnitude (Kd around 2)." Or L432: " However, dCo concentrations (from 10 to 20 pM) were among the lowest ones measured during stratification period in Western Med Sea (~120 pM, Dulaquais et al., 2017)."*

I suggest restructuring this manuscript to combine the Results and Discussion sections. This will make this manuscript easier to follow and stop the unnecessary repetition. It would also help to reduce the length of the manuscript.

*R: We haven't change the structuring of the paper. We thought that the structuring in two parts seems to us to be the most adapted to distinguish the primary data that could be used as they are from the interpretation that we have made of them. The different changes have reduced the length of paper and removed major repetitions.*

One of the sticking points for me is the mismatch between SML sampling, ML sampling and rain sampling. I fully appreciate the challenges associated with sampling the SML. Rain, wind speed and sea state all impact the integrity of the SML making it extremely challenging to collect a SML sample immediately prior to and after a wet deposition event. It has been demonstrated that the SML reforms rapidly when wind speed drops below a threshold and that TMs have a short residence time in the SML (minutes to hours, compared to a few days in a 20 m mixed layer). I am concerned that the delay in sampling caused by the meteorological conditions would have missed the TM dynamics in the SML in response to the wet deposition, likely skewing the results towards a higher ratio of dissolved to particulate TMs being observed. The authors note that dissolution of TMs continues after deposition. I think the data is of value but I would like to see some more discussion of the limitations and challenges of sampling the SML at the resolution that would be needed to demonstrate the 'true' impact on TM concentrations following wet deposition. The authors do go some way towards discussing the limitations of the SML sampling towards the end of the manuscript, but I feel it needs to be addressed sooner and in more detail.

*R: As noted by the reviewer, the interpretation of the SML data is greatly complicated by the fact that the samples could not be taken simultaneously with the ML sampling and 24h after the Rain FAST, and also by the potential effect of waves, rain... on the results obtained for this layer. From these observations, a large part of conclusions were speculative. Moreover, the analysis of SML was not key in this purpose of this manuscript since the aim of this manuscript was to estimate the role of atmospheric inputs as source of TMs for surface waters. The analysis of SML data made the discussion more cumbersome and masked the main highlights. In consequence, we have decided to remove all the data (data being available in the database and publishing in partly in Tovar-Sanchez et al., 2020) and discussion on the SML. However, we considered the role of SML in the interpretation and discussion of ML stocks, by using the conclusions of Tovar-Sanchez et al., 2020 (see all the revised section 4.2.1).*

I also feel that the significance of these findings is a little overstated as only two rain events were sampled. However, combined with findings from previous studies, it is apparent that the data supports the assertion that wet deposition impacts the mixed layer

inventory of TMs. As I mentioned in the General Comments, one of the most interesting findings for me was the relative proportions of dissolved to particulate fractions and how this changed in response to the wet deposition, even with the limitations associated with the SML data.

*As mentioned after in specific replies, the particulate to dissolved ratio were added in Figures 5 and 8, and these values were used for supporting discussion both on the atmospheric fluxes and on the comparison with marine stocks.*

Line 34. Define DOC: *done*

Line 44. ML not yet defined: *done*

Line 49. Give examples/define continental aerosol in brackets
*R: We don't think that we need to define here "continental aerosol", this term is used in opposition to marine aerosol to evoke the influence of continental particles emission even in the remote sea (in agreement with the cited reference). The continental aerosol in the case of Med Sea is described after in the text.*

Line 52. Biosphere is too general. I suggest replacing it with microbial as it's this community that are directly accessing the dissolved nutrients in seawater: *done*

Line 52. State which nutrients: *done*

Line 79. I suggest preceding the sentence that starts in this line with 'To the best of our knowledge, …' as there are studies that have determined concentrations of TMs in rainwater and seawater simultaneously. There are several other studies that have combined rain and seawater concentration data in recent years (e.g. Buck et al., 2010; Shelley et al., 2017) and others that have incorporated wet and dry deposition, combined with seawater concentrations and SML concentrations, although they may not have published their findings yet. What differs with this study is that a suite of elements were determined from the same location, rather than when the ship was underway, in a region of highly variable atmospheric dust inputs where no such data has previously been published. Therefore, I also suggest changing 'reported' to 'published' (line 80) to avoid instances of reporting of unpublished data at conferences or in theses.
*R: We agree so we have started the sentence as suggested and we have specified in the text what made our strategy original, p2 L89: " Yet, to the best of our knowledge, the direct impact of wet deposition on TM concentrations in surface seawater has not been studied in a same location by concurrently collecting both rainwater and seawater samples before this work, whether in the Med Sea or in other oceanic regions.*

Line 95. Possibly more important in terms of why coastal atmospheric TM concentrations might not be representative of open ocean samples is gravitational settling during atmospheric transport. Add a statement to this effect.
*R: As the majority of literature contains data reported from coastal rain sampling, this part of text is to show that these rain samples may not be representative, not only in term of concentration but also of composition of offshore deposition events. We agree that intensity and variety of emission sources is not the only factor to explain this difference, however the gravitational settling is a process among others (dispersion, chemical processes, rain-out..) to explain why the TM composition issued from continental aerosol are not representative in open sea. So we have changes the sentence by adding the processes during transport, p3 L104:   "However due to the continental and local sources of pollution and the variety of anthropogenic aerosol sources (Amato et al., 2016) and the physico-chemical*

*processes during atmospheric aerosol transport (chemical ageing, dispersion, gravitionnal settling, in-cloud reactivity..)(Weinzierl et al., 2017), the TM rain composition of the coastal zone may not be representative of atmospheric deposition to the remote Mediterranean"*

Line 109. Delete 'Research Vessel' and remove brackets from around R/V: *done*

Line 116. Delete (PC) as you don't use this abbreviation anywhere else in the text: *done*

Line 112. Change to 'The rain collector': *done*

Line 126. Reorder this list and restructure this section. The focus of this manuscript is TMs so they should come first. I suggest i) TMs, ii) DOC, iii) major seawater and atmospheric ions, iv) pH. After the list describe the methods for each parameter in the order they are listed. Currently, the list and the methods are mixed together.

*R: done: "The dissolved fraction was separated into four aliquots dedicated to i) TM determination by inductively coupled plasma coupled methods (ICP), ii) major ions by ion chromatography (IC), iii) dissolved organic carbon (DOC) determination by high-temperature catalytic oxidation (HTCO) on a Shimadzu total organic carbon analyzer (as described in van Wambeke et al., 2021a), and iv) pH measurement."+ description of sample treatments in this order.*

Line 144 and 147. Not all analytes are discussed in the manuscript. Please indicate which ones are discussed.
*R: A sentence has been added in p5, L168: "Only TMs, major nutrients, i.e. N and P forms, and Al are discussed in this manuscript."*

Line 154. More detail needed. I am not clear what the rain blanks were. What were the blanks for the dissolved fraction? Empty bottles swished out with MQ or acidified MQ - one unfiltered aliquot for total wet deposition and a filtered aliquot for the dissolved fraction?
*R: The description of blanks has been completed in p5 L175: "Blank samples were prepared by rinsing the funnel with 150 mL of ultrapure water (18.2 mΩ cm) with the same protocol of rain collection."*

Line161. Delete: 'For the sample concentration computations, we subtracted these blanks values to elemental concentrations obtained in rain samples.' Replace with 'Blank concentrations were subtracted from all sample concentrations.' State where the analytical figures of merit can be found.: *done*

Line 172. '..., detection limit for all analytes was 0.5 ppb, ...': *done*

Line 174. Inconsistent use of date format. Sometimes words - day month, sometimes - month day, sometimes dd/mm. Also include the year in the chosen date format.

*R: Done, the date format has been homogenised.*

Line 198. By rinsing the glass plate with 0.5 L of ultra-pure water – presumably this water was collected as the blank solution? If so, state this.

*R: As mentioned before, this part was for SML sampling so it has been removed in the revised manuscript.*

Line 196. Delete MQ water and replace with 18 mΩ cm-1.: *Done (unit for resistivity is mΩ.cm)*

Line 206. Word order – change to: TM samples were also collected in the water column using a titanium trace metal clean (TMC) rosette (mounted with 24 teflon-coated Go-Flo bottles) before and after the rain events (Bressac et al., 2021). Although rosette deployments were performed over the whole water column, we focus here on the 0-20 m mixed-layer (ML): *Done*

Line 217. Replace digested with UV-treated.: *Done*

Line 222. Which metals? Mo and Pb or not all samples were preconcentrated for all metals listed? Reword to make this clearer.

*R: Now metals are listed (p7, L224)*

Line 236. Not exactly true but the vast majority of Al is crustal. Therefore, it is currently accepted as one of the best proxy elements. It is predominantly crustal, not only.:
*R: "Only" has been changed by "predominantly"*

Line 239. Reword to: considered significantly enriched, which points to a non-crustal source (Rahn, 1976). :
*R: Done and completed with "For most metals, enrichment shows important input from anthropogenic sources, due to their low content in other non-crustal sources such seaspray or biogenic aerosols (Jickells et al., 2016)".*

Line 244. Change to wet deposition fluxes.: *Done*

Line 254 and many other instances. Sometimes you use ship, sometimes R/V, perhaps best to stick with one for consistency.: *We have homogenised in using "R/V"*

Line 256. Include equation and explanation for using VWM rainfall.
*R: The sentence was modified and completed with units (L273): "the wet deposition fluxes in our rain samples were calculated by multiplying the TM concentrations (µg L-1 or µmol L-1) by the total precipitation (mm).". We though that we don't need to add the equation (F= CxP), the sentence being precise enough.*

Line 261. best tool to estimate rainfall in the surroundings of the R/V because....
*R: the sentence has been completed by "since this method is a direct measurement of precipitation with both a best time and spatial resolution in comparison of model estimation."*

Line 264. Replace more or less with approximately:
*R: Approximately is not adapted here. "More or less" was used to give the extent of area where rain rates were integrated, so "more or less" has been changed by "±" (L284)*

Line 267. Integrated rain rates, not integral: *done*

Line 271. Replace layer with SML: *done*

Line 276. ML, ML depth or MLD – multiple examples throughout the text. Check for consistent use.: *done*

Line 286. Is there a reference for this approach?

*R: We could not provide a reference. Partition coefficient is a usual concept in chemistry to describe the partition of a compound between two phases (or two solvents). Here, this concept is adapted for TM distribution between particulate and dissolved phases.*

Line 291 on – Section 3.1. No need to use the word hereafter.: *OK*

Line 302. Replace surrounding with vicinity: *done*

Line 307. Replace rate with volume. Rate requires a time dimension. Word order improvement suggested in edited version of manuscript.:
*R: "rate" has been replaced by "precipitation" and suggestions of word order has been done (Line 337)*

Fig. 1. Format the date consistently: *done (now Fig. 2)*

I would like to see Fig. S3 here for comparison rather than it be buried in the Supplementary Material.: *The Figure S3 have added in Figure 2 to compare rainfall of the two rain events.*

Line 337. Reword for clarity. Suggested: The dust plume was concentrated between 3 and 4 km at the beginning of the station occupation, then expanded down to the marine boundary layer (about 500 m amsl) by the end of the day on 3 June 2017. The mass integrated concentration of dust aerosols derived...: *done*

Fig. 2. Can you devise another labelling scheme? You have panels a-c and labelling within panel a of arrows a-c, and in panel c of box a-c. All these a-cs get a bit confusing.
*R: Now Fig. 3: We changed letters (a, b, c) for panel labelling to panel position (top, middle and bottom panels) and letters (A,B,C) for selected periods to numbers (1,2 and 3).*

Line 355. Change to time then date to make it easier to follow.: *done*

Line 367. Repetition. Delete estimates.: *done*

Line 368. Delete around: *done*

Line 388. What exactly do you mean by stocks? Sometimes it seems to mean concentrations, others partitioning. I think it would be better to replace stocks with partioning, if that is what is meant.
*R: The stocks calculation is described in the section 2.7, stocks are not concentrations or partitioning but the integrations of marine concentrations in the ML. Indeed, in the section 3.3, only concentrations were discussed, so the title has been changed, it is only "Marine Concentrations" (Line 419).*

Line 389. Are you really talking about deep water masses or just water below the ML?
*R: We have precised: " water below the ML and deeper (e.g. for Fe: Bressac et al., 2021)" (L420)*

Section 3.3. Values are needed. Please add where indicated in edited version of manuscript.
*R: values have been added where indicated in commented manuscript between Lines 450 and 462.*

Line 397. Add concentration of dMo in seawater (~107 nM from the reference you cite).
+
Line 400. Did any of these studies determine pMo and/or dMo? If not, say that your ones are the first

*R: done L428: "..the highest TM concentrations in the surface seawater were found for Mo in the dissolved fraction (~120 nM), these values are the first measurements published in Med Sea and are in agreement with the high abundance of dissolved Mo in seawater in other oceanic regions (~107 nM, Smedley and Kinniburgh, 2017)."*

Line 403. Add values of dZn and pZn.: *done*

Line 404. How high? Is contamination suspected?
*R: As explaining in Line 463, the observed high concentrations remain in the range of reported Zn concentrations in the literature. However, we cannot be sure that the outlier values didn't reflect a contamination, so we have added a sentence to evoke this possibility L438: "even if we cannot exclude a possible contamination for these outlier concentrations."*

Line 405. The concentrations in the SML were lower than in the ML and Pb dominated both in dissolved and particulate phases. I don't understand, what did Pb dominate?
*R: As mentioned before all data of SML has been removed in the revised manuscript*

Fig. 4. Box plots are two words not one. Could the panels be plotted on the same scale to make comparison easier?
*R: Now Fig. 5: Scales have been homogenised and we have added the Kd values in this figure to easily observe if particulate or dissolved forms predominate in the ML for each TM. This adding enable to support all the discussion on the evolution of TM fractions after rain in section 4.2.2*

Line 418. Med to Mediterranean: *done*

Line 424. Reword. The FAST rain concentrations were within the published range, whereas the ION rain was in the low range, confirming a background signature at this station. : *done*

Line 426. Define DIN/DIP:
*R: DIN and DIP are now defined in section 2.1. " The speciation of dissolved P was estimated by determining dissolved inorganic phosphorus (DIP) from phosphate concentrations expressed as P and the dissolved organic phosphorus (DOP) from the difference between total dissolved phosphorus (TDP), obtained by ICP-MS, and DIP, obtained by IC. The dissolved inorganic nitrogen (DIN) were defined as sum of NO2, NO3- and NH4+, expressed as N."*

Line 428. Bulk deposition – do you mean wet + dry? Clarify. : *done*

Line 430. (in the case of DIN/TDP), but averaged about 100 in bulk deposition – state if this was in unfiltered rain. You have switched from talking about DIN/DIP to presenting data for DIN/TRP. Note this.
*R: TDP (total dissolved phosphorus) is the dissolved fraction of phosphorus which could be measured technically only after filtration. The switch (DIP in TDP) was just for the extreme value of ratio, i.e. 1200 not for all the sentence, but we are aware it was not clear. To clarify this, we have changed the sentence with: " with average ratio about 100, the highest reaching 170 for DIN/DIP and 1200 for DIN/TDP" (L458)*

Line 433. Include range : "*no high concentration" means "under limit of detection" so we have clarified*

Line 434. Replace emphasised with observed: *done*

Line 436. Add 'the' before Mediterranean. Wrong tense – change have been to were and include value : *done*

+

Delete Islands: *done*

Fig. 5 caption. Suggested reword: Comparison of dissolved (D) and total (T) TMs concentrations along with data from 14 former studies carried out in to previous studies in the eastern and western Mediterranean Sea.: *done in agreement with reword in the RC2.pdf (now Fig. 6)*

Line 452. Replace notably with especially: *done*

Line 456. Are you comparing like with like here? You state that concentrations likely decrease offshore and, although emissions of TMs may have decreased, if the literature data is from coastal sites you don't have direct evidence for this decline over the open ocean. However, you could infer it from reductions in leaded fuel and coal combustion. Is the decreasing atmospheric input evident in the full depth water column samples? Perhaps this is what the Fu et al data shows but the reader can't check this as the reference is 'in prep'.
+
Later you mention the Dulac thesis as evidence. This paragraph needs rewriting and making more concise in order to strengthen your final, important statement. As it stands, this section is too speculative.
+
Line 456. Switch order to Pb and Cd as the impact of a reduction in atmospheric Pb inputs is well documented. You could cite the GEOTRACES IDP as evidence here.
+
Line 464. Reword: '...related to the decrease in anthropogenic emissions. Thus, our results show the data from before [year] should not be used as a current reference for coastal rain composition due to recent environmental mitigation of TM emissions. '
+
Line 465. Your rain was open ocean not coastal so doesn't really show that historical coastal TM concentrations are higher. The comparison is with the thesis. You could say that coastal values are not representative of open ocean values in general.

*R: All the comments from Line 456-465 were about the paragraph on the comparison with TM concentrations in previous studies. We agree that the paragraph was confusing by discussing conjointly coastal/offshore concentrations and previous/recent dataset. We have rewritten the paragraph in ordering arguments:*
*"Most of the referenced works on coastal rainwaters date from the late 1990s. There is a continuous decline of TM emissions since 90's due to regulatory efforts (Pacyna et al., 2007). The subsequent decrease of the anthropogenic Cd and Pb imprint on atmospheric inputs from European coasts to open sea is well documented (OSPAR, 2008; Travnikov et al., 2012; Geotraces IDP). Since the phasing-out of leaded automobile gasoline, the decrease of atmospheric Pb concentrations is also observed in Mediterranean atmosphere (Migon et al., 2008). The low TM concentrations of ION and FAST rain samples, in particular Cd and Pb, suggest a pronounced decrease of TM inputs in open Mediterranean due to environmental mitigation on TM emissions. Moreover, the coastal deposition is generally not representative of open sea inputs, e.g. due to proximity of anthropogenic sources in coastal areas. Thus, the 90's data should not be used as a current reference for open Mediterranean rain composition."*

*Added references:*

*Travnikov,O., Ilyin I., O.Rozovskaya, M.Varygina, W.Aas, H.T.Uggerud, K. Mareckova, and R. Wankmueller, Long-term Changes of Heavy Metal Transboundary Pollution of the Environment (1990-2010), EMEP Status Report 2/2012*
*OSPAR Commission, 2008. Atmospheric deposition of selected heavy metals and persistent organic pollutants to the OSPAR maritime area (1990 – 2005). Publication 375/2008.*

Section 4.1.2. This section jumps around a bit, Rain ION then Rain FAST, the Rain ION again, then both, etc. It would be easier to read if one station was discussed and the other one contrasted to it
+
Line 474. Reword and clarification needed - For both rain samples, the EF of Zn was on average five times higher than the EF found in rain samples from coastal sites (?) in the Mediterranean region (Özsoy and Örnektekin, 2009; Al-Momani et al., 1998; Losno, 1989). However, extremely high enrichments of Zn in rainwater have been reported from island sites in the Mediterranean Sea, for example, Frau et al. (1996) reported geometric mean EFs of ~ 6500 in both crust-rich and crust-poor rains from two sites in southern Sardinia, and Fu et al. (2017) reported EF > 1000 for Zn in atmospheric bulk deposition on Lampedusa Island. The Zn EF at station ION is the same order of magnitude as at these island sites which suggests... – something about the evidence for an anthropogenic background signal in the open Med.
+
Line 486. Potential local influences - what do you mean here? To resolve the contribution of various local industries and shipping or local inputs from long range transport?
+
Line 487. As the atmospheric gas and particle measurements do not indicate an anthropogenic influence, you cannot say that the concentration and EF values confirm an anthropogenic influence – your data tells a conflicting story.

*R: Regarding the different comments on the section 4.2.1, this section has been rewritten considering Rain ION results and concluding on the anthropogenic background on scavenged particles by rain:*
*"The anthropogenic origin of particulate TM and P have been reported by several studies on atmospheric deposition monitoring in the western Mediterranean (e.g., Guieu et al., 2010; Sandroni and Migon, 2002; Desboeufs et al., 2018). For example, Desboeufs et al. (2018) showed that there is a large contribution of anthropogenic combustion sources to the P, Cr, V and Zn background deposition fluxes. Aerosol composition monitoring over the Mediterranean coastal area showed the role of land-based sources and ship traffic sources on TM contents (Bove et al., 2016; Becagli et al., 2017). As all the deposition measurements sites were located in coastal areas, it was difficult to exclude the influence of these local sources for explaining the observed anthropogenic contribution. Here, EF values showed a clear anthropogenic signature for all TMs except Ti, Fe and Mn in the offshore Rain ION sample. In particular, the EF of Zn in Rain ION was on average five times higher than the EF found in the rain samples previously studied from coastal sites in the Mediterranean region (Özsoy and Örnektekin, 2009; Al-Momani et al., 1998; Losno, 1989). Nevertheless, extremely high enrichments of Zn in rainwater have been reported from island sites in the Med Sea, for example Frau et al. (1996) reported geometric mean EF of ~6500 in both dust-rich and dust-poor rains from two sites in southern Sardinia, and Fu et al. (2017) reported EF >1000 for Zn in atmospheric insoluble bulk (wet+dry) deposition on Lampedusa Island. As previously discussed (section 3.1.1.), Rain ION was representative of a Mediterranean background marine rain event. The Zn EF at station ION was the same order of magnitude as at these island sites which suggests a high anthropogenic background signal of Zn even in open Med. More generally, the high EFs in Rain ION mean that even over the remote Med Sea, the chemical composition of background aerosol particles is likely continuously impacted by anthropogenic sources."*

Line 468. Define EF here: *it is already defined in section 2.*

Line 469. Replace Earth with continental crust : *done*

Line 471. Delete slightly. You have stated that TMs with EFs >10 are considered significantly enriched. : *done*

Line 474. Was the atmospheric bulk deposition on Lampedusa bulk wet deposition or bulk wet + dry? If so, for clarity best to just say rain. If bulk dry you need to state that too. : *done*

Line 485. New paragraph. '...particulate TM and P concentrations in seawater have...': *We did not change since the cited references are for TM and P concentration in deposition, not in seawater.*

Line 497. How does this study compare to the factor of three increase in concentrations between background and dust influenced rains – use your values – and state how this would impact EFs – more Al drives down EFs, Al predominantly from mineral dust, etc.
+
Line 505. Vague, just say that concentrations were higher and EFs suggest the reason for this was the impact of the dust plume. The EFs don't really tell us this. This section needs tightening up.
*R: the paragraph on dust contribution in EF values have been completed as suggested by reviewer (values and explanation on the effect of Al concentrations on EF values) p21, L533:*
*"The EF values of TMs for Rain FAST were significantly lower than for Rain ION (Fig. 7) but similar to Saharan rains (Guerzoni et al., 1999b; Özsoy and Örnektekin, 2009), confirming the dust signature for this rain. The comparison between dust-rich and background rains generally reveals a net difference of concentrations (at least higher by a factor 3 in dust-rich), notably for Al, Fe, Mn and Cr (Guerzoni et al., 1999b; Özsoy and Örnektekin, 2009). Here, an increase in concentrations between rains ION and FAST was observed for the majority of TMs: Al (x28), Ti (x50), Mo (x23), Fe (x13), Mn (x9), V (x5), Pb (x3.5), Co(x3) and Cu (x2) but also for P(x4) (Table 2). The combination of higher concentrations and EF values <10 found in Rain FAST show that the dust contribution was important on deposition fluxes of many TMs and P during this event. However, the high Al concentrations in rain FAST drives mathematically down EF values, masking potentially other sources signature."*

Line 516. Can you suggest a reason for the difference?
*R: We have added two sentences to suggest an explanation (p22, L558):*
*"It is known that anthropogenic Fe is more soluble than dust-bearing Fe (Desboeufs et al., 2005, Jickells et al., 2016). Regarding the evolution of TM emissions (see section 4.1.1), we suspect that this difference could be due to a higher contribution of anthropogenic signal for Fe in dust-rich rains in 90's in Sardinia that in the recent rain samples."*

Fig. 6. Could you make the two panels the same size and the bars the same width?: *done (now Fig. 7)*

Line 525. Could draw parallels with aerosol samples here, e.g. Jickells et al. (2016)

Line 535. Dust loading is only one proposed control and may not be a direct control – provide explanation e.g. perhaps the CaCO3 concentration is or the reduced impact of acidic gases, relative humidity, etc. – ref Baker et al., 2021.
*R: We agree that the dust loading is not a direct control. Here, we noted that the dust load reflects the contribution of dust vs anthropogenic particles in rain samples. Even if the dust loading have an indirect control on solubility, e.g. through modification of pH due to CaCO3 dissolution, or through the saturation effect of high dissolved Fe concentrations, the Mediterranean rains monitoring showed that*

*the TM solubility decrease with the dust load increase because high dust load implies a more crustal origin of metals (Theodesi et al., 2010).*

Line 538. Therefore, your data does not support this argument. Disagreement within statement. The EFs close to crustal TMs in Rain FAST combined with the higher concentrations point to the role of mineral dust in reducing the fractional solubility of TMs in rainwater – or the presence of mineral dust overwhelms the background signal resulting in a net decrease in fractional solubility – although the net effect is the same, I favour the latter explanation.

*R: We agree that the arguments on this section were not very clear and that the discussion was not supported by the results. We have rephrased all this section and described more in details the different steps of the discussion from reviewer's proposition:*

*"From our two rain samples, it is difficult to propose a control explaining the difference in solubility values. However, the pH values were very close in the two samples (Table 2), excluding a pH effect on solubility values. A much lower solubility of TMs in Rain FAST is consistent with the EFs indicating a crustal origin of TMs in Rain FAST (Fig. 7). As discussed before, background atmosphere in Med Sea seems to be continuously influenced by anthropogenic particles. Even in dusty-rain FAST, it is highly probable that a part of metals presented an anthropogenic imprint, not visible on EF values but with solubility similar than in Rain ION. Thus, the decrease of solubility between the two rain samples could be either due to the lowest solubility of TMs in mineral dust (as suggested by from aerosol leaching experiments), either results from the presence of mineral dust which by increasing the TM total concentrations overwhelms the anthropogenic background signal."*

Line 540. Mo has a high concentration in SW. If it has a predominantly marine source this could explain your uniform solubility. It would not explain your increase in EF necessarily but if there was an increase in seasalt aerosols it could do. You have the data to test this hypothesis. How reliable was the Mo solubility data? How close to LoD and blanks?

*R: We had indeed completely neglected the marine origin of Mo in our discussion to explain the non-crustal part. We are very grateful to the reviewer for directing us to this explanation. The uncertainties of solubility were displayed on Figure 7, and confirm the solubility stability is significant in the two samples. So we have used this explanation in EF discussion and completed the arguments by calculating EF ratio in rain relative to seawater (p20, L504):*

*"Mo is the most abundant TM in seawater (Smedley and Kinniburgh, 2017) and in particular in Med Sea (see section 3.3). Thus, at the difference of other TMs, the non-crustal part of Mo could be associated to seasalt aerosols rather than anthropogenic signal. The EF ratio relative to seawater ( (Mo/Na)seawater = 8.9 $10^{-7}$ in mass ratio, Millero, 2013) were 7.4 for Rain ION and 4.6 for Rain FAST confirming the marine origin of this element in both rain samples."*

*And mention that in solubility discussion (p23, L587): "As discussed from EF values Mo was associated to seasalt aerosols in both rain samples, explaining the similarity of solubility."*

*Moreover, due to the marine origin of Mo, we have excluded it in the comparison between atmospheric fluxes and marine stocks (section 4.2.2)*

Line 551. How large? *done: "here ~25% RSD for total precipitation and 1 to 74% RSD for concentrations"*

Line 566. This is different than the Al percentage used in your EF calculations (~ 8% Rudnick and Gao, 2003). Your EFs would be lower if you used concentrations based on 7.1% Al (Guieu et al., 2002) or your fluxes would be higher if you used the Rudnick and Gao %. It would be worth noting this variability. There is certainly an argument for using Saharan elemental ratios in your EF calcs (as discussed in Shelley et al., 2105).

*R: We used 7.1% Al to estimate the dust flux in order to compare this value with the flux value found by Bressac et al. (2021) on the same event since they used this value. Doing the flux calculation with 8% Al gives a value of 58 mg. m$^{-2}$ which falls within the uncertainty of the dust flux given here. The variability related to Al content is lower than the one related to the total precipitation. We do not believe that a discussion of the variability of dust flux is necessary here, since the absolute value is not relevant to the discussion, which focuses on the magnitude of the event (only the order of magnitude is key and it is not modify by the using of different Al content).*

Line 563. Fluxes aren't measured directly so reported, calculated or estimated are better words.: *done. We changed by "calculated"*

Line 565. Reword: The aerosol columnar concentration during the dust event was estimated to be between...: *done*

Line 566. This is x3-4 higher than your flux estimates. A sentence to clarify why you support this view.:
*R: the explanation is already provided in the next sentences suggesting that all the aerosol column was not totally washed-out by rain as observed on depolarization values from lidar measurements.*

Fig. 7 caption. Add 'Note different scales on the y axes.': *done (Now fig. 8)*

Fig. 7. It would be useful to see the Al wet deposition flux as this is what you are using as the basis to estimate the bulk wet deposition fluxes. It would also be nice to have a third panel showing the ratio of dissolved to particulate TMs – or Kds.: *done (Now Fig. 8)*

*During the reprocessing of Figure 8, we identified that we had a problem of unit conversion in the used script to produce the figure from the dissolved TM concentrations in pM in the rain ION. In consequence, all the estimated fluxes were higher than a factor 10. So we have corrected all these fluxes in the revised figure.*

Line 579. Explain how these two parameters differ. Are they not the same thing in the context of this work?:
*R: It is probably an English problem since here the two parameters are "chemical composition" and "total precipitation" which are not the same thing. We have reworded to remove this ambiguity: "derived from the chemical composition and total precipitation of rain samples"*

Line 579. Explain why. They (Co, Cd, Mo) also have the greatest uncertainties – low concentrations and high blank contributions:
*R: The uncertainties on Co, Cd and Mo are now described in several parts of the manuscript, notably in section 2.1. for LoD and 3.2 for uncertainties on their concentrations. We did not think that we don't need to re-mention here this fact (moreover, the uncertainties appear clearly on the figure 8).*

Line 583. Because these elements (Cr and Ti) were primarily found in the particulate phase? *We have added a comment to explain this point: "due to their low solubility in rain FAST"(p25, L637)*

Line 584. Emphasised is stretching it for data from just two samples
*R: The sentence starts by "Our results suggest", we believe we have exercised due restraint in extrapolating our results.*

Line 586. With the exception of Mn, these are predominantly considered pollution derived elements. It would be worth mentioning this.

*R: We have added a sentence to this effect: "Yet, these former elements (except Mn) are usually considered issued from anthropogenic sources."*

Line 587. Poorly soluble, lithogenic elements. Al should also be reported here. I'm assuming it showed the same behaviour as Fe and Ti?
*R: We didn't mention Al in this part which is focused on trace metals. Al being used for calculating dust flux, it is also poorly soluble and indeed its behaviour is very close to Fe and Ti.*

Line 596. Do you mean the orders of magnitude range? Say how many orders of magnitude.
*R: No, we mean "orders of magnitude", not range, e.g. pNi fluxes around $10^{-2}$ µmol m$^{-2}$, whereas pCd fluxes are around $10^{-4}$ in case of wet dust deposition.*

Line 590. What does 'dust-related elements' mean? All elements discussed or only lithogenic elements?
*R: Dust-related elements are also the lithogenic elements. It is a question of vocabulary, in the atmosphere community working on desert dust emissions and transport, the term lithogenic is not used unlike the oceanographic community. To precise, we have added desert dust-related elements: Fe, Mn and Ti (p26, L647)*

Section 4.2.2. I'm wondering if concentrations or partitioning are better words than stocks (this applies throughout the paper).
*R: Here, they are neither concentrations, nor partitioning. As described in the method, we have integrated TMs concentrations on the depth of mixed layer. The result is so the total content of TMs by m-2 in the mixed layer. Another term could be inventories of TMs. We used "stocks" since that implies also the notion of availability of these TMs.*

Line 594-597. Move to Methods.: *done*

Fig. 8. Move to methods + Line 671. It would be useful to see this – put in Supplementary Material.
*R: We have added a section 2.4 "Concurrent sampling strategy" detailing the information about sampling time-resolution and calculation of ML enrichment after rain (Now fig. 1)*

Fig. 8 caption. Given how quickly the SML TM concentrations respond to atmospheric inputs and their relatively short residence times in the SML (mins - hours in Ebling and Landing, 2015), how can you be sure that your SML sampling resolution was capturing the impact of the wet deposition events on the SML, especially given that wind speed (and sea state) is a critical factor in determining the integrity of the SML?
+
Line 614. As it reforms so quickly and rain disrupts it, how can you be sure that the SML sampled after the rain at FAST was in contact with the atmosphere and accumulating TMs from wet deposition?
+
Line 616. Suggesting that there was a surface advective current?
+
Line 619. Dissolved or particulate or both?
+
Line 623. I wonder if this is because of the mismatch between SML sampling and rosette casts.
+
Line 632. Again, the SML samples may not be representative of the wet depo inputs. In contrast, the ML likely was

+

Line 655. Some ML residence time calculations could be of use here

+

Line 648. If we assume that the SML samples were representative of this layer before and after rain (and even if we don't), the similarity in Mo concentrations suggests to me that this element is not primarily delivered by atmospheric inputs

+

Line 661. Many people have shown a delayed response of the dissolved pool to atmospheric inputs and hypothesised why there is a lag.

+

Line 671. I'm not completely convinced it does because of the resolution problem. It certainly suggests it might given Tovar-Sanchez's findings of increases 24 h after rain but the signal is likely to be (significantly) diminished after this time.

*R: Figure 8 is now Figure 1. All the discussion excludes now the case of SML and the case of Mo due to its marine origin. Moreover, as mentioned in the initial manuscript "For ML, the large variability in total and dissolved stocks between the two casts ML25 and ML27 before the rain makes the establishment of a background concentration levels before rain difficult.". Indeed at the difference of FAST station (Guieu et al., 2020), we have not the proof that the surface lateral advection was limited during ION station and hence that the ML sampling "before" rain (ML27 in figure 1) which occurred 2 days before the rain, was really representative of initial conditions. In consequence, in order to clarify our discussion on the role of atmospheric inputs in the marine TM stocks, we have decided to exclude also the results concerning ION station and to keep only the case of FAST Station which the most constrained both by time-and depth- resolution samplings and by analysis of dynamic conditions of water masses (Guieu et al., 2020). On this basis and the reviewer's comment, all the section 4.2.2. was completely rewritten by integrating Kd values and residence time calculations (from Bressac et al., 2021 and Tovar-Sanchez et al., 2020) and Figure 9 was simplified. The interpretation of results was also discussed in term of post-deposition processes.*

Line 629. Additional to what? *R: we have changed by "external"*

Line 686. Delete 'at the scale' and replace with 'in the': *done*

Line 704. Which ones? Theodosi reports this for the TMs he studied but not your full suite. There are some that do, some that don't. See Jickells et al., 2016 and Baker et al., 2020. I appreciate these papers report data for the Atlantic rather than the Med but as they report data from Saharan and European air masses they are relevant.
*R: Jickells et al. (2016) show that the dust loading effect is current for all dust-related metals, as Fe or Al. By considering recent results of Baker et al.(2020), we have completed the sentence: " Fe, Mn and Pb solubility decreases with increasing dust load in Mediterranean rain samples (Theodosi et al., 2010), suggesting that this estimation is probably a maximum flux for such deposition event. However, recent studies from aerosol collected in Atlantic Ocean showed that Co and Mn solubility was little affected by dust load at the difference of Fe (Baker et al., 2020)."(p29, L761)*

Line 734. Only dissolved or dissolved and particulate?
*R: This conclusion is only for "dissolved" since the comparison in Fig. 10 was limited to dissolved fluxes and stocks*

Line 752. How about an extra statement suggesting additional dry deposition sampling to directly compare the inputs of wet and dry deposition in future? The contribution of dry deposition seems to have been overlooked in this study, in the sense that there is no comment about the contribution of dry deposition. What is thought to be the relative

contribution of wet-dry dust deposition events in the Med. Is wet deposition thought to have a disproportionally large impact on TM seawater concentrations during stratification?
*R: We have added a specific comment on dry deposition in the conclusion (p31, L309): "As the wet deposition fluxes decreased since the 90's due to mitigation, it is highly probable that the dry deposition fluxes were also changed. Further measurements of dry deposition in open Med Sea are needed in order to estimate its contribution in TMs atmospheric inputs."*

**Technical corrections**

There are many examples of incorrect agreements and syntax. I have attempted to catch them but may have missed a few. For the most part, these small errors do not impact too much on the readability of this manuscript, but it would be greatly improved by correcting them. I have included an edited version of the manuscript to help identify where these occur as they are too numerous to include in this review.

*All corrections have made in the revised manuscript*

It seems unnecessary to use the Med Sea abbreviation as it is used inconsistently throughout the manuscript.
*R: We homogenised the using of Med Sea*

Trace metal or trace element? You switch part way through.
*R: We homogenised the using of trace metals (TMs)*

Referee2:

This is an interesting study that examined the wet deposition fluxes of trace metals with co-located measurements in surface seawater and marine stocks in the Mediterranean Sea prior to and after the rain events. The study showed that wet deposition contributed to trace metals in surface seawater and marine stocks. However, there are some scientific questions that have not been addressed in the paper. The role of dry deposition of particles is undermined considering that dry deposition of trace metals is often equivalent to or greater than that of wet deposition based on a literature review of worldwide measurements (Cheng et al. 2021). The duration of the wet deposition monitoring is too short and limited to two rain events. The paper reported one event representing regional wet deposition and another event representing wet deposition from a dust episode. One question is whether these single event wet deposition fluxes can be extrapolated to seasonal or annual fluxes, which is typically what is measured in other wet deposition monitoring studies. Indeed, wet deposition does contribute to trace metals in surface seawater and marine stocks, but it is highly uncertain to what degree when compared with post deposition processes, effluent discharge to the sea, shipping pollution, etc. To better understand the relative importance of these processes, a mass balance analysis on the water chemistry is something to consider. Another important question is that based on the trace metals deposited to the Mediterranean Sea, would this result in negative effects on aquatic organisms.

We agree that dry deposition can be an important input for marine environment. However, the purpose of this paper is to study the role of wet deposition and strategy has been elaborated to limit the role of dry deposition since our conclusions are based on a comparison between wet

deposition fluxes and marine stocks. Nous somems conscient que les conclusions de cette étude sont soumises à des limitations, du fait du rôle des processus post-deposition. Toutefois, il est clair que les stocks sont augmentés avec la pluie et que même s'il s'agit de transfer lateral, cet apport est lié aux apports par els précipitaions (comme mentionné dans le texte en partie…). D'autre part, concernant les processus cités: comme effluent discharge or shipping pollution, the measuerments sont mené en zone hautière et les conclusions ne sont pas pensés pour un autre domaine qu'en offshore. Il est connu même pour des élements connus pour être fortement liés aux apports atmopshériques qu'en zone côtière, la contribution des apports atmopshériques est largement négligeable.

Concernant l'extrapolation de nos données à l'échelle de la saison ou de l'année, nous nous sommes bien garder d'en parler puisque nous avons déterminé deux pluies au printemps.. même si nous extrapolons à l'échelle de la zone de mesure, nous ne pouvons aller plus en avant dans nos conclusions sur le rôle à l'échelle annuelle ou même saisonnière, il faudrait pour cela un suivi à plus long terme et également avoir les valeurs de dépôt sec. Mais encore une fois ce n'était pas l'objet de ce papier.

Concernant l'eefft négatifs, vus les concentrations apportées par les pluies, il y a peu de chance que cela n'induise un effet négatif mais il faudrait pour cela discuter du lien entre apports atmo et développement phytoplanctonique qui ne fait aps l'objet de ce papier.

Specific comments

Line 59: The sentences emphasized wet deposition, but the importance of dry deposition of metal-containing aerosols was not discussed.

Line 110: Rainwater was collected during the period between 11 May and 10 June 2017. How many samples were collected?

Line 290: The subheading can be more detailed. E.g. Atmospheric conditions prior to rain events.

Line 303: "in the night between June 28 and 29," Why are these dates different from those in Table 1?

Section 3.1: The low sample size seems to be an issue. There was only one rain event representative of regional background conditions and one rain event representing wet deposition from a dust episode.

Lines 378-379: It is unclear if this is the dissolved or total concentrations of Fe and Zn in rain.

Section 3.2: There should more discussion on how the chemical composition between the two rain events differ, e.g. are there different sources contributing to the scavenging of metals for the ION and FAST events? Are the sources of TMs in dust natural or anthropogenic? The dissolved concentrations for metals are not very different between the ION and FAST rain events. Are they statistically different? It appears that the total concentrations were much higher for the FAST event than the ION event. Any possible explanations as to why the dissolved concentrations are much more comparable between the two events than total concentrations?

Lines 474-475: It was mentioned that the rainwater chemistry from this study cannot be compared to those from previous studies because emissions levels were much higher back then. Is it valid to compare the enrichment factors from this study with those from earlier studies? Why is the enrichment factor for Zn in this study higher than that of previous studies despite regulations on toxic trace metal emissions?

Line 535: What causes the lower solubility of TMs in high dust events?

Line 556: Is this the deposition flux over the course of the rain event? Can you quantify the time period associated with this deposition flux, e.g. mg/m2/day? Given the rain events last up to a few days, can the deposition fluxes be extrapolated to seasonal or annual fluxes?

Lines 591-592: How many intense deposition events occur in a typical year?

Section 4.2.2: Have you compared the trace metals profile in wet deposition and in seawater and in marine stocks? Are they comparable? Based on the trace metals deposited to the Mediterranean Sea, would this result in negative effects on aquatic ecosystems? Although the authors qualitatively discussed the role of post deposition processes on seawater concentrations, some data are needed to elucidate the importance of atmospheric deposition relative to the post deposition processes.

Lines 745-747: "we suggest to use the chemical composition of PEACETIME rains as a new reference for the studies TMs on wet deposition in Med Sea". I suggest rewording this statement. This is only one wet deposition monitoring study in the Mediterranean region, and the paper presented the findings from two rain events. Other studies have been conducted to capture background and dust episodes in this region. The results on the trace metals composition and their solubility in wet deposition are not particularly new compared with previous studies.

Lines 750-753: HNLC has not been defined.

---

## Author Comment (AC2)

Referee2:

*We thank the reviewer for his/her review of our manuscript. Below are our detailed responses to questions and comments. The reviewer's comments are in plain text, the author's answers in italic, quotes from the manuscript are in quotation marks.*

This is an interesting study that examined the wet deposition fluxes of trace metals with co-located measurements in surface seawater and marine stocks in the Mediterranean Sea prior to and after the rain events. The study showed that wet deposition contributed to trace metals in surface seawater and marine stocks. However, there are some scientific questions that have not been addressed in the paper. The role of dry deposition of particles is undermined considering that dry deposition of trace metals is often equivalent to or greater than that of wet deposition based on a literature review of worldwide measurements (Cheng et al. 2021). The duration of the wet deposition monitoring is too short and limited to two rain events. The paper reported one event representing regional wet deposition and another event representing wet deposition from a dust episode. One question is whether these single event wet deposition fluxes can be extrapolated to seasonal or annual fluxes, which is typically what is measured in other wet deposition monitoring studies. Indeed, wet deposition does contribute to trace metals in surface seawater and marine stocks, but it is highly uncertain to what degree when compared with post deposition processes, effluent discharge to the sea, shipping pollution, etc. To better understand the relative importance of these processes, a mass balance analysis on the water chemistry is something to consider.

*We agree that dry deposition can be an important input for marine environment. However, the purpose of this paper is to study the role of wet deposition and strategy has been elaborated to limit the role of dry deposition since our conclusions are based on a comparison between wet deposition fluxes and marine stocks in short period before and after rain. Indeed, the Mediterranean climate with short periods of intense rains favours the role of wet deposition. It is the reason why the paper is here focused on wet deposition. We have added a specific comment in the conclusion about dry deposition which should be also studied in open Med Sea: "As the wet deposition fluxes decreased since the 90's due to mitigation, it is highly probable that the dry deposition fluxes were also changed. Further measurements on dry deposition in open Med Sea are needed in order to estimate its contribution in TM atmospheric inputs."*

*We are aware that the conclusions of this study are subject to limitations, due to the fact that only two rains were studied and due to the role of post-depositional processes. We have mentioned these limitations (e.g. p27, L 691: " However, we cannot exclude that the pCu and pNi inputs were masked by the uncertainties of stock calculations.") and taken precautions about our conclusions by using "suggest" or "could.." and by mentioning always "our results". For example p29, L734: " Finally, our results show that the studied atmospheric dust event was a net source of particulate TMs and dissolved Fe and Co for ML at FAST. Even if the wet deposition delivered TMs already as soluble forms (Fig. 8), our results showed that the wet deposition constitutes only a source of some of dissolved TMs for surface waters. Due to various marine post-deposition processes, it is more complicated to observe the effect of wet deposition on dissolved stocks. The post-deposition dissolution of particulate rain inputs could represent an additional pathway of dissolved TMs supply for the surface ocean, notably for low soluble TMs in wet deposition. Thus, the dissolved atmospheric inputs could be underestimate from the only measurements of atmospheric fluxes."*

*We agree that post-deposition processes could be critical for mari,e TMs stocks, besides these processes are discussed in the revised manuscript (section 4.2.2.). However, our results showed that the stocks were clearly increased after rain and that even if it is a lateral transfer, these input were linked to*

*surrounding precipitations. It is also the reason why we estimated the atmospheric fluxes on a radius of 25 km around the R/V position. Moreover, about effluent discharge and shipping pollution, the measurements were carried in open sea with specific material and procedures to limit all contamination by other sources than atmospheric deposition. It is not intended to extrapolate our findings to coastal areas where river inputs or coastal pollution need to be considered and where atmospheric inputs are known to be often negligible.*

*The mass balance of N, P, Al and Fe on the water column was made for estimating the contribution of atmospheric inputs relative to other inputs (advection, diapycnal fluxes ..) in these elements budgets, in other publications issued from PEACETIME cruise, as mentioned from p2, L652 to 671 (van Wanbeke et al. (2020) and Pulido-Villena et al. (2021), Bressac et al., 2021). This kind of calculations was subject too much uncertainties in the case of TMs due to both the depth-resolution and time-resolution sampling of ML and water column during the cruise.*

*Regarding the extrapolation of our data to the seasonal or annual scale, we have refrained from talking about it since we determined two rain samples during the stratified period of spring, p31, L792: " The marine TM concentrations measured during the cruise being typical of Mediterranean surface seawater concentrations, we can conclude that wet deposition events were an external supply of dissolved Fe, Co and Zn for the Med Sea, and more generally for all TMs in case of intense wet dust deposition, during the period of thermal stratification." Indeed, even if we extrapolate to the scale of the measurement area, we cannot go any further in our conclusions on the role on the annual or even seasonal scale, for that we would need a longer term monitoring and also have the dry deposition values. But again, this was not the purpose of this paper.*

Another important question is that based on the trace metals deposited to the Mediterranean Sea, would this result in negative effects on aquatic organisms.

*Our measurements show that there has been a decrease in TM concentrations in rain since the 1990s. Only dissolved Co and Fe have their marine stock impacted by wet deposition event. As these metals are known for their positive physiological role on marine organisms, the probability that the atmospheric inputs induce a negative impact on marine biosphere is probably limited. However, this would require monitoring of rainfall fluxes in relation to phytoplankton growth. This is not the topic of this paper either.*

Specific comments

Line 59: The sentences emphasized wet deposition, but the importance of dry deposition of metal-containing aerosols was not discussed.

*The deposition from L62 to 69 is presented as total deposition, then L77 to 86. Even if fluxes of wet deposition could be in the same order of magnitude of dry deposition, see inferior, the rain inputs are delivered already in a soluble (and potentially bioaccessable) form. Moreover, due to Mediterranean Climate, the wet deposition were sporadic but often intense. In order to mention the contribution of dry deposition, we have added L86:" Due to Mediterranean sporadic and intense storms, the rain events by scavenging loaded air masses with anthropogenic aerosols or Saharan dust could lead to higher deposition TM fluxes than dry deposition (Desboeufs et al., 2021). Moreover, even if annual wet and dry deposition are equivalent in Mediterranean (Theodosi et al., 2010), wet deposition is known to provide soluble, and potentially bioavailable forms of TMs (Jickells et al., 2016)."*

Line 110: Rainwater was collected during the period between 11 May and 10 June 2017. How many samples were collected?

*R: One sample by rain, the detail of sampling is provided in section 2.*

Line 290: The subheading can be more detailed. E.g. Atmospheric conditions prior to rain events.

*R: We have changed the title by "Atmospheric conditions during wet deposition events"*

Line 303: "in the night between June 28 and 29," Why are these dates different from those in Table 1?

*R: The table 1 give the perio of sampling, whereas "in the night between 28 and 29 June" described the period of rain in the vicinity of the R/V position.*

Section 3.1: The low sample size seems to be an issue. There was only one rain event representative of regional background conditions and one rain event representing wet deposition from a dust episode.

*R: We are aware that this study was based on two rain samples. Rain sampling during cruise is very dependent on meteorological conditions and positions of R/V. Even if the strategy during the cruise was to chase wet deposition event, only two rain were sampled. This study was not dedicated to obtain a large database of rain composition, but was focused to estimate recent rain deposition composition and fluxes in open sea, since no data was published since 80's.*

Lines 378-379: It is unclear if this is the dissolved or total concentrations of Fe and Zn in rain.

*R: We have clarified (L413): "Regarding TMs in rain, Fe and Zn presented the highest concentrations both in the dissolved fraction and in total deposition with the same order of magnitude (10 to 25 μg L-1)."*

Section 3.2: There should more discussion on how the chemical composition between the two rain events differ, e.g. are there different sources contributing to the scavenging of metals for the ION and FAST events? Are the sources of TMs in dust natural or anthropogenic? The dissolved concentrations for metals are not very different between the ION and FAST rain events. Are they statistically different? It appears that the total concentrations were much higher for the FAST event than the ION event. Any possible explanations as to why the dissolved concentrations are much more comparable between the two events than total concentrations?

*R: This discussion is provided in the sections 4.1.2 and 4.1.3 as a function of EF and solubility values. Here, in the results section, only concentration values are given.*

Lines 474-475: It was mentioned that the rainwater chemistry from this study cannot be compared to those from previous studies because emissions levels were much higher back then. Is it valid to compare the enrichment factors from this study with those from earlier studies? Why is the enrichment factor for Zn in this study higher than that of previous studies despite regulations on toxic trace metal emissions?

*R: EF values enable to determine the origin of TMs between desert, anthropogenic or marine sources. Even if the emission have been reduced during the last decade, the EF values determine an enrichment relative to the upper crust, so this enrichment is dependent on aerosol composition. All the*

*anthropogenic sources present this enrichment whatever the intensity of their emission, it is only related to the composition of the emission.*

Line 535: What causes the lower solubility of TMs in high dust events?

*We agree that the reason of the lowest solubility for TMs associated with dust origin is not clearly explained. So we have added a sentence in L579: "However, it is known that metals that are mainly associated with crustal aluminosilicate mineral lattices such as Fe and Ti have very low solubility values, due to the difficulties to breaking bonds in the lattices (Jickell et al., 2016)."*

Line 556: Is this the deposition flux over the course of the rain event? Can you quantify the time period associated with this deposition flux, e.g. mg/m2/day? Given the rain events last up to a few days, can the deposition fluxes be extrapolated to seasonal or annual fluxes?

*Fluxes are calculated on the period of rain event. As mention before, we don't think that the extrapolation of our data to the seasonal or annual period is relevant since it is only two rain samples. Besides no discussion is carried in this sense.*

Lines 591-592: How many intense deposition events occur in a typical year?

*Dust event are sporadic, so it is not a regular phenomenon. It is the reason why the dust flux of our event is compared in the published data, L 617 to 624, in order to situate the sampled wet dust deposition event relative to the most intense event: "Although low compared to deposition fluxes reported in the western Mediterranean (Bergametti et al., 1989; Loÿe-Pilot and Martin, 1996; Ternon et al., 2010), our flux estimates are in the same order of magnitude of the most intense weekly dust deposition fluxes calculated more recently in Corsica between 2011 and 2013 (14% of fluxes >50 mg m-2) and is comparable to the mean weekly flux (93 mg m-2) reported for Majorca during the same period (Vincent et al., 2016)."*

Section 4.2.2: Have you compared the trace metals profile in wet deposition and in seawater and in marine stocks? Are they comparable? Based on the trace metals deposited to the Mediterranean Sea, would this result in negative effects on aquatic ecosystems? Although the authors qualitatively discussed the role of post deposition processes on seawater concentrations, some data are needed to elucidate the importance of atmospheric deposition relative to the post deposition processes.

*R: The atmospheric fluxes and marine stocks were calculated since these parameters are comparable at the difference of concentrations which is dependent on dilution effect. As mentioned before, it is impossible to estimate the negative effect of atmospheric deposition in this study and it is not the aim. No measurements of post-deposition processes have been quantify during the cruise, for example biological uptake, making impossible to known the importance of atmospheric deposition compared to these processes.*

Lines 745-747: "we suggest to use the chemical composition of PEACETIME rains as a new reference for the studies TMs on wet deposition in Med Sea". I suggest rewording this statement. This is only one wet deposition monitoring study in the Mediterranean region, and the paper presented the findings from two rain events. Other studies have been conducted to capture background and dust episodes in this region. The results on the trace metals composition and their solubility in wet deposition are not particularly new compared with previous studies.

*R: As mentioned in the text to our knowledge, no data on rain composition in open Med Sea was published since the 80's.In consequence, our results are the only recent measurements available for this region.*

Lines 750-753: HNLC has not been defined.

*R: done*

---

## Author Response (AR2)

Dear Dr Zhang,

Below are the point-by-point replies to the reviewer's comments and corrections. The reviewer's comments are in plain text, the author's answers in blue and italic. We provide with this reply a revised version of the manuscript.

Thank you,

Karine Desboeufs

Refere 1: Rachel Shelley

The revised version of the manuscript, 'Wet deposition in the remote western and central Mediterranean as a source of trace metals to surface seawater' by Desboeufs et al. is much improved. By removing the sections where the SML was discussed, the manuscript has become more focused. The updates made to the figures also greatly improve this manuscript. I am happy that the changes made adequately address my earlier comments and appreciate the author's detailed response to those comments. I have a few minor suggestions detailed below.

*All editorial and technical corrections requested by Dr Shelley have been made in the revised manuscript.*

Referee2:

This discussion related to mass balance of N, P, Al and Fe on the water column and contributions of atmospheric inputs relative to other inputs was not presented in the paper. I suggest a paragraph summarizing the mass balance analyses and the uncertainties associated with these calculations.

*The mass balance of N, P, Al and Fe on the water column are the topic of three other publications of the SI as mentioned in the manuscript. These papers are summarized in this manuscript: p668-684: "The impact of the dust wet deposition on nutrients stocks in the Mediterranean surface waters is discussed in details in van Wanbeke et al. (2020) and Pulido-Villena et al. (2021). To briefly summarise, both nitrate and DIP increased in the ML following the rain. Although the closure of the N and P budgets had to necessarily take into account post-deposition processes such as new nutrient transfer through the microbial food web (uptake, remineralisation, and adsorption/desorption processes on sinking particles), it was shown that wet deposition was a significant source of nutrients for the ML during the cruise. For example, atmospheric supply of phosphate could contribute to 90% of new production at FAST (Pulido-Villena et al., 2021). Bressac et al. (2021) studied the response to Al and Fe cycles to dust deposition during the cruise. They showed that total Fe and Al stocks were increased by dust wet deposition, and that the dissolved Fe atmospheric inputs were transient in the ML and were accumulated in the subsurface waters (100-1000m). The low depth-resolution of marine TM concentration samplings prevent the possibility to make TM inventories on the water column. We focus here on the role of dust wet deposition events as a source of TMs to the surface mixed-layer, except Mo due its marine origin in the rain samples."*

*This kind of calculations is subject different uncertainties inherent in the assumptions that are used to estimate the various fluxes (atmospheric inputs, advection, diapycnal fluxes ..) and biological uptake of these elements in the mass budgets, it will be too long to discuss the choice made for these assumptions in this paper which the topic is not the mass balance of these elements. However, nutrients and Fe*

*being the main elements governing marine phytoplankton growth, the purpose of this part of the manuscript is to inform that the discussion on these topics is available in other papers.*

While the authors stated that the purpose of this study was not to conduct longer term monitoring, the results of the two rain samples is quite limiting and needs to be stated. Sample size is an important indicator of the representativeness of the findings and reliability of the statistical summaries. Data comparability is another issue. Given that most wet deposition monitoring studies were conducted over a longer time period (e.g. seasonal or annual scales), is it possible to compare the fluxes obtained from this study with other studies?

*We agree that the comparability of fluxes at the scale of event with long-term measurements is an issue. This is why we compared and discussed our data with concentrations in rain samples collected in network measurements (L491-505), rather than flux values (daily or yearly period). Indeed, the fluxes obtained in this study were only wet deposition fluxes on a period of 3 weeks in 2 different locations, we do not think it is relevant to compare them with long-term measurements. As suggested by the reviewer, since the number of samples was limited, there would be no consistency in extrapolating the values of these two rain events to the scale of the year for comparison with long-term fluxes or even to use long-term daily fluxes when it is known that wet atmospheric deposition is very sporadic in the Mediterranean climate.*

The study also measured metals and metalloids that are toxic to biota; therefore, there should be some discussions on the potential negative effects of toxic metals on aquatic ecosystems. If there are no negative effects on aquatic organisms, that should be stated. If there is uncertainty regarding the negative effects or requires further study, that should be stated as well.

*We do not really understand this suggestion and what it is expected as "statement" by the referee. We mentioned in the introduction of the manuscript the previous literature showing that the TM effect could be positive or negative (L75-85). However, we cannot discuss these effects since we did not follow the biological response after rain events during the cruise. We measured atmospheric inputs of metals to the surface sea water. To our knowledge, it is not possible to assess simply by their marine concentrations whether metals have an effect on phytoplankton. It would have been necessary to do seeding experiments in culture for this (e.g. Mackey et al., 2012) or else to make estimates by considering atmospheric inputs in a biogeochemical model (e.g. Richon et al., 2018 for P). The data that we provide are dedicated to be used by this type of models (at least to constrain them) but at the stage of this study this estimate cannot be made.*

The authors mentioned, "...Moreover, even if annual wet and dry deposition are equivalent in Mediterranean (Theodosi et al., 2010), wet deposition is known to provide soluble, and potentially bioavailable forms of TMs (Jickells et al., 2016)." Dry deposition also provides soluble forms of TMs; see Muezzinoglu and Cizmecioglu (2006) for an example. According to that study, the mean soluble fraction of TMs in dry deposition were similar to those in wet deposition. Doesn't this suggest that dry deposition can also provide soluble forms of TMs? The revised sentence does not seem to justify the importance of wet over dry deposition of TMs. I suggest a brief discussion of literature comparing dry and wet deposition fluxes in the Mediterranean region. Dry and wet deposition of TMs is likely equally important in regions with high dust loadings.

Muezzinoglu, A., and Cizmecioglu, S. C. (2006). Deposition of heavy metals in a Mediterranean climate area. Atmospheric Research, 81(1), 1-16.

*We agree that dry deposition is also a source of soluble metals, as shown by numerous studies including the work of Muezzinoglu and Cizmecioglu (2006). However, in this paper, measurements have been done in urban area, where the continental aerosol sources are continuous and hence dry deposition*

*fluxes are higher than in open Sea. Moreover, contrary to wet deposition which provides immediately dissolved metals for marine surface after deposition, the dissolution of metals after dry deposition could be on several days, e.g. as mentioned in L729: "Mackey et al. (2015) show that in case of dry deposition, aerosol Co and Fe dissolution in seawater can be gradual and continue up to 7 days after contacting seawater.". To clarify this difference between these two kinds of inputs and justify the interest to focus our work on wet deposition, we have changed "soluble" by "dissolved" in the introduction L 99: "Moreover, even if annual wet and dry deposition are equivalent in Mediterranean (Theodosi et al., 2010), wet deposition is known to provide **dissolved**, and potentially bioavailable forms of TMs (Jickells et al., 2016).".*

*Concerning the requested brief discussion to compare dry and wet deposition in Med., we don't think that this comparison would not support or enrich the rationale and the conclusion of our manuscript. We think that our work on wet deposition fluxes is of interest by itself, even without comparison to dry deposition. The purpose of this paper is not to estimate the atmospheric fluxes of TMs but to characterize the TM composition of rain water in open sea and to study the impact of wet deposition on TM marine pool. Even if the annual dry deposition is higher, the wet deposition is limited in the time and intense, so the impact on marine TM pool will be different. It is not antagonistic, the study of wet and dry deposition are complementary. In this idea, we suggested the need to do new measurements for dry deposition in the abstract (L51) and in the conclusion (L821-826).*